# Assessing and improving cloud-height based parameterisations of global lightning flash rate, and their impact on lightning-produced NOx and tropospheric composition in a chemistry-climate model

Ashok K. Luhar[1], Ian E. Galbally[1], Matthew T. Woodhouse[1], and Nathan Luke Abraham[2,3]

[1]CSIRO Oceans and Atmosphere, Aspendale, 3195, Australia
[2]National Centre for Atmospheric Science, Department of Chemistry, University of Cambridge, Cambridge, UK
[3]Department of Chemistry, University of Cambridge, Cambridge, UK

*Correspondence to*: Ashok K. Luhar (ashok.luhar@csiro.au)

**Abstract.** Although lightning-generated oxides of nitrogen (LNO$_x$) account for only approximately 10% of the global NO$_x$ source, it has a disproportionately large impact on tropospheric photochemistry due to the conducive conditions in the tropical upper troposphere where lightning is mostly discharged. In most global composition models, lightning flash rates used to calculate LNO$_x$ are expressed in terms of convective cloud-top height via the Price and Rind (1992) (PR92) parameterisations for land and ocean, where the oceanic parameterisation is known to greatly underestimate flash rates. We conduct a critical assessment of flash-rate parameterisations that are based on cloud-top height and validate them within the ACCESS-UKCA global chemistry-climate model using the LIS/OTD satellite data. While the PR92 parameterisation for land yields satisfactory predictions, the oceanic parameterisation, as expected, underestimates the observed flash-rate density severely, yielding a global average over the ocean of 0.33 flashes s$^{-1}$ compared to the observed 9.16 flashes s$^{-1}$ and leading to LNO$_x$ being underestimated proportionally. We formulate new/alternative flash-rate parameterisations following Boccippio's (2002) scaling relationships between thunderstorm electrical generator power and storm geometry coupled with available data. The new parameterisation for land performs very similar to the corresponding PR92 one, as would be expected, and the new oceanic parameterisation simulates the flash-rate observations more accurately, giving a global average over the ocean of 8.84 flashes s$^{-1}$. The use of the improved flash-rate parameterisations in ACCESS-UKCA changes the modelled tropospheric composition—global LNO$_x$ increases from 4.8 to 6.6 Tg N yr$^{-1}$; the ozone (O$_3$) burden increases by 8.5%; there is an increase in the mid- to upper-tropospheric NO$_x$ by as much as 40 ppt (by volume); a 13% increase in the global hydroxyl (OH); a decrease in the methane lifetime by 6.7%; and a decrease in the lower tropospheric carbon monoxide (CO) by 3–7%. Compared to observations, the modelled tropospheric NO$_x$ and ozone in the Southern Hemisphere and over the ocean are improved by this new flash rate parameterization.

# 1 Introduction

Oxides of nitrogen ($NO_x \equiv NO$ (nitric oxide) + $NO_2$ (nitrogen dioxide)) play an important role in tropospheric chemistry by acting as a precursor to ozone ($O_3$) and the hydroxyl (OH) radical, which are the principal tropospheric oxidants (Labrador et al., 2005). As a greenhouse gas, $O_3$ is most potent in the upper troposphere, whereas near the Earth's surface it is an air pollutant, adversely impacting human health and plant productivity. The OH radical plays a critical role in the chemical cycles of many trace gases, including methane ($CH_4$) and carbon monoxide (CO), in the atmosphere.

Lightning is the dominant source of $NO_x$ in the middle to upper troposphere, and the only direct natural source remote from the Earth's surface (Schumann and Huntrieser, 2007; Banerjee et al., 2014). Lightning predominantly occurs over land in the tropics associated with deep atmospheric convection. The extreme heat in a lightning flash channel in the atmosphere, which can extend over tens of kilometres, allows for the dissociation of nitrogen ($N_2$) and oxygen ($O_2$) molecules into free N and O within the flash channel. These in turn react with ambient $N_2$ and $O_2$ to produce NO, which remains after the lightning channel cools. There is a conversion between NO and $NO_2$, during which ozone is generated in the presence of $HO_2$ and organic peroxy radicals, collectively called $RO_2$ ((Bucsela et al., 2019). A large uncertainty in the amount of $NO_x$ produced by lightning has been reported, with most global estimates ranging between 2 and 8 Tg nitrogen (N) per year (Schumann and Huntrieser, 2007).

Although lightning accounts for only approximately 10% of the global $NO_x$ source, the lightning-generated $NO_x$ (referred to as $LNO_x$) has a disproportionately large contribution to the tropospheric burdens of $O_3$ and OH (Murray, 2016). For example, although $LNO_x$ emissions are of similar magnitude as those from biomass burning or soils, their contribution to the total tropospheric ozone column is about three times larger (Dahlmann et al., 2011). This is because in the middle to upper troposphere where $LNO_x$ is directly released, the $O_3$ production efficiency per unit $NO_x$ is much higher (~ 100 molecules $O_3$ per molecule $NO_x$) than that near the surface (~ 10–30 molecules $O_3$ per molecule $NO_x$) as a result of the higher amount of UV radiance, lower concentrations and longer lifetimes of $NO_x$ (a few days rather than hours), and lower temperatures affecting ozone loss chemistry at such altitudes (e.g. Williams, 2005; Dahlmann et al., 2011). Of any major emission source, variability in global mean OH is most sensitive to $LNO_x$ (Murray, 2016). Using a global chemistry transport model, Labrador et al. (2005) observed marked sensitivity of $NO_x$, $O_3$, OH, nitric acid ($HNO_3$) and peroxyacetyl-nitrate (PAN) to the magnitude and vertical distribution of $LNO_x$. Modelling studies by Grewe et al. (2007) and Dahlmann et al. (2011) found that of all the major sources of $NO_x$, $LNO_x$ is the dominant source for tropospheric ozone (up to 40%) in the tropics and Southern Hemisphere.

$LNO_x$ is also important when studying natural variability of tropospheric composition because lightning occurrence is influenced by natural climate variability drivers such as El Niño and La Niña in the tropics. Similarly, potential changes in deep convection as a result of future climatic change has a bearing on $LNO_x$ and thus tropospheric ozone and associated radiative feedbacks, and some of these have been explored through modelling (e.g. Banerjee at al., 2014, 2018; Iglesias-

Suarez et al., 2018). In addition, lightning intensity and distribution, and its uncertainty, in a future climate has implications for projections of lightning-induced fire activity (Krause et al., 2014).

A realistic representation of LNO$_x$ source strength and its global distribution is thus of vital importance for understanding tropospheric chemistry and its impacts. In most global circulation models used for climate applications, convection is diagnosed/parameterised (i.e. clouds are not explicitly resolved) with limited cloud microphysics. A pragmatic way to predict lightning flash rates in these models is to use parameterisations based on simple physics of electrical charge and correlations between the flash rate and appropriate parameters representing convection.

The most common conceptual model used to compute the amount of LNO$_x$ in global models is

$$LNO_x = F \times P_{NO},\qquad\qquad(1)$$

where $F$ is the lightning flash rate and $P_{NO}$ is the amount of NO produced per flash. This calculation is carried out in atmospheric models by first calculating the lightning flash rate within a model grid column at every model time step, partitioning it into intracloud (IC) and cloud-to-ground (CG) flash-rate components, applying an amount of NO mass produced per flash, and then vertically distributing the calculated NO mass in the column.

Various approaches to estimate lightning flash rate in global models have been followed in the past. Price and Rind (1992, hereafter PR92) developed simple empirical parameterisations for calculating lightning flash rate in terms of convective cloud-top height over land and ocean. The model of PR92 was based on the assumption of an electric dipole structure for a thunderstorm with two equal but opposite charge volumes, separated by a distance of the order the vertical cloud dimension as developed by Vonnegut (1963) and Williams (1985). Flash-rate parameterisations based on convection parameters other than cloud-top height have also been developed, e.g. convective precipitation and upward mass flux (Allen and Pickering, 2002); convective available potential energy (CAPE) (Choi et al., 2005), cold-cloud depth (Futyan and Del Genio, 2007; Yoshida et al., 2009), maximum vertical velocity and updraft volume (Deierling and Petersen, 2008), upward cloud ice flux (Finney et al., 2014; 2016), product of CAPE and precipitation rate (Romps et al., 2014; Zhu et al., 2019), and multi-parameter regression fits (Luo et al., 2017).

The PR92 parameterisations are by far the most widely used ones by default in global chemistry transport models (CTMs) and coupled chemistry-climate models, such as GEOS-Chem (Hudman et al., 2007), MOZART (Emmons et al., 2020), CAM-Chem (Lamarque et al., 2012), ECHAM5/MESSy (Jöckel et al., 2006), and UM-UKCA (Archibald et al., 2020), perhaps primarily because they are based on convective cloud-top height which can be easily diagnosed from a model's convection scheme. Additionally, with its use of an electric dipole structure for a thunderstorm, the framework underlying the PR92 parameterisations has some theoretical support. The PR92 parameterisations also perform reasonably well. For example, Clark et al. (2017) tested flash-rate parameterisations based on cloud-top height, cold cloud depth (CCD), mass flux, convective precipitation rate, and cloud-top height with column-integrated cloud droplet number concentration, in a

global model, and found that the PR92 parameterisations had the best correlation with the observations, closely followed by the CCD based parameterisation of Yoshida et al. (2009). The PR92 scheme had a higher value of the spatial standard deviation compared to observations due to a large land-ocean contrast in this parameterisation. The quantity CCD is defined as the convective cloud-top height minus the freezing level. The thunderstorm data analysis presented by Price and Rind (1993) indicates that freezing levels remains relatively constant compared to CCD values, meaning that it is largely the cloud-top height that provides the variation in the lightning flash rate in the CCD based scheme, which suggests that the cloud-top height and the CCD based schemes would perform very similarly. Finney et al. (2014) found that the PR92 lightning flash parameterisation had considerable biases compared to satellite observations of spatial flash density but performed 2nd best (behind their parameterisation based on ice flux) out of the 5 parameterisations tested. A recent comparison by Gordillo-Vázquez et al. (2019) of six flash-rate parameterisation schemes showed a relatively good performance by those based on cloud-top height. However, a number of global modelling studies has demonstrated that the PR92 parameterisations underestimate the lightning flash density over the ocean compared to satellite observations (Allen and Pickering, 2002; Tost et al., 2007; Finney et al., 2014, 2016; Clark et al. (2017).

In this paper, we critically assess the performance of currently used flash-rate parameterisations for land and ocean based on convective cloud-top height. We also derive new/alternative flash-rate parameterisations following Boccippio's (2002) (Bo02) scaling relationships founded on theory linking thunderstorm electrical generator power and storm geometry applied to the available data. We implement in a global chemistry-climate model, viz. ACCESS-UKCA (Woodhouse et al., 2015), (a) these new parameterisations, (b) an oceanic flash-rate parameterisation suggested by Michalon et al. (1999) (Mi99), and (c) use the standard PR92 flash-rate parameterisation as a default. The model results are tested using global satellite data of flash density from the Optical Transient Detector (OTD) and Lightning Imaging Sensor (LIS) (Cecil et al., 2014), which are available as global climatological, inter-annual and seasonal distributions. The veracity of the new modelled $LNO_x$ estimates are examined through comparison of modelled and reanalysis tropospheric $NO_2$ column amounts. The impacts of the modelled $LNO_x$ based on the new flash-rate parameterisations on tropospheric composition involving $NO_x$, ozone, the hydroxyl radical, methane lifetime, and carbon monoxide are examined, including comparison with observations where available or appropriate.

## 2 The ACCESS-UKCA global chemistry-climate model

We use the United Kingdom Chemistry and Aerosol (UKCA) global atmospheric composition model (Abraham et al., 2012; http://www.ukca.ac.uk) coupled to ACCESS (Australian Community Climate and Earth System Simulator) (Bi et al., 2013; Woodhouse et al., 2015). In our simulations, ACCESS is essentially the same as the U.K. Met Office Unified Model (UM) (vn8.4) since the ACCESS specific ocean and land-surface components are not invoked as the model is run in atmosphere-only mode with prescribed monthly-mean sea surface temperature (SST) and sea ice fields, and the UM's original land-surface scheme (viz. JULES) is used. The atmosphere component of the UM vn8.4 is the Global Atmosphere (GA 4.0)

(Walters et al., 2014). The UKCA configuration used here is the so-called StratTrop (or Chemistry of the Stratosphere and Troposphere (CheST)) (Archibald et al., 2020), which, in essence, combines the tropospheric chemistry scheme described by O'Connor et al. (2014) and the stratospheric chemistry as described by Morgenstern et al. (2009). The tropospheric chemistry scheme includes the $O_x$, $HO_x$ and $NO_x$ chemical cycles, and the oxidation of CO, methane, and other volatile organic carbon species (e.g. ethane, propane and isoprene). The Fast-JX photolysis scheme (Neu et al. 2007; Telford et al. 2013) is used, and ozone is coupled interactively between chemistry and radiation. The aerosol component includes sulphur chemistry. The total number of reactions, including aerosol chemistry, is 306 across 86 species.

The atmospheric model has a horizontal resolution of 1.875° in longitude and 1.25° in latitude, and 85 staggered terrain-following hybrid-height levels extending from the surface to 85 km (the so-called N96L85 configuration). The vertical resolution decreases with height, with the lowest 65 levels (up to ~ 30 km) lying within the troposphere and lower stratosphere. The model dynamical timestep is 20 minutes, the UKCA chemical solver is called every 60 min. It is a symbolic backward Euler solver with Newton-Raphson iteration and runs to convergence, halving the step when required (see Esentürk et al. (2018) for more details). A global monthly-varying emissions database for reactive gases and aerosols is used, which includes both anthropogenic, biomass burning and natural components, whereas for carbon dioxide, methane, nitrous oxide and ozone depleting substances, concentrations are prescribed instead of emissions (Woodhouse et al., 2015).

We have backported the $LNO_x$ subroutines from a more recent version of the model (at UM vn11.0) to vn8.4. This is to ensure that any refinements that may have occurred in the new version are used in our study. However, it is found that there are no major $LNO_x$ parameterisation differences between the two versions, with the new version continuing to use the original PR92 flash-rate parameterisations, except that the amount of NO produced per flash is taken to be the same for both IC or CG flashes in the new version, which leads to small changes in the $LNO_x$ production compared to the old version.

The ACCESS-UKCA setup used here incorporates some additional modifications compared to the base UM-UKCA version 8.4, and these include:

- Dry deposition of ozone to the ocean is now based on a process-based scheme developed by Luhar et al. (2018) (the Ranking 1 configuration in their Table 1).
- Dry deposition of all relevant species is applied at the lowest model level, instead of it being distributed through the vertical extent of the atmospheric boundary layer (Luhar et al., 2017).
- The coefficient 8.53 is corrected to 4.8 in the following branching ratio expression used to compute the rate constant $k_2$ (cm$^3$ molecule$^{-1}$ s$^{-1}$) for the chemical reaction $HO_2 + NO \rightarrow HNO_3$:

$$k_2/k = \{[(530 \pm 10)/T] + 4.8 \times 10^{-4} \times P - (1.73 \pm 0.07)\}/100, \qquad (2)$$

where pressure $P$ is in hPa and temperature, $T$ is in K, and $k$ is the rate constant (cm$^3$ molecule$^{-1}$ s$^{-1}$) for the included chemical reaction $HO_2 + NO \rightarrow NO_2 + HO$. The coefficient 8.53 is correct when $P$ is expressed in Torr

(Butkovskaya et al., 2007), but it should be 4.8 when $P$ is in hPa[1]. Additionally, in the parameterisation $k = c_0 \times 10^{-12} \exp(270/T)$, the model uses $c_0 = 3.3$ for this reaction, but for $k$ in the branching ratio Eq. (2) it uses $c_0 = 3.6$, which we change to 3.3 for consistency.

The above changes lead to an increase in the modelled tropospheric ozone burden by about 12% (the first two changes by ~ 7%) and the last by ~ 5%) to 284 Tg $O_3$ and this increase is towards the global modelling average (see Section 4.2).

In the model, the tropopause is calculated every timestep. In the extratropics (latitude $\geq |28|°$), the tropopause is the pressure level of the 2-PVU (potential vorticity units) surface, and in the tropics (latitude $\leq |13|°$) it is the pressure level of the 380 K potential temperature isentropic surface. Between the two latitudes, a weighted average of the two definitions is used following the method of Hoerling et al. (1993).

## 2.1 Implementation of lightning flash-rate parameterisation in the model

In ACCESS-UKCA, the convective cloud bottom level ($H_b$) and top level ($H$) are diagnosed on a timestep basis from the UM convection scheme. This scheme represents the sub-grid scale transport of heat, moisture and momentum associated with cumulus clouds within a grid box. The scheme (at GA4.0) is summarised by Walters et al. (2014); it uses a mass flux convection scheme based on Gregory and Rowntree (1990) with various extensions to include downdraughts and convective momentum transport. It consists of three stages: (a) diagnosis to determine whether convection is possible from the boundary layer, (b) a call to the shallow or deep convection scheme for all points diagnosed deep or shallow by the first step, and (c) a call to the mid-level convection scheme for all grid points. $H_b$ is taken to be the air parcel ascent start level and $H$ is set to be the top of the ascent. On each timestep, the flash rate is then determined by using the calculated values of $H_b$ and $H$ at each grid point. Examples of evaluation of the distribution of cloud depths simulated by the UM include those by Klein et al. (2013) and Hardiman et al. (2015).

A minimum convective cloud scale needs to be specified for it to constitute a thunderstorm. In our model, a minimum convective cloud thickness ($H - H_B$) of 5 km is required for the lightning $NO_x$ to be activated. The selected threshold of 5 km is very similar to observations of the smallest vertical scale of thunderstorms presented by several researchers, viz. Price and Rind (1992, 1993), Molinié and Pontikis (1995), and Ushio et al. (2001). While prescribing a minimum convective cloud thickness of 5 km for lightning is somewhat arbitrary, having no such threshold value is unrealistic because then it would be implicitly assumed that a convective cloud always translates to a thunderstorm, and this would likely lead to unrealistically

---

[1] This inconsistency stems from a typo in the document "IUPAC Task Group on Atmospheric Chemical Kinetic Data Evaluation – Data Sheet I.A3.45 NOx15" (http://iupac.pole-ether.fr/htdocs/datasheets/pdf/NOx15_HO2_NO.pdf) in which a constant of 8.53 instead of 4.8 is mistakenly specified in the branching ratio expression in the "Rate coefficient data" table.

high flash rates. (We found that increasing or decreasing the minimum cloud thickness value by 1 km from 5 km resulted in a change of -3.2% and 1.7%, respectively, in the modelled global flash rate using the PR92 scheme.)

The model diagnosed $H$ is used in the flash-rate parameterisations, which calculate the lightning flash rate ($F$, flashes per minute) as a function of $H$ (km) (see Section 3.2).

The PR92 flash-rate expressions (discussed later in Section 3.2) were developed based on observations of individual thunderstorms. Price and Rind (1994) (see their Figure 1) developed a spatial calibration factor ($c$) to adjust these expressions for varying model resolutions:

$$c = 0.97241 \exp(0.048203 \, \Delta x \, \Delta y), \tag{3}$$

where $\Delta x$ is the longitudinal resolution and $\Delta y$ is the latitudinal resolution of the model (in degrees).

The total flash rate ($f$) within a grid cell (flashes per minute per grid box) is then calculated as

$$f_{L,O} = \frac{c \, F_{L,O} \, A}{A_c}, \tag{4}$$

where $A$ is the model grid box area (which is a function of latitude), $A_c$ is the area of model grid box centred at 30°N (Allen and Pickering, 2002), and the subscripts $L$ and $O$ to refer to land/continental and ocean/marine, respectively. Any model grid cell that has a non-zero land surface fraction is considered land for the purposes of lightning $NO_x$ calculation. Conversely, only grid cells with 100% water surface coverage are considered ocean.

We use Eq. (4) along with Eq. (3) to calculate the total flash rate $f_{L,O}$ which is then apportioned into cloud-to-ground (CG)
and intracloud (IC) flash rates using an empirical parameterisation for the ratio $z_R$ = IC/CG developed by Price and Rind (1993) (PR93) based on thunderstorm observations in the western United States. In this parameterisation, $z_R$ increases as a function of the thickness ($dH$) of the cold cloud region in thunderstorms (from 0°C to cloud top), and $dH$ is parameterised as a decreasing function of latitude. The PR93 parameterisation has been used frequently, with further validation for case studies reported by Pickering et al. (1998) and Fehr et al. (2004). Allen and Pickering (2002) and Grewe et al. (2001) used it
in global atmospheric chemistry models, with the former evaluating it for cases in the US. The averaged values of $z_R$ and the CG to total flash ratio obtained from the PR93 parameterisation in the present study are 3.14 and 0.24, respectively. These values are comparable to $z_R \sim 4$ and the CG to total flash ratio $\sim 0.2$ obtained by Barthe and Barth (2008) using $dH$ calculated directly from modelled cloud temperature and total hydrometeor mixing ratio in the PR93 parameterisation, and to $z_R = 2.64$–2.94 obtained by Boccippio et al. (2001) using satellite- and ground-based lightning observations over the continental US.
Using IC/CG measurements, Bond et al. (2002) derived a parameterisation for $z_R$ as a linearly decreasing function of latitude and obtained $z_R = 3.76$ and the CG to total flash ratio = 0.21 over the tropics (35°N–35°S).

**2.2 Production of NO per flash**

Both the moles of NO produced per flash, $P_{NO}$, and the variation of this parameter between CG and IC flashes are poorly constrained by atmospheric observations. The overall rate, $P_{NO}$, regulates the amount of nitrogen oxides produced by lightning, whereas the variation of $P_{NO}$ between CG and IC flashes regulates the level at which lightning nitrogen oxides are introduced into the atmosphere, both are critical variables. In this study, $P_{NO}$ is set at $S_f \times 10^{26}$ molecules NO per flash where the scaling factor $S_f = 2$ by default irrespective of whether a flash is IC or CG, which is equivalent to 330 moles NO per flash. Assuming a mean energy release of 0.67 GJ per IC flash and 6.7 GJ per CG flash (Price et al., 1997), with 24% of the total modelled flashes being CG, the production of 330 moles NO per flash corresponds to $9.4 \times 10^{16}$ molecules NO J$^{-1}$. If we use a mean energy release of 0.9 GJ per IC flash and 3.0 GJ per CG flash based on Schumann and Huntrieser (2007), then the NO production is calculated to be $14.2 \times 10^{16}$ molecules NO J$^{-1}$.

In bottom-up models, in addition to flash rate, $P_{NO}$ is a key source of uncertainty, with a review by Schumann and Huntrieser (2007) suggesting a range of ~ 33–660 with an average of 250 moles NO per flash, and similarly 70–700 moles (Bucsela et al., 2019). This large range, in part, reflects spatial variation in the frequency and uncertainty in the yield of CG and IC flashes, which may involve a varying level of dependencies on environmental variables such as peak current, rate of energy dissipation, channel length, air density, and strokes per flash (Murray et al., 2016).

The value $P_{NO} = 330$ moles NO per flash used in our model lies close to the middle of the range of current literature. Recent estimates include: a global average value of 310 moles NO per flash obtained by Miyazaki et al. (2014) using an assimilation of multiple satellite measurements of atmospheric composition and the LIS/OTD lightning flash data into a global CTM; 665 moles NO per flash estimated by Nault et al. (2017) using airborne observations of atmospheric composition, satellite based Ozone Monitoring Instrument (OMI) NO$_2$ columns and the GEOS-Chem model; $280 \pm 80$ moles NO per flash by Marais et al. (2018) using the OMI NO$_2$ columns and satellite based lightning data together with GEOS-Chem; $180 \pm 100$ moles NO per flash by Bucsela et al. (2019) for three northern midlatitude regions that were primarily continental; and $170 \pm 100$ moles NO per flash by Allen et al. (2019) for the tropics. The last two stem from the same OMI NO$_2$ columns and ground-based lightning measurements. Values used in calculating global estimates of LNO$_x$ include: 360 moles NO per flash by Ott et al. (2007), and 500 moles NO per flash for selected extratropical regions and 260 moles NO per flash for the rest of the globe by Murray et al. (2012).

We assume that both CG and IC flashes yield the same amount of NO, which follows studies such as DeCaria et al. (2005), Ridley et al. (2005), Ott et al. (2007, 2010) and Cummings et al. (2013). On the other hand, some studies consider or find that the less frequent CG flashes yield a greater amount of NO per flash than IC flashes (Price et al., 1997; Koshak et al., 2014; Luo et al., 2017), A few studies suggest that $P_{NO}$ may not be constant over the globe, with higher production rates in extratropics than tropics (Huntrieser et al., 2008; Murray et al., 2012) and globally variable production rates (Miyazaki et al., 2014). Differences in land and ocean production rates have also been noted. Boersma et al. (2005) found that land flashes

were ~1.6 times more productive than those over the ocean, and conversely Allen et al. (2019) estimated marine flashes to be twice as productive than those over land. Clearly, further measurements and process understanding are needed to reconcile differences in $LNO_x$ production.

Details about global $LNO_x$ reported in previous studies are given in Section 3.7.1.

In our model, the calculated amount of $LNO_x$ at a grid point location at a given time step (60 min) is distributed evenly in the vertical in log-pressure coordinate from 500 hPa to the cloud top for intra-cloud (IC) flashes, and from 500 hPa to surface for cloud-to-ground (CG) flashes (see Section 3.7.2). The method is motivated by the data analysis of Price and Rind (1993). Their observations from 139 thunderstorms cover cold cloud thickness (i.e., the cloud top height minus the freezing level) values ranging between 5.5–15 km and freezing level values between 2.7–5 km. The ratio $z_R$ = IC/CG increases from 0 to 4.6

with cold cloud thickness from 5.5 to 15 km but remains relatively constant with freezing level. We take the level below which the CG generated $LNO_x$ is distributed as the observed minimum freezing level plus half of the minimum cold cloud thickness, i.e. (2.7+5.5/2) ≈ 5.5 km. The selected 500 hPa level is closest to this 5.5 km value. Since the amount of NO produced per flash is taken to be the same for both IC and CG flashes, the partitioning of the total flash rate into the CG and IC flash rates only influences the shape of the vertical distribution, with the total $LNO_x$ released remaining independent of

the partitioning.

## 3 Flash-rate parameterisations based on convective cloud-top height and $LNO_x$

The approach of parameterising lightning flash frequency in terms of cloud-top height has its origins in the simple scaling relationships suggested by Vonnegut (1963) for the electrical power output of a thundercloud with the cloud size. Vonnegut's model assumes an electric dipole structure with two equal but opposite charge volumes, separated by a distance

on the order of the vertical cloud dimension, and a cloud aspect ratio of approximately unity. The charge transport velocity (charging current) that supports the dipole, flows in the vertical and is assumed to exist across the horizontal cross-section of the cloud. The electrical power generated is proportional to the fifth power of cloud dimension and the lightning flash rate is proportional to the electrical power generated. Williams (1985) extended this work using Vonnegut's (1963) model and available atmospheric observations over land to demonstrate from observations that (a) the convective velocity for clouds

ranging in size from a few kilometres to 17 km is proportional to the size of the cloud and (b) cloud height and flash-rate data from three US locations (New England, Florida and New Mexico) are in good agreement with the predicted slope of 5 from Vonnegut (1963). An important consequence of the last finding is that charge transport velocity must scale with cloud dimension, in the same way as convective velocity does. However, as Williams (1985) writes, this does not establish whether convective motion or falling precipitation is directly responsible for the charge separation.

### 3.1 Bocippio's (2002) extension of Vonnegut's model and derivation of scaling relationships for land and ocean

Bocippio (2002) presents a systematic physical account of Vonnegut's (1963) model and the subsequent work on this issue by Williams (1985), PR92 and others. Bocippio (2002), using the laws of electricity, derived a fundamental scaling relationship between thunderstorm electrical generator power and storm geometry, which provides a possible theoretical basis for linking lightning to thunderstorm dynamics, microphysics, and geometry. Bocippio (2002) takes Vonnegut's conceptualized thunderstorm of a quasi-steady state electrical dipole, with the horizontal and vertical scales of the two dipole charge centres comparable and varying with storm scale. The storm generator current is conceptualized as a net charge transport velocity, which maintains the dipoles. The electrical generator power (watts) is calculated as the generator current multiplied by the potential drop between the dipoles. The generator power is further expressed by assuming tangential spherical charge centres with volumetric charge density ($\pm\rho$) and radius $R$, maintained by a generator current density (= charge density × charge transport velocity). Scale similarity between horizontal and vertical dimensions is invoked, and the dipole separation (= $2R$) is assumed to vary linearly with cloud-top height, a more readily observable parameter. This assumption is based on observations that in many storms the lower negative charge region remains relatively constant in height and that most upper positive charge is carried on small ice crystals with negligible terminal velocity. The lightning flash rate is taken to vary linearly with lightning power dissipation, the latter is assumed to vary monotonically with generator power. A further significant and explicit simplification replaces charge transport velocity with storm updraft velocity $w$. With the assumptions above, the flash rate $F$ scaling relation is

$$F \sim k_1 w H^4, \tag{5}$$

where the coefficient $k_1 \sim \gamma\, \varepsilon^{-1} \rho^2$, $H$ is cloud-top height (m), $w$ is updraft velocity (m s$^{-1}$), $\rho$ is charge density (coulomb m$^{-3}$), $\varepsilon$ is permittivity constant (coulomb$^2$ joule$^{-1}$ m$^{-1}$), and the coefficient $\gamma$ has units of joule$^{-1}$. By assuming that that each flash is responsible for a constant electrostatic energy and that the variability in the charge density of the dominant dipole regions is small, Bocippio (2002) treated $k_1$ as a single constant, irrespective of whether the flash is over land or ocean.

In all the following parameterisations, $F_{L,O}$ is in flashes per minute, $H$ is in km, and $w$ is in m s$^{-1}$.

It is apparent that in the above approach, flash rate is set by storm dimension, with a small direct contribution from generator current density (which is taken to vary linearly with updraft velocity). In the case where $w(z)$ for land and ocean are considered to be power law fits of $H$, i.e.

$$w_{L,O} = k_{L,O}\, H^{a_{L,O}}, \tag{6}$$

the above approach based on Eq. (5) leads to the following self-consistent scaling relationships, where $a_{L,O}$ and $k_{L,O}$ are coefficients for land and ocean:

$$F_{L,O} = k_1 \, k_{L,O} \, H^{a_{L,O}+4},$$
(7)

$$F_{L,O} = k_1 \, k_{L,O}^{-4/a_{L,O}} \, w^{1+(4/a_{L,O})}.$$
(8)

These self-consistent scaling relationships are central to this study.

Using these scaling relationships, the data of Williams (1985), PR92 and LIS satellite data, Bocippio (2002) derived

$$F_L = 2.13 \times 10^{-5} \, H^{5.09}$$
(9)

(note there is a negative sign missing in the first exponent of this equation in Bocippio (2002)), and

$$F_O = 4.09 \times 10^{-5} H^{4.38},$$
(10)

where $k_1 = 1.4314 \times 10^{-5}$ was used in deriving the relationships (9) and (10), which are plotted in Figure 1.

**3.2 Price and Rind's (1992) (PR92) parameterisations**

PR92, following Vonnegut (1963) and Williams (1985), present a simple lightning parameterisation for calculating global lightning distributions. Using continental storm observations used by Williams (1985) to establish the fifth power dependency, PR92 derived an empirical relationship between continental lightning flash rate and $H$

$$F_L = 3.44 \times 10^{-5} \, H^{4.9}.$$
(11)

There were no direct $F$ vs. $H$ data to fit a relationship similar to Eq. (11) for the marine environment. Therefore, PR92 made
the assumption that the charge separation velocity and the convective updraft velocity are equal, noted that an increase in the intensity of convective updrafts enhances cloud electrification, and derived empirical relations based on observations between maximum updraft velocity ($w$, m s$^{-1}$) and $H$ (km) for continental and marine clouds

$$w_L = 1.49 \, H^{1.09},$$
(12)

$$w_O = 2.86 \, H^{0.38}.$$
(13)

Eliminating $H$ from Eqs. (11) and (12) yields

$$w_L = 14.66 \, F_L^{0.22}.$$
(14)

Now, PR92 made the crucial assumption that Eq. (14) is independent of location, so is valid for the marine environment too.
Thus, by equating Eq. (14) and Eq. (13) they obtained the following expression for marine environment

$$F_O = 6.4 \times 10^{-4} \, H^{1.73}.$$
(15)

(Our calculation suggests that the coefficient 6.4 in the above equation should actually be 5.94.) The relationships (11) and (15) are plotted in Figure 1, which show that the oceanic flash frequencies that are roughly 2 to 3 orders of magnitude smaller than those obtained for continental clouds.

According to Boccippio (2002), the derivation of Eq. (15) includes significant formal inconsistency and yields non-physical cloud-height predictions upon inversion and other non-physical behaviours, e.g. inverse relationship between updrafts and cloud heights.

The PR92 parameterisations (11) and (15) are widely used in global chemistry transport models and coupled chemistry-climate models.

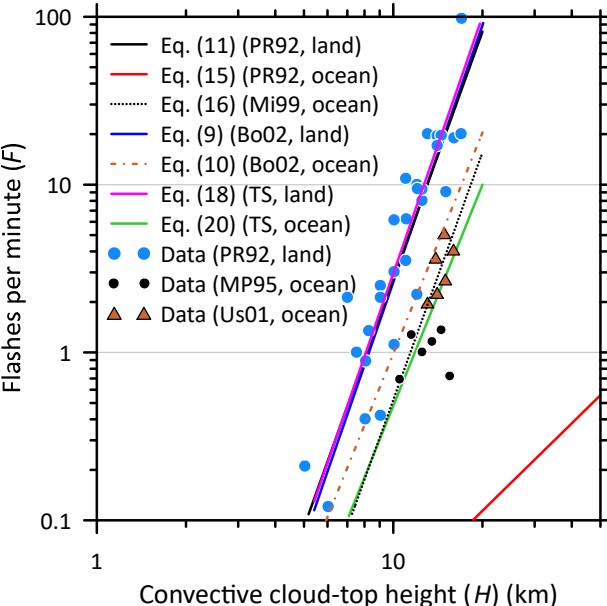

**Figure 1: Various relationships, including those from this study (TS), between lightning frequency *F* (flashes per minute) and convective cloud-top height *H* (km). The land data points are from PR92, and the ocean data are from Molinié and Pontikis (1995) (MP95) and Ushio et al. (2001) (Us01); Bo02 – Boccippio (2002), Mi99 – Michalon et al. (1999), and PR92 – Price and Rind (1992).**

### 3.3 Michalon et al.'s (1999) ocean parameterisation

Michalon et al. (1999) identified that the PR92 parameterisation, when used over water in global models, was producing lightning flash frequencies that did not agree with observations. To address this, they proposed that cloud electrification is directly related to cloud droplet concentration ($N$) and droplet size, and derived $F = A\,N^{2/3}\,H^5$, where $A$ is a proportionality constant. They retained the continental PR92 expression (11) by considering that it has been directly calibrated by using observed $F$ and $H$ values, and used it for the ocean by multiplying it with a factor of $(N_O/N_L)^{2/3}$ assuming that cloud droplet concentrations for continental ($N_L$) and marine ($N_O$) clouds are different:

$$F_O = 3.44 \times 10^{-5} \left(\frac{N_O}{N_L}\right)^{2/3} H^{4.9} = 6.57 \times 10^{-6} H^{4.9}, \tag{16}$$

where two "standard" continental and maritime cloud droplet concentrations of $N_L = 600$ and $N_O = 50$ per mg of air, respectively, were used. Eq. (16) is plotted in Figure 1. One factor for a smaller droplet concentration in marine clouds is suggested to be more intense droplet coalescence in such clouds (Rosenfeld and Lensky, 1998).

In the above approach, the values of $N_L$ and $N_O$ are not well constrained. There is a large variability in cloud droplet concentration over land and ocean, as reflected in values observed in field experiments as well as those prescribed in cloud microphysics schemes in global models (convective cloud droplet concentrations are not usually predicted in global climate models) (Rosenfeld and Lensky, 1998; Gultepe and Isaac, 2004). Boccippio (2002) cautions against this approach, and we suggest that given the uncertainty in the mean droplet concentrations, the approach of Michalon et al. (1999) is essentially empirical.

### 3.4 This study: Alternative flash-rate parameterisations

This study includes (a) a reanalysis of the Williams (1985) and PR92 data for lightning flashes versus cloud top height over land into the self-consistent scaling relationships framework of Boccippio (2002) and (b) the derivation of a new relationship for the oceanic environment using these scaling relationships.

For land, considering initially the relationship of updraft velocity with cloud-top height: equating the scaling relationship Eq. (6) with the observed maximum updraft velocity from PR92 given in Eq. (12) gives $k_L = 1.49$ and $a_L = 1.09$. Substituting these values into the relationship of flash rate with cloud-top height Eq. (7) yields

$$F_L = 1.49 \, k_1 H^{5.09}. \tag{17}$$

At this stage $k_1$ is undetermined. We proceed by fitting Eq. (17) directly to the $F$ vs. $H$ data for land reported by Williams (1985) and compiled by PR92 (which are reproduced in Figure 1 as blue solid circles). This gives $k_1 = 1.612 \times 10^{-5}$ and

$$F_L = 2.40 \times 10^{-5} H^{5.09}. \tag{18}$$

The relationships PR92 Eq. (11), Boccippio (2002) Eq. (9) and this study (TS) Eq. (18) are presented in Figure 1 along with the $F$ vs. $H$ data just discussed. Although these behaviours look almost identical and are well within the scatter of the data, Eq. (18) shows a slightly higher flash rate for higher $H$ than the other two. The almost fifth power dependence on $H$ makes $F_L$ very susceptible to even small changes in the $F_L$-$H$ relationship and in how cloud-top height is calculated in the model. Note that variation of flash rate at higher $H$ is important for both global distributions of flash rate and how it may change with changing convective activity, e.g. climate change.

Concerning the oceanic parameterisation: there are limited data on flash rate versus cloud-top height for marine environments. Figure 1 presents Ushio et al.'s (2001) (Us01) data over the ocean in the tropics and extratropics (their Figure 3b and 3d), which they obtained by averaging flash rates every 1 km in cloud height. These are shown as triangles in Figure 1. Also shown are the Molinié and Pontikis' (1995) (MP95) flash-rate data over French Guyana coastal zone which we averaged over every 1 km in cloud height for heights greater than 10 km (below this height, the number of data points is not sufficient for averaging). Because these are coastal observations, it is possible that the air masses in which the thunderstorm clouds developed could be mixed air masses (i.e. both continental and marine). However, these data do show a qualitative agreement with the Ushio et al. (2001) data in Figure 1.

Applying the scaling relationships (Boccippio 2002) to obtain an equivalent relationship for marine clouds involves equating Eq. (6) with the observed oceanic convective updraft velocity versus cloud top height from PR92 given in Eq. (13). This yields $k_O = 2.86$ and $a_O = 0.38$. Substituting these into the scaling relationship Eq. (7) gives

$$F_O = k_1 \times 2.86 \times H^{4.38} = A_O H^{4.38}. \qquad (19)$$

For this study, fitting Eq. (19) to the ocean data leads to $A_O = 2 \times 10^{-5}$ (which gives $k_1 = 0.7 \times 10^{-5}$) and

$$F_O = 2.0 \times 10^{-5} H^{4.38}. \qquad (20)$$

This marine parameterisation yields flash rates that are approximately an order of magnitude smaller than the PR92 continental formula and roughly two orders of magnitude larger than the PR92 marine parameterisation. In Figure 1 are shown the relationships for lightning flash rates and associated cloud-top height over the ocean for PR92, Boccippio's (2002) Eq. (10), Michalon et al. (1999) and this study (TS) Eq. (20). Clearly the PR92 ocean equation is unrealistic, and the relationship of Boccippio (2002) gives marine flash rates that are twice as large as Eq. (20) and are not supported by the data plotted. The relationships of Michalon et al (1999) and this study group together around the oceanic data.

The values of $k_1$ implicit in Eqs. (18) and (20) are different for land and ocean, although as per Boccippio's (2002) theory, they should be the same (as used in deriving Eqs. (9) and (10)). This implies that either or both assumptions in the theory that each flash is responsible for a constant electrostatic energy and that the variability in $\rho$ is small are not true, possibly due to differences in cloud microphysics between land and ocean. If we use the same logic concerning aerosol microphysics as Michalon et al. (1999) used in deriving Eq. (16), then the different values of $k_1$ in Eqs. (18) and (20) for land and ocean can be interpreted in terms of $N_O$ and $N_L$ being different with $N_O < N_L$.

## 3.5 Global model runs with various flash-rate parameterisations

In order to assess the above flash-rate parameterisations against global lightning flash observations, and to investigate how they influence tropospheric composition via their impact on LNO$_x$ generation, we conduct the following five runs of the ACCESS-UKCA global chemistry-climate model incorporating seven specified flash-rate parameterisations:

- Run 1 (PR92): The default PR92 parameterisations: continental Eq. (11) and marine Eq. (15),
- Run 2 (this study - TS1): The new parameterisations: continental Eq. (18) and marine Eq. (20),
- Run 3 (this study - TS2): The PR92 continental Eq. (11) and new marine Eq. (20), and
- Run 4 (Mi99): The PR92 continental Eq. (11) and Michalon et al.'s (1999) marine Eq. (16).
- Run 5 (Bo02): Boccippio's (2002) formulae Eqs. (9) and (10).

Note that Runs 2 and 5 only differ in the values of the linear coefficients used in the flash-rate parameterisations. We also did a model run with no $LNO_x$ emissions for a broad comparison, where appropriate, with the modelled changes in tropospheric composition resulting from the changes in the flash-rate parameterisations.

ACCESS-UKCA was setup as a free running simulation for 2 years (2005–2006) for each of the above runs, and the simulation was started using model initial conditions taken from a previously spun-up, nudged model run that used a Newtonian relaxation nudging (Uhe and Thatcher, 2015) within model levels 20–45 (between altitudes ~ 3 km to 14 km) and the default lightning scheme. The variables nudged were the horizontal wind components and potential temperature by using ECMWF's ERA-Interim reanalyses (Dee et al., 2011) on pressure levels. The idea was to start the simulation with meteorological/transport errors minimised in the free troposphere to the extent possible. The first year of the free running simulation was used as a spin-up period and the model output for the year 2006 used for analysis reported below. (We also did Runs 1 and 2 with nudging for the years 2005–2006 with the same initial conditions as for the free running simulations, and the results are summarised in Section 5).

### 3.6 Comparison of the modelled lightning flash rates with satellite observations

We analyse the global gridded lightning flash data from the Optical Transient Detector (OTD) on the OrbView-1 (formerly MicroLab-1) satellite and the Lightning Imaging Sensor (LIS) on the Tropical Rainfall Measuring Mission (TRMM) satellite, which are described by Cecil et al. (2014) and available from https://lightning.nsstc.nasa.gov/data/data_lis-otd-climatology.html (V2.3.2015). The high resolution (0.5° × 0.5°) mean annual flash climatology (HRFC), low resolution (2.5° × 2.5°) mean annual flash climatology (LRFC), and low resolution (2.5° × 2.5°) monthly time series (LRMTS) data products are useful for our analysis. The climatology and time series data are flash density values with the units of flashes $km^{-2}$ $yr^{-1}$ and flashes $km^{-2}$ $day^{-1}$, respectively. The OTD data available from July 1995 to January 2000 cover all latitudes, whereas the LIS data available from February 2000 to February 2014 cover ± 42.5°. The climatology data cover all latitudes. For comparison with the modelled flash parameters, the satellite data were spatially regridded to the model N96 resolution (1.875° longitude × 1.25° latitude) using the Climate Data Operators (CDO) software.

Table 1 gives the observed and modelled lightning flash frequencies (flashes $s^{-1}$) averaged over the globe, Northern Hemisphere (NH), Southern Hemisphere (SH), land, and ocean for the five model runs, and these are plotted in Figure 2. In Figure 2a, the observations are the combined LIS and OTD climatological data, whereas in Figure 2b, the observations are

the LIS data for the year 2006 which are only available for the latitudinal range ± 42.5º with the modelled values also given for this range. The observations for the year 2006 are very well correlated to the climatology, and the two are very similar in magnitude as well since most lightning activity would fall within ± 42.5º. The data show that ~ 80% of the global lightning flashes are over land, and ~ 55% are in the Northern Hemisphere. On average, the default PR92 parameterisations (Run 1)

underestimate the flash frequency data by 28% for the globe, by ~ 13% for land, and by ~ 96% for the ocean. Clearly, the oceanic PR92 flash-rate parameterisation Eq. (15) does not work well at all over the ocean, yielding an almost zero flash frequency compared to the data. In contrast, the new flash-rate parameterisations (Run 2, Eqs. (18) and (20)) greatly improve the estimation of flash frequency over the ocean, with some overestimation (by ~ 15%) of the climatology and giving nearly the same value as the year 2006 observational data. There is an improvement over land too compared to the PR92 formula.

Globally, the new parameterisations yield almost the same flash frequency as the data. Both the PR92 and the new parameterisations lead to an almost equal partitioning of flashes in the Northern and Southern hemispheres, compared to ~ 55% in the Northern Hemisphere indicated by the data. As expected, the Run 3 flash rate is nearly the same as that for Run 1 for land, and that for Run 2 for the ocean. For Run 4, with the PR92 continental Eq. (11) and Michalon et al.'s (1999) marine Eq. (16), the flash rate over land is almost the same as Run 1, whereas for the ocean Run 4 gives a flash rate that is about

50% higher than the climatology and 23% higher than the data for 2006. Over the ocean, Run 5 (Bo02) predicts flash rate nearly twice as large as that observed and that from Run 2, which leads to an overestimation of the observed global total. Over land the Run 5 estimate is about 10% lower than Run 2.

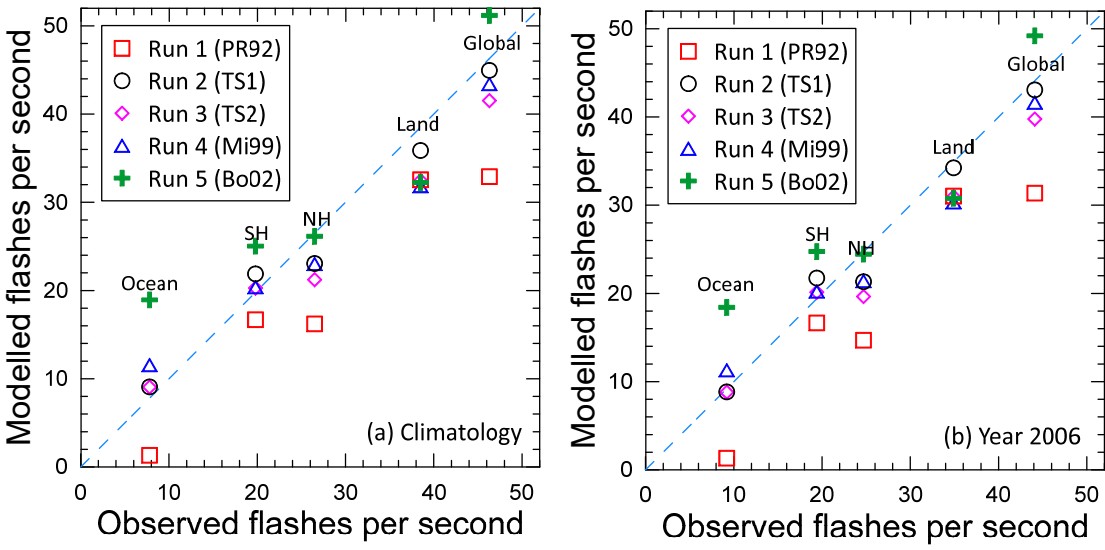

**Figure 2: Observed versus modelled flash rates from the five model runs (corresponding to PR92, this study 1 and 2, Mi99, and Bo02) conducted for the year 2006. The vertical clusters of points are the values over the globe, ocean, land, Southern Hemisphere (SH), Northern Hemisphere (NH). (a) The observations are the LIS/OTD climatological data, and (b) the observations are the LIS data for the year 2006 that are available for the latitude range ± 42.5º (with the modelled values also corresponding to this range).**

**Table 1: Observed and simulated lightning flash frequencies for the various model runs. The values in the parentheses are for the latitudinal range ± 42.5º (which correspond the latitudinal coverage of the LIS data).**

| Data/model | Lightning flash frequency (flashes s$^{-1}$) | | | | |
| --- | --- | --- | --- | --- | --- |
| | Global | NH | SH | Land | Ocean |
| LIS/OTD climatology | 46.26 (43.55) | 26.48 (23.86) | 19.78 (19.69) | 38.54 (36.00) | 7.72 (7.55) |
| LIS observations – year 2006 | (44.08) | (24.73) | (19.35) | (34.92) | (9.16) |
| Run 1 (PR92) | 32.92 (31.36) | 16.22 (14.70) | 16.70 (16.66) | 32.56 (31.03) | 0.36 (0.33) |
| Run 2 (TS1) | 44.96 (43.07) | 23.07 (21.33) | 21.89 (21.74) | 35.88 (34.23) | 9.08 (8.84) |
| Run 3 (TS2) | 41.53 (39.77) | 21.24 (19.64) | 20.29 (20.13) | 32.47 (30.96) | 9.06 (8.81) |
| Run 4 (Mi99) | 43.42 (41.65) | 23.03 (21.42) | 20.39 (20.23) | 31.85 (30.34) | 11.57 (11.31) |
| Run 5 (Bo02) | 51.19 (49.20) | 26.15 (24.45) | 25.04 (24.75) | 32.25 (30.78) | 18.94 (18.42) |

The monthly variation of the observed and modelled flash rates is shown in Figure 3. The observed variation for the year 2006 is very similar to the climatological variation, mostly within the one standard deviation climatological variability shown. The model runs underpredict in spring in both hemispheres and overpredict in autumn in the Southern Hemisphere. In the Northern Hemisphere (Figure 3a), the model simulates the observed variation qualitatively with a peak in July, but while Run 1 underestimates the observed flash rate for all months, the other Runs mostly underestimate during February – May and do well for the other months. In the Southern Hemisphere (Figure 3b), again the model is able to simulate the observed variation well qualitatively, but a significant overprediction for January – April and an underprediction for August – October is apparent. The underprediction in spring in the Northern Hemisphere and overprediction in autumn in the Southern Hemisphere could be due to a displacement of lightning activity across the equator. The underprediction in spring

in the Southern Hemisphere appears to be due to model deficiency over land (Figure 3d). In the global plot (Figure 3c), while Run 1 always underestimates, which is mostly because of its underprediction of the flash rate over the ocean (Figure 3e), the other Runs underestimate in the first three months of the year and overestimates during September – November, and these differences can be explained in terms of the hemispheric differences shown above. The nature of monthly variation in Figure 3d for land-based flash rates is very similar to Figure 3c, indicating that continental flashes dominate the global total. The large underprediction by the PR92 scheme in Run 1 and an overprediction in Run 4 over the ocean can be seen in Figure 3e; there are also differences in the monthly variation. Run 5 overestimates the observed oceanic variation the most out of all Runs, and this leads to an overprediction of the observed global variation except for September and October.

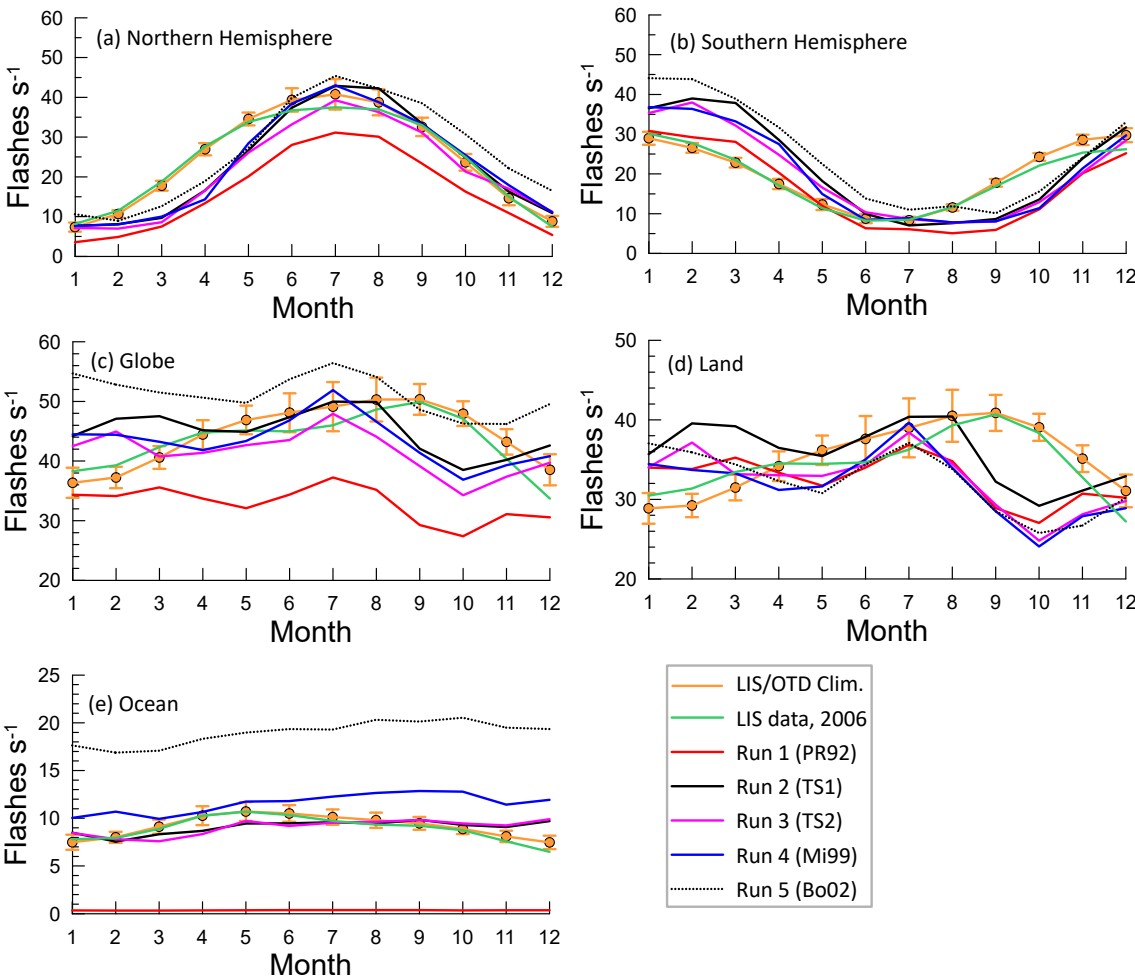

**Figure 3: Monthly variation of the observed and modelled flash rates from the five model runs for the year 2006: (a) Northern Hemisphere (NH), (b) Southern Hemisphere (SH), (b) ocean, (c) land, (d) globe Southern Hemisphere (SH), and (e) The observations are the LIS/OTD climatological data (with 1 standard deviation variability) and the LIS data for the year 2006.**

The normalised mean square error NMSE $= \overline{(M - O)^2}/(\bar{M}.\bar{O})$ calculated from the monthly-varying observed climatology ($O$) and modelled ($M$) flash rate timeseries shown in Figure 3 are given in Table 2 for the globe, land and ocean. Also given are the values of fractional bias FB $= 2(\bar{O} - \bar{M})/(\bar{O} + \bar{M})$, which varies between -2 (overestimation) and +2 (underestimation), and the (Pearson) correlation coefficient ($r$). Considering these performance statistics together suggests that the Run 2 flash-rate parameterisations from this study yield the best comparison with the data.

**Table 2: Normalised mean square error (NMSE), fractional bias (FB), and correlation coefficient ($r$) for the monthly-varying observed climatology and modelled flash rates shown in Figure 3.**

| Flash-rate scheme | NMSE | | | FB | | | Temporal correlation ($r$) | | |
|---|---|---|---|---|---|---|---|---|---|
| | Globe | Land | Ocean | Globe | Land | Ocean | Globe | Land | Ocean |
| Run 1 (PR92) | 0.164 | 0.079 | 18.939 | 0.343 | 0.185 | 1.813 | 0.12 | 0.08 | 0.35 |
| Run 2 (TS1) | 0.029 | 0.048 | 0.064 | 0.034 | 0.088 | -0.213 | 0.32 | 0.16 | 0.33 |
| Run 3 (TS2) | 0.046 | 0.084 | 0.068 | 0.114 | 0.188 | -0.211 | 0.22 | 0.06 | 0.18 |
| Run 4 (Mi99) | 0.031 | 0.088 | 0.233 | 0.069 | 0.207 | -0.449 | 0.41 | 0.19 | 0.22 |
| Run 5 (Bo02) | 0.036 | 0.094 | 0.987 | -0.096 | 0.195 | -0.884 | 0.18 | -0.08 | 0.23 |

Figure 4 presents the observed and modelled zonal mean flash density (flashes km$^{-2}$ yr$^{-1}$) over the globe, land, and ocean. All modelled global distributions and the data agree that the flash density is largely concentrated in the tropics. The observed peak for the year 2006 in Figure 4a is better simulated by Runs 2–4 than by Run 1 (underestimation) or Run 5 (overestimation). The results over land (Figure 4b) are very similar for all Runs. Over the ocean (Figure 4c), while the default oceanic parameterisation (Run 1) yields a near-zero flash density distribution and Run 5 overestimates considerably, the new flash parameterisation (in Runs 2 and 3) performs much better. There are significant distributional differences compared to the data. It is clear in these plots that the observed latitudinal distributions of flash density are wider than the modelled ones, with larger observed flash densities in the subtropics stretching into the mid latitudes (roughly 20–40° in both hemispheres) than modelled. The reason for this may be the inherent limitation of the simple flash parameterisation approach based on convective cloud-top height or uncertainty/biases in the modelled convection (e.g. Allen and Pickering, 2002; Tost et al., 2007). Another potential factor could be greater vertical wind shear outside the tropics which extends the horizontal

lightning channel length (Huntrieser et al., 2008), which is not accounted for in the cloud-top height-based approaches. The LIS/OTD observations have some limitations too, such as a short sampling duration (just minutes) for a particular global location and lightning detection efficiencies not being perfect (Clark et al., 2017).

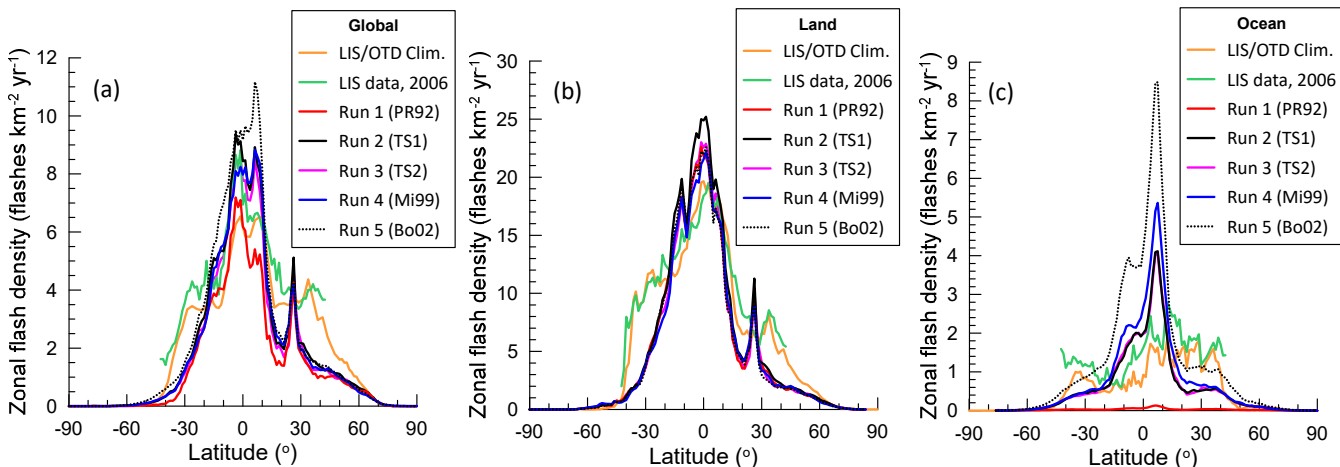

**Figure 4: Observed and modelled zonal mean flash densities (flashes km⁻² yr⁻¹) from the five model runs over the (a) globe, (b) land and (c) ocean. The observations are the LIS/OTD climatology and the LIS data for the year 2006.**

Hereafter, we only present plots from Run 1 (default) and Run 2 (new), but the results from Runs 3–5 are included in all comparison Tables except Table 5.

Figure 5 compares the various global distributions of the mean annual lightning flash density at N96 resolution. The LIS data for the year 2006 in Figure 5b are only available for the latitudinal range ± 42.5° and have no sampling for some regions within this range. Where there are data, there is a good agreement between the observed distribution in Figure 5b and the LIS/OTD climatology in Figure 5b, showing high flash density over land in the tropics and subtropics, and also lower mid latitudes. There is also some significant flash density over the ocean at these latitudes, particularly over the Pacific, western Atlantic, western Indian Ocean near southern Africa, and the seas around the maritime continent (i.e., largely Indonesia, the Philippines and Papua New Guinea). The distribution modelled using ACCESS-UKCA with the default PR92 flash-rate scheme (Run 1) shown in Figure 5c is very similar to other global modelling studies that use the same PR92 scheme, e.g. Allen and Pickering (2002) using GEOS-STRAT DAS, Tost et al. (2007) using ECHAM5/MESSy, Murray et al. (2012) using GEOS-Chem, Finney et al. (2014) using ERA-Interim reanalyses, Finney et al. (2016) using UM-UKCA vn8.4, and Clark et al. (2017) using CAM5. It is remarkable that the simple PR92 scheme based on the convective cloud-top height is able to simulate the broad observed global distribution of flash density over land at low latitudes (except parts of India), but does not properly reproduce the extension of lightning flash density into the temperate latitudes, particularly in the Northern

Hemisphere. Over the ocean, in contrast to the observations, the PR92 scheme predicts almost zero flash density. However, as shown in Figure 5d, ACCESS-UKCA with the new flash-rate parameterisations (Run 2) simulates the oceanic distribution of flash density much better than the PR92 scheme, although it is clear that there are some significant spatial differences (e.g. low bias over western Indian Ocean near southern Africa, and high bias over equatorial Indian Ocean and the Pacific) compared to the corresponding observations and climatology. The modelled flash-density distributions over land in Figure 5c and Figure 5d are nearly the same.

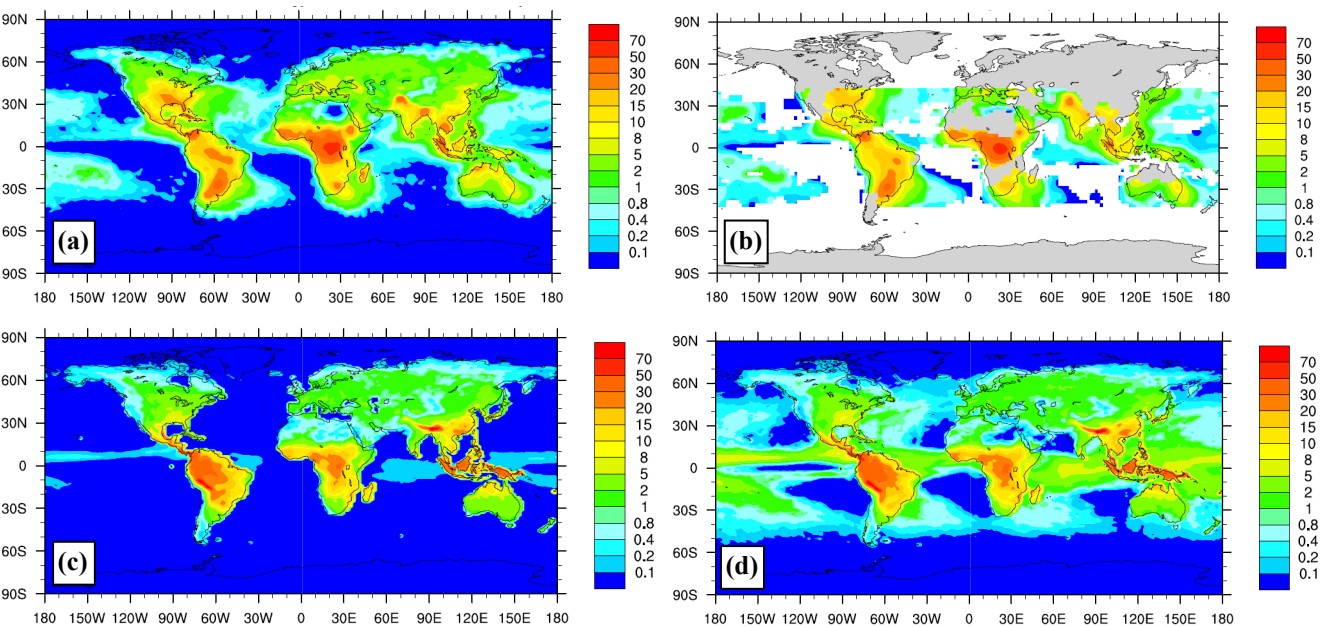

Figure 5: Global distribution of the mean annual lightning flash density (flashes km⁻² yr⁻¹): (a) LIS/OTD satellite climatology, (b) LIS satellite data for the year 2006 (available only for ± 42.5° latitudes), (c) model simulation with the default PR92 flash-rate parameterisations (Run 1), and (d) model simulation with the new flash-rate parameterisations from this study (Run 2).

The area-weighted NMSE, FB and correlation coefficient (*r*) comparing the spatial patterns of the observed climatology presented in Figure 5a with the annually-averaged modelled patterns of flash rate are given in Table 3 for the five model runs. The NMSE and FB values clearly show that Run 1 performs the worst, a result dominated by the oceanic component. While the NMSE values are nearly the same for Runs 2–4, Run 2 has the best FB values for both land and ocean. While Run 5 has the best FB value for the globe, this is fortuitous because the model underestimation and overestimation for land and ocean, respectively, counteract in the calculation of global FB. Thus, these statistics should be considered separately for land and ocean in examining the model performance. The spatial pattern correlation stays essentially the same for all Runs (presumably because the underlying independent model variable, the cloud-top height, is the same in all model runs), and it

is lower for the ocean – suggesting that further understanding of convection and lightning processes and their parameterisations are needed. The global correlations in Table 3 are very similar to those reported by Gordillo-Vazquez et al. (2019) for cloud-height based schemes, but those for the ocean are lower in our study.

Based on the above flash-rate comparisons, Run 2 (TS1) performs the best, followed by Run 3 (TS2), Run 4 (Mi99) and Run 1 (PR92).

Modelled flash rates depend critically on modelled convection parameters (e.g. the cloud-top height) used by flash-rate parameterisations and on the representativeness of these parameterisations themselves. Thus, it is common in a global model to match the globally averaged modelled lightning flash rate to the observed value, e.g. that based on the LIS/OTD climatology (~ 46 flashes s$^{-1}$), by applying a constant scaling factor to the modelled global flash-rate spatial distribution (e.g., Tost et al., 2007; Finney et al., 2016; Clark et al., 2017; Gordillo-Vázquez et al., 2019), where the scaling factor is the ratio of the observed global average flash rate to the modelled global average flash rate and is calculated by doing a model pre-run. However, such a scaling would be misleading when there are large differences in the spatial representativeness of the flash rate computed by the parameterisation used in a model. For example, scaling the PR92 derived global flash-rate distribution would over-adjust the flash rate (and hence LNO$_x$) over land to compensate for the deficiency in the oceanic parameterisation. (Scaling can also be also applied to tune the amount of NO produced per flash to get a desired total global LNO$_x$ amount, as per Eq. (1).). In the present study, no scaling factor was applied to the modelled flash rate, nor was it necessary.

**Table 3: Normalised mean square error (NMSE), fractional bias (FB) and correlation ($r$) for the spatially-varying annual mean modelled flash rates and observed climatology shown in Figure 5a.**

| Flash-rate scheme | NMSE | | | FB | | | Spatial correlation ($r$) | | |
|---|---|---|---|---|---|---|---|---|---|
| | Globe | Land | Ocean | Globe | Land | Ocean | Globe | Land | Ocean |
| Run 1 (PR92) | 4.52 | 1.63 | 164.92 | 0.44 | 0.27 | 1.84 | 0.72 | 0.74 | 0.41 |
| Run 2 (TS1) | 3.66 | 1.66 | 6.77 | 0.13 | 0.18 | -0.06 | 0.72 | 0.74 | 0.38 |
| Run 3 (TS2) | 3.63 | 1.65 | 6.92 | 0.21 | 0.28 | -0.06 | 0.72 | 0.74 | 0.38 |
| Run 4 (Mi99) | 3.44 | 1.63 | 6.68 | 0.17 | 0.30 | -0.30 | 0.72 | 0.74 | 0.39 |
| Run 5 (Bo02) | 3.58 | 1.76 | 8.56 | 0.01 | 0.28 | -0.76 | 0.68 | 0.72 | 0.38 |

### 3.7 Modelled LNO$_x$ and comparison

#### 3.7.1 Global LNO$_x$

The modelled global lightning-generated NO$_x$ using the various lightning flash-rate schemes are presented in Table 4. With the new flash-rate parameterisations (Run 2), the modelled global LNO$_x$ increases to 6.6 Tg N yr$^{-1}$ from 4.8 Tg N yr$^{-1}$ for Run 1, an increase of 38%, most of which is due to the change in the oceanic flash-rate component. Of the total global LNO$_x$, about 20% is generated over the ocean in Run 2, compared to ~ 1% in Run 1. The partitioning into NH and SH is almost equal for both schemes. The Run 3 and Run 4 total LNO$_x$ emissions are similar to the Run 2 value, whereas the Run 5 value is 14% greater than that for Run 2 due to the higher value of the oceanic component. Given the same value of NO emitted per flash used for both IC and CG flashes, the partitioning of the global LNO$_x$ into NH, SH, and Land and Ocean for all Runs in Table 4 is very similar to the partitioning of flash rate for the corresponding Runs in Table 1.

**Table 4: Modelled global lightning-generated NO$_x$ using various lightning flash-rate schemes (Tg N yr$^{-1}$).**

| Flash-rate scheme | Lightning generated NO$_x$ (Tg N yr$^{-1}$) | | | | |
|---|---|---|---|---|---|
| | Global | NH | SH | Land | Ocean |
| Run 1 (PR92) | 4.84 | 2.39 | 2.45 | 4.79 | 0.05 |
| Run 2 (TS1) | 6.61 | 3.41 | 3.20 | 5.27 | 1.34 |
| Run 3 (TS2) | 6.11 | 3.14 | 2.97 | 4.77 | 1.34 |
| Run 4 (Mi99) | 6.39 | 3.40 | 2.99 | 4.69 | 1.70 |
| Run 5 (Bo02) | 7.53 | 3.86 | 3.67 | 4.74 | 2.79 |

The amount of global LNO$_x$ produced, $L_G$ (Tg N yr$^{-1}$), is a function of the global average flash rate, $f_s$ (flash s$^{-1}$), and the moles of NO produced per flash, $P_{NO}$:

$$L_G = 441.5 \times 10^{-6} \, f_s . P_{NO}, \qquad (21)$$

If the climatological average $f_s = 46.5$ flash s$^{-1}$ based on the LIS/OTD satellite data is used, then

$$L_G = 20.5 \times 10^{-3} \, P_{NO}. \qquad (22)$$

If the NO production per flash differs for IC and CG flashes then $P_{NO}$ can be taken as a weighted average over mean IC and CG flash fractions. The values in Table 4 are consistent with Eq. (21) when $P_{NO} = 330$ moles NO per flash as used in ACCESS-UKCA and the modelled $f_s$ from Table 1 are substituted. These values can be compared with a global estimate of $L_G = 5 \pm 3$ Tg N yr$^{-1}$ based on Schumann and Huntrieser's (2007) review. More recently, there have been estimates of LNO$_x$ incorporating top-down approaches, which we divide into (a) verification studies using other constraints and (b) model experiments.

Verification studies include: Using a CTM representing the LIS/OTD flash data, Martin et al. (2007) obtained an estimate of $6 \pm 2$ Tg N yr$^{-1}$ that best reproduced satellite observations of tropospheric NO$_2$, O$_3$ and HNO$_3$. Miyazaki et al. (2014) obtained $L_G = 6.3 \pm 1.4$ Tg N yr$^{-1}$ and a global average production of 310 moles NO per flash using an assimilation of multiple satellite measurements of NO$_2$, O$_3$, HNO$_3$ and CO, and the LIS/OTD flash data into a global CTM. The global values for Runs 2–4 in Table 4 compare very well with the above constrained estimates, considering their differences in NO per flash used (the ratio $L_G/P_{NO} \approx 0.02$ in these estimates as per Eq. (22)). Using upper tropospheric airborne observations of LNO$_x$ and global satellite-retrieved tropospheric NO$_2$ column densities along with GEOS-Chem, Nault et al. (2017) estimated 665 moles NO per flash with global LNO$_x$ at $\sim 9$ Tg N yr$^{-1}$. Using global satellite data of NO$_2$ columns and lightning flashes together with the GEOS-Chem model, Marais et al. (2018) derived a global production rate of $280 \pm 80$ moles NO per flash, with a global LNO$_x$ emission of $5.9 \pm 1.7$ Tg N yr$^{-1}$.

Boersma et al. (2005) analysed above-cloud tropospheric NO$_2$ column retrievals from the GOME satellite observations for the year 1997 for cloudy scenes over tropical oceans and continents, and found that the above-cloud annual-mean NO$_2$ column increases sharply with convective cloud-top height ($H$)—as $H^{5.1}$ for continents and $H^{4.6}$ for oceans, where $H > 6.5$ km. Considering that these above-cloud NO$_2$ columns primarily consist of contributions from lightning-generated NO$_x$, which is a direct function of flash rate, there is a very good agreement between these power-law exponents and those in the flash-rate relationships (11) and (18) for continents, and (20) for oceans. The analysis of Boersma et al. (2005) demonstrates that the exponent of 1.73 in the PR92 oceanic flash-rate relationship (15) is unrealistic.

Model experiments include: Using a 3-D cloud resolving model coupled with observations from a thunderstorm and assuming $P_{NO} = 360$ moles NO per flash, Ott et al. (2007) estimated a global LNO$_x$ of 7 Tg N yr$^{-1}$. Using a CTM constrained by the LIS/OTD flash together with a production of 500 moles NO per flash for all extratropical lightning north of 23°N in America and 35°N in Eurasia, and 260 moles NO per flash for the rest of the globe, Murray et al. (2012)[2] determined a global LNO$_x$ of $6 \pm 0.5$ Tg N yr$^{-1}$.

---

[2] In Murray et al. (2012), these production values are given as N per flash, but a cross-referencing suggests that these should be in NO per flash.

The modelled mean global distributions of LNO$_x$ from the two runs presented in Figure 6a and b are essentially in proportion to the flash density distributions given in Figure 5c and Figure 5d, respectively. The new flash-rate scheme (Run 2) leads to a larger and broader distribution of LNO$_x$ over the ocean compared to the PR92 scheme, while over land they are very similar.

In the absence of any direct measurements of global spatial distribution of LNO$_x$ for comparison we present in Figure 6c the annual LNO$_x$ distribution obtained by Miyazaki et al. (2014) using an assimilation of satellite measurements of atmospheric composition and the LIS/OTD lightning flash data into a global CTM for the year 2007. This plot is a reproduction of their Figure 6 (middle-left plot) based the data[3] supplied by K. Miyazaki (personal communication, 2020) at a horizontal resolution of 2.8° × 2.8°. Over the ocean, the new flash-rate scheme (Figure 6b) agrees much better with the assimilated field than does the PR92 scheme, but is clear that the oceanic LNO$_x$ distribution in the plot with assimilation is broader, more diluted in the tropics, and even extends to high latitudes which is not seen in Figure 6b nor indicated by the observed flash-rate distributions in Figure 5a and Figure 5b (this could be due to limitations of the data assimilation used). Over land, the LNO$_x$ distributions predicted by both PR92 and the new scheme are similar and broadly agree with Figure 6c at low latitudes (except parts of India), but do properly not describe the extension of LNO$_x$ into the temperate latitudes, particularly in the Northern Hemisphere. Figure 6c yields a total LNO$_x$ of 6.36, 3.67, 2.69, 5.58 and 0.78 Tg N yr$^{-1}$ for the globe, NH, SH, land and ocean, respectively, which except for SH are closer to the Run 2 values than to the Run 1 values in Table 4. Direct and more extensive measurements would be necessary for a better evaluation of the predicted LNO$_x$ distribution.

---

[3] The units in Miyazaki et al.'s (2014) plot are incorrect – they should be $10^{-13}$ kg N m$^{-2}$ s$^{-1}$ instead of $10^{-12}$ kg N m$^{-2}$ s$^{-1}$ (K. Miyazaki, personal communication, 2020). The reproduced Figure 6c has the correct units.

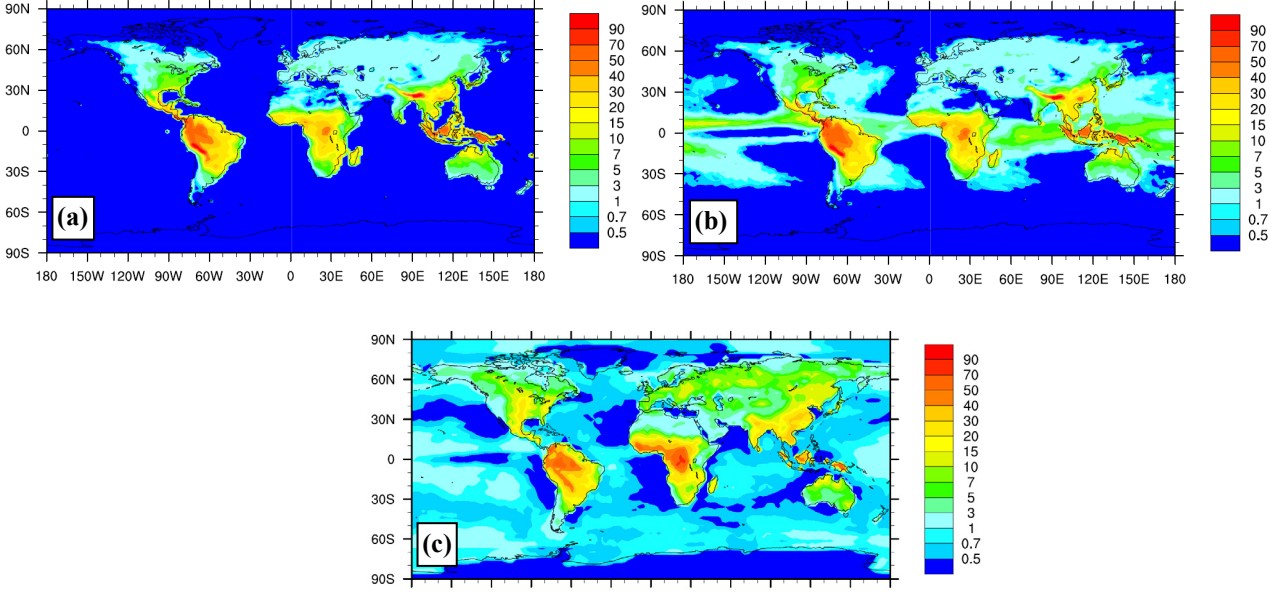

**Figure 6: Global distribution of the annual-averaged LNO$_x$ (10$^{-13}$ kg N m$^{-2}$ s$^{-1}$) for the year 2006: (a) model simulation with the default PR92 flash-rate parameterisations (Run 1), (b) model simulation with the new flash-rate parameterisations from this study (c) the distribution obtained by Miyazaki et al. (2014, plot redrawn) using assimilation of multiple satellite datasets into a global CTM for the year 2007. The respective global LNO$_x$ totals are 4.8, 6.6 and 6.3 Tg N yr$^{-1}$.**

### 3.7.2 Vertical distribution

The vertical distribution of LNO$_X$ in the model at a grid point location is a parameter (see Section 2.2) that is essentially unconstrained by observations. Figure 7 presents the modelled average vertical distribution of percentage of LNO$_x$ mass in each 1-km layer for (a) tropical continental, (b) tropical marine, (c) midlatitude continental, and (d) subtropics regimes. The non-uniform shape of the averaged modelled vertical distributions is largely caused by the averaging of the LNO$_x$ profile from every time step over spatial and temporal variations in the cloud-top height. Also shown for comparison are the average profiles based on thunderstorm cases simulated by Pickering et al. (1998) using a 2-D convective cloud-resolving tracer transport model and those by Ott et al. (2010) using a 3-D convective cloud-resolving chemical transport model, with both studies using parameterised lightning. The profiles of Pickering et al. (1998) show peaks near the surface, as significant mass is transported to the boundary layer by downdrafts, and in the upper troposphere (the so-called 'C-shaped' profile), whereas those by Ott et al. (2010) show very little LNO$_x$ mass in the boundary layer with the majority of LNO$_x$ remaining in the middle and upper troposphere (the so-called 'backward C-shaped' profile) where it is originally produced. Our model profiles match better with the Ott et al. (2010) profiles, but it gives an almost uniform distribution of LNO$_x$ mass below 5 km whereas the latter decrease to almost zero. There is not a large difference between the modelled profiles for the various regimes, except that the tropical ones are almost uniformly distributed between 5 km and 12 km whereas the midlatitude and

subtropical ones show a peak at 6.5 km. There are no direct measurements to verify any of the LNO$_x$ profiles and we believe further work is needed to constrain them.

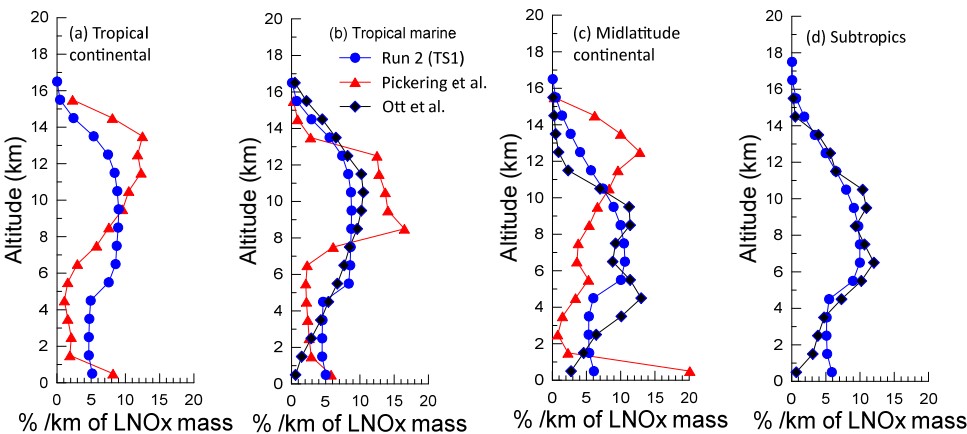

Figure 7: Average vertical distribution of percentage of LNO$_x$ mass per kilometre for (a) tropical continental, (b) tropical marine, (c) midlatitude continental, and (d) subtropics regimes. The total LNO$_x$ for these regimes calculated using the model Run 1 (PR92) is 3.69, 0.035, 1.09 and 0.74 Tg N yr$^{-1}$ respectively, whereas that calculated using Run 2 (TS1) is 4.10, 1.09, 1.16 and 0.92 Tg N yr$^{-1}$ respectively. The vertical profiles from Pickering et al. (1998) and Ott et al. (2010) are also shown (where available).

### 3.7.3 Modelled tropospheric total column NO$_2$ and validation

As lightning impacts atmospheric NO$_x$ directly, any changes in the modelled total tropospheric NO$_x$ can be examined and compared with available observations. The modelled variations of the zonal averaged tropospheric total NO$_2$ column ($N_{v,trop}$) for the globe presented in Figure 8 show a broad peak in the industrialised Northern Hemisphere dominated by surface NO$_2$ emissions. Within the latitudes ± 30°, the new lightning flash-rate parameterisations yield 12% larger values of tropospheric total NO$_2$ column compared to the default PR92 scheme and the difference between the two is ≈ 0.1 × 10$^{15}$ molecules cm$^{-2}$.

Figure 8 also presents tropospheric NO$_2$ column retrievals from the Ozone Monitoring Instrument (OMI) satellite overpasses at ~ 13:30 local time (LT) for the year 2006. These zonal averages were derived from the OMI monthly-mean tropospheric NO$_2$ columns (QA4ECV, version 1.1) given at a horizontal resolution of 0.125° × 0.125° (https://www.temis.nl/airpollution/no2.php; Boersma et al., 2018). While the model simulations qualitatively agree with the satellite observations in Figure 8, it is apparent that except for the latitudes 30°S–60°S the modelled values are generally higher than the observations. Notwithstanding any model shortcomings in predicting the global NO$_2$ distribution, there are limitations of the OMI satellite data used. Firstly, there is limited sensitivity of the OMI sensor to NO$_2$ below the cloud level, where most NO$_2$ is situated, and thus cloudy tropospheric NO$_2$ retrievals (with cloud radiance fraction > 0.5 or cloud fraction > 0.2) cannot be interpreted as valid down to the Earth's surface (Boersma et al., 2017). Secondly, given that the OMI

satellite data are representative of overpass time and thus comparing them with the mean model fields averaged over full diurnal periods introduces uncertainty.

As an alternate to the OMI data, we use data from the global reanalysis of atmospheric composition produced by the Copernicus Atmosphere Monitoring Service (CAMS) (Inness et al., 2019; https://ads.atmosphere.copernicus.eu/cdsapp#!/dataset/cams-global-reanalysis-eac4-monthly) for comparison with the monthly-averaged modelled $NO_2$. The reanalysis was produced by assimilating space observations of aerosols and reactive gases using a 4D-Var method in an ECMWF global atmospheric model with 60 pressure levels (from 1000 to 1 hPa) and a horizontal resolution of $0.75° \times 0.75°$. For $NO_2$, the model assimilated the tropospheric column retrievals from the SCIAMACHY, OMI, and GOME-2 satellite overpasses at ~ 10:00 LT, 13:30 LT and 09:30 LT, respectively (for the year 2006, only SCIAMACHY and OMI data were available for assimilation). Monthly-averaged total vertical column $NO_2$ reanalysis data (version - ECMWF Atmospheric Composition Reanalysis 4) are available and used here.

We obtain the tropospheric $NO_2$ column ($N_{v,trop}$) from the CAMS total $NO_2$ column ($N_v$) as follows:

$$N_{v,trop} = N_v - N_{v,180} + N_{v,trop,180}.$$
(23)

where $N_{v,180}$ is the CAMS total $NO_2$ column over the Pacific (180°W) and $N_{v,trop,180}$ is the tropospheric $NO_2$ column over 180°W. This is one approach used with satellite data to separate the tropospheric and stratospheric amounts (Inness et al., 2019), which assumes a longitudinal homogeneity of the stratospheric $NO_2$ column amounts and a constant and negligibly small $N_{v,trop,180}$ (Lauer et al., 2002). $N_{v,trop,180}$ is not available from the CAMS data, but this obtained from the OMI data discussed above is shown in Figure 8 (orange data points) for the year 2006. While these $N_{v,trop,180}$ values are small, they are neither constant nor negligibly small compared to the modelled or OMI $N_{v,trop}$ plotted in Figure 8 and would reflect contributions from sources such as lightning, aviation, shipping, and possibly regional transport over the ocean. The $N_{v,trop,180}$ values are also greater and possibly more uncertain than the differences between the two model simulations.

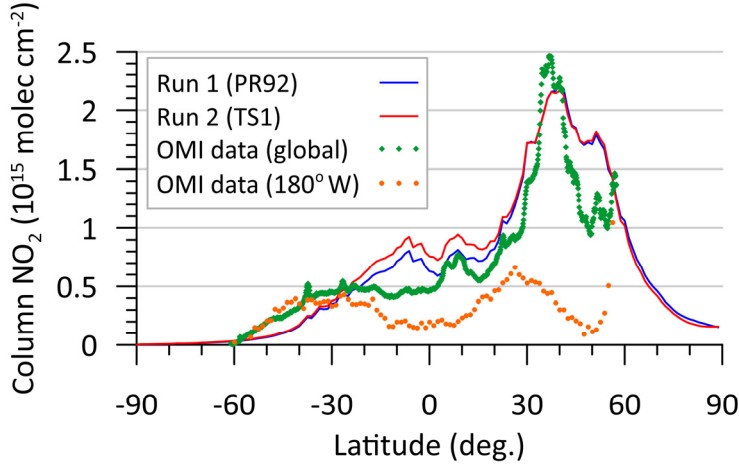

**Figure 8: Zonal averaged tropospheric total NO₂ column ($N_{v,trop}$) (in units $10^{15}$ molecules cm⁻²) obtained from Run 1 (with the default PR92 flash-rate parameterisations) and Run 2 (with the new flash-rate parameterisations from this study), and the OMI satellite data (green diamonds), for the year 2006. The orange data points are the OMI satellite data ($N_{v,trop,180}$) over the longitude 180°W.**

Since $N_{v,trop,180}$ is not available from the CAMS data, we take the OMI derived $N_{v,trop,180}$ shown in Figure 8 and use it in Eq. (23) to determine $N_{v,trop}$ from the CAMS data. (While these OMI NO₂ column data at 180°W have the same limitations as the global OMI data mentioned earlier, we expect that the impact of these limitations would be much smaller at 180°W with almost no surface source contributions compared to terrestrial locations.) The resulting CAMS zonal annual-mean $N_{v,trop}$ as a function of latitude for the globe is presented in Figure 9a, together with the corresponding modelled variations. The modelled variations agree well with the CAMS data, better than with the direct OMI data in Figure 8. A comparison with the model variation without LNOₓ in Figure 9a suggests that the modelled increase in NOₓ due to lightning is largely confined to ±35°.

In Figure 9b for the ocean, the agreement with the data is again good, except for considerable model underestimation for latitudes greater than 35° which, presumably not related to LNOₓ, may be due to factors such as possible underestimation of the calculated $N_{v,trop,180}$ and overestimation of shipping emissions of NOₓ for these latitudes. Within ±35°, the new parameterisation yields 22% larger NO₂ column values compared to the default model setup, whereas as expected the two model curves are virtually the same for the other latitudes. The CAMS NO₂ columns are somewhat better simulated by the model with the new oceanic flash rate than with the PR92 parameterisation, particularly in the northern tropics.

For land (Figure 9c), the two model curves are nearly identical and overall compare very well with the CAMS reanalysis data. An overestimation of the reanalysis data in the southern tropical region is evident.

The statistics in Table 5 show that PR92 and TS1 have similar agreement with CAMS, except for a noticeably better FB over the ocean with TS1. We expect no significant changes over land because the LNO$_x$ is almost unchanged in the two runs.

Clearly, the comparison also depends on the selected value of NO produced per flash. We have used the model default value of 330 moles NO per flash. However, if we were to match the average CAMS column value in Table 5, the new parameterisation with 310 moles NO per flash, the value suggested by Miyazaki et al. (2014), would probably yield a somewhat better prediction. Obviously, there are other sources of uncertainties in the NO$_2$ comparison, such as those associated with the CAMS reanalysis and the assimilated satellite columns, which are documented in the appropriate references cited above, as well as those to do with model inputs (e.g. emissions) and processes.

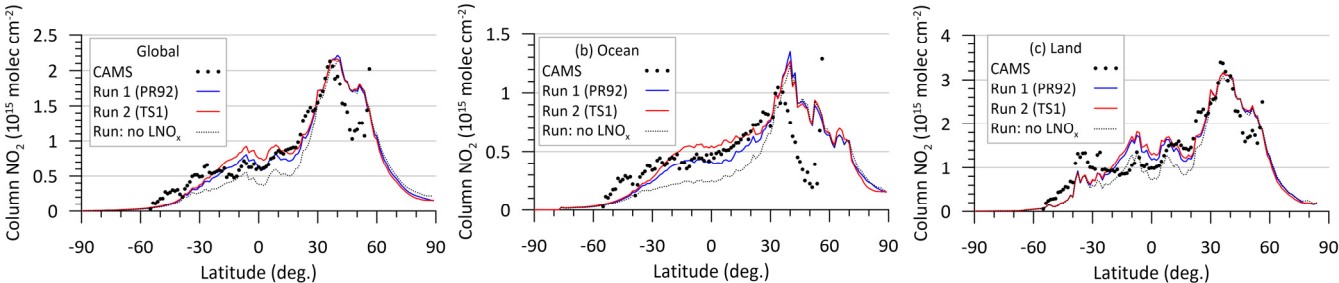

**Figure 9: Zonal averaged total tropospheric NO$_2$ column ($N_{v,trop}$) (in units $10^{15}$ molecules cm$^{-2}$) modelled using the default PR92 parameterisations (Run 1), the new lightning flash-rate parameterisations from this study (Run 2), and the CAMS reanalysis data for the year 2006 over (a) globe, (b) ocean and (c) land. The model variation without LNO$_x$ is also shown.**

**Table 5: Mean ($10^{15}$ molecules cm$^{-2}$), normalised mean square error (NMSE), fractional bias (FB) and correlation ($r$) for the modelled and CAMS reanalysis NO$_2$ columns shown in Figure 9 for latitudes between ± 30º.**

| Flash-rate scheme | Globe | | | | Ocean | | | | Land | | | |
|---|---|---|---|---|---|---|---|---|---|---|---|---|
| | Mean | NMSE | FB | $r$ | Mean | NMSE | FB | $r$ | Mean | NMSE | FB | $r$ |
| Run 1 (PR92) | 0.74 | 0.033 | 0.058 | 0.90 | 0.45 | 0.032 | 0.140 | 0.92 | 1.35 | 0.058 | -0.023 | 0.72 |
| Run 2 (TS1) | 0.83 | 0.040 | -0.059 | 0.86 | 0.55 | 0.025 | -0.061 | 0.82 | 1.42 | 0.067 | -0.073 | 0.68 |
| CAMS data | 0.78 | - | - | - | 0.52 | - | - | - | 1.32 | - | - | - |

# 4 Impact on tropospheric composition

We present the impact of $LNO_x$ determined from the flash-rate parameterisations from Runs 1 and 2 on tropospheric composition, namely total $NO_x$, $O_3$, OH and CO.

## 4.1 Oxides of nitrogen ($NO_x$)

The modelled tropospheric $NO_2$ columns and their comparison with observations have already been presented in Section 3.7.3. Figure 10 presents the zonal distribution of total $NO_x$ (as $NO_2$) from the two model simulations and the difference between the two. In the lower troposphere, the two modelled distributions of the zonal annual-mean $NO_x$ (as $NO_2$) are virtually identical, with highest levels predicted within latitudes 20–60º N. These levels are governed by surface emissions of $NO_x$ which are the same in both simulations. The secondary concentration maximum at ~15 km is due to the lightning-generated $NO_x$. The new lightning parameterisations cause an increase in the mid- to upper-tropospheric $NO_x$ (Figure 10c), particularly within the tropics and subtropics, and this increase is by as much as 40 ppt (by volume) in the northern tropics. There are some localised decreases in concentration in the lower troposphere over the Northern Hemisphere.

The volume-weighted global tropospheric $NO_x$ obtained from the PR92 scheme is 55.1 ppt, and it is 35.2 ppt over the ocean and 94.0 ppt over land. With the new flash-rate scheme, these values increase by 8.7 ppt (15.7%), 9.9 ppt (28.0%) and 6.3 ppt (6.7%), respectively, and can be compared with the values 36.9, 20.7 and 68.5 ppt, respectively, obtained from the model simulation with no $LNO_x$ emissions. To some extent, the modelled tropospheric averages also depend on how the tropopause is defined.

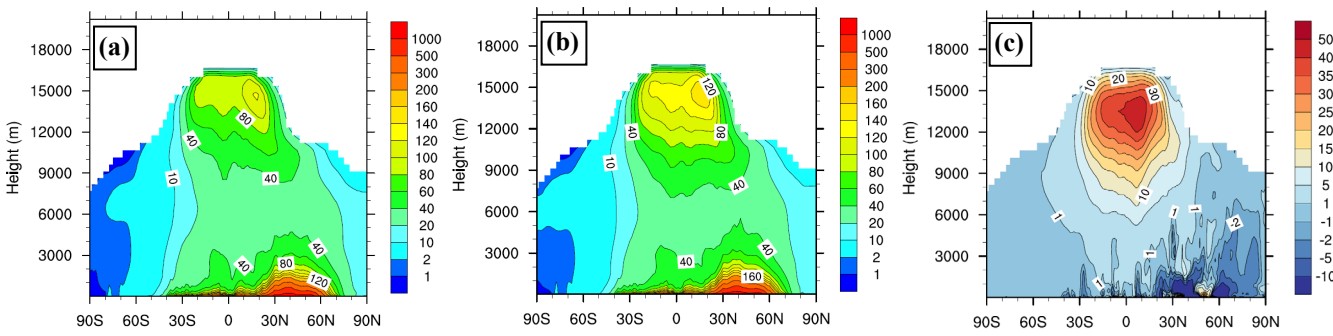

**Figure 10: Zonal annual-mean total tropospheric $NO_x$ (as $NO_2$, ppt, by volume) modelled using (a) the default PR92 parameterisations (Run 1) and (b) the new lightning flash-rate parameterisations from this study (Run 2). The difference between Run 2 and Run 1 is shown in (c).**

## 4.2 Ozone ($O_3$)

Tropospheric ozone is a by-product of the oxidation of carbon monoxide (CO), $CH_4$ and other volatile organic compounds in the presence of $NO_x$ and is thus impacted by $LNO_x$.

With the new flash-rate parameterisations (Run 2), the modelled tropospheric $O_3$ burden increases from 284 to 308 Tg $O_3$, a rise of 8.5% over the default PR92 scheme (Run 1) (cf. 219 Tg $O_3$ with no $LNO_x$ emissions in the model). The new burden is closer to a the ACCMIP (Atmospheric Chemistry and Climate Model Intercomparison Project) multi-model mean of 337 $\pm$ 23 Tg $O_3$ reported by Young et al. (2013), the latter value is consistent with measurement climatologies (this, however, does not necessarily mean that $LNO_x$ in these models is represented correctly). The Run 3 and Run 4 ozone burdens are 306 and 308 Tg, respectively.

The mean relative difference (%) between the global ozone mixing ratios predicted using the new lightning flash-rate parameterisations and the default PR92 parameterisations is shown in Figure 11. Near the surface (Figure 11a), there are significant increases in ozone over the tropical oceans, especially in the Pacific and western Indian Ocean, and in most of the Southern Hemisphere (roughly by 8% on average). Over land, there are regions (e.g. south-eastern U.S.A. and northern Australia) where ozone has increased, and there are a few regions in the mid to high latitudes in the Northern Hemisphere where ozone has decreased very slightly. Tropospheric ozone chemistry is complex, but broadly speaking the $O_3$ increases in the Southern Hemisphere are influenced by low ambient $NO_x$ concentrations where the $O_3$ production increases with NO concentration. $O_3$ is produced through photodissociation of $NO_2$ which is produced through oxidation of NO by $HO_2$ and $RO_2$ radicals (e.g., $NO + HO_2 \rightarrow NO_2 + OH$). In the Northern Hemisphere, the increase in $O_3$ is less beyond the tropic, partly because the smaller oceanic area results in a smaller increase in $LNO_x$ through the use of the new oceanic flash-rate parameterisation. Carpenter et al. (1997) suggest that the tropospheric production potential of the Southern Hemisphere is more responsive to the availability of NO than that of the (more polluted) Northern Hemisphere.

At an altitude of 6400 m (~ 450 hPa) (Figure 11b), there are even bigger increases in global ozone using the new flash-rate parameterisations, particularly in the tropics, by as much as 25%, and in the Southern Hemisphere. This is because most $LNO_x$ emissions occur in the middle to upper tropical troposphere where the photochemical production of ozone is most efficient.

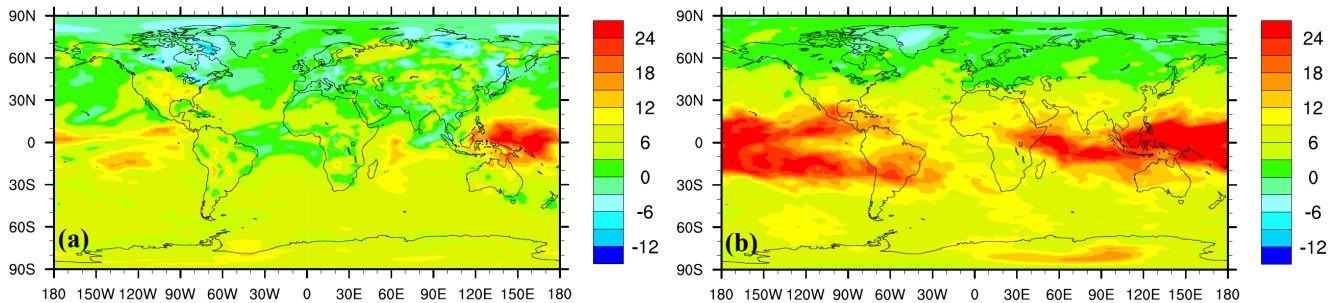

**Figure 11: Mean relative difference (%) between the global annual-mean ozone mixing ratios predicted by Run 2 using the new lightning flash-rate parameterisations from this study and the default PR92 parameterisations (Run 1): (a) at 20 m (the lowest model level) and (b) at 6400 m (~ 450 hPa).**

In Figure 12, we compare the modelled monthly averaged ozone with ground-based in-situ observations from the World Data Centre for Reactive Gases (GAW-WDCRG, http://ebas.nilu.no; https://www.gaw-wdcrg.org;) for the year 2006 at five stations: Ushuaia (54.85ºS, 68.31ºW), Cape Grim (40.68ºS, 144.69ºE), Mauna Loa (19.54ºN, 155.58ºW), Minamitorishima (24.29ºN, 153.98ºE), and Mace Head (53.33ºN, 9.90ºW). Apart from data availability and covering a range of latitudes, the

site selection was based on these sites being either oceanic or coastal so that the relatively large difference between the PR92 oceanic flash rate parameterisation and the new one could be examined against the observations. The hourly data were averaged to monthly values, and only those observational months were considered for which there were more than 75% valid hourly data points. Mauna Loa is located at an elevation of 3397 m on an island which is smaller in size than the grid resolution of the model and therefore it is difficult to correspond the sampling height to a particular vertical model level. We

used the modelled concentrations from the bottom model level for all sites. The two model simulations describe the observed monthly variations reasonably well, except at Mauna Loa and Mace Head (the relatively large disagreement at Mauna Loa is likely due to the model resolution issues). There are small, but noticeable, differences in the modelled ozone from the two simulations. The relative change in the modelled yearly averaged $O_3$ at these ground-based sites with the use of the new lightning parameterisation is small, at 5.9%, 1.3%, -1.9%, 5.9% and 0%, respectively. There is some improvement in the

modelled seasonal variation at Ushuaia, Cape Grim and Minamitorishima with the new $LNO_x$ scheme, but for the other two sites the model-data differences are much larger than those due to the $LNO_x$ changes. Generally, factors such as model's transport and chemical mechanisms, and input precursor emissions and their distributions are probably more influential in governing ozone model-data differences than $LNO_x$ near the Earth's surface. There is no clear indication if the differences in ozone between the two models are larger in the winter or summer, except for Ushuaia where the differences are larger during

winter to spring.

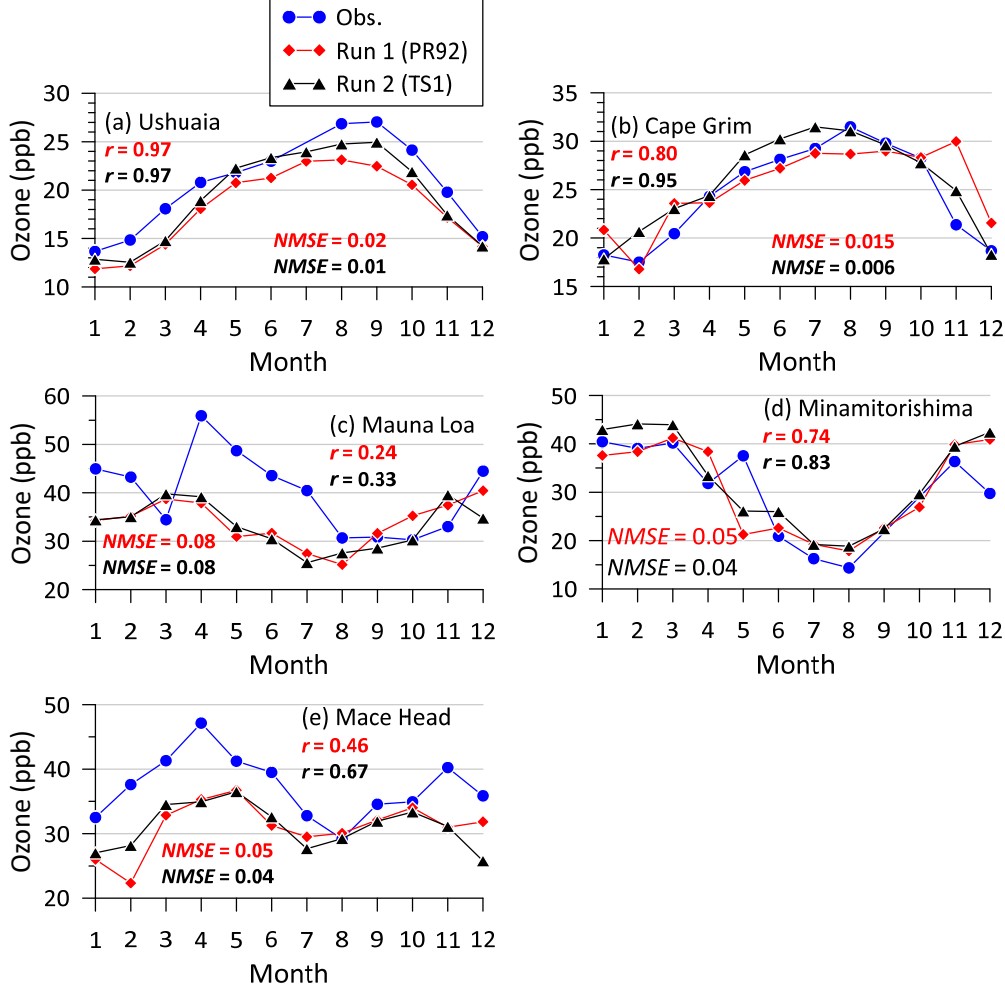

**Figure 12: Comparison of the modelled monthly-averaged ozone concentrations with observations at five oceanic/coastal ground stations for the year 2006. The two model runs are with the (a) default PR92 parameterisations (Run 1) and (b) new lightning flash-rate parameterisations (Run 2, this study). The values of correlation coefficient (*r*) and normalised mean square error (NMSE) are also shown.**

In Figure 13, we compare the modelled profiles of annual-mean ozone with those obtained from the Southern Hemisphere ADditional OZonesondes (SHADOZ) ozonesonde measurements at eight tropical/subtropical sites (Witte et al., 2017; https://tropo.gsfc.nasa.gov/shadoz) for the year 2006. These sites are: Hilo (19.40°N, 155.0°W), Paramaribo (5.81°N, 55.21°W), Costa Rica (9.94°N, 84.04°W), Natal (Brazil) (5.42°S, 35.38°E), Ascension Island (7.98°S, 14.42°W), Irene (25.9°S, 28.22°E), Nairobi (1.27°S, 36.80°E), La Reunion Island (21.1°S, 55.48°E). The above eight sites were selected based on data availability and to have a mix of northern and southern hemisphere locations where $O_3$ could be more impacted by $LNO_x$. The profile data are given at a greater resolution in height than the vertical model resolution and were

binned in the model vertical levels and averaged. The number of observed profiles at a location typically varied between 1 to 5 per month. At Irene, Nairobi and La Reunion Island, there were no data for 3-4 months. The observed profiles were averaged over the year. The modelled monthly-averaged profiles were averaged over the months for which profile observations were available. Clearly, the number of observed profiles is not sufficient for averaging over the year, and, therefore, the model-data comparison is essentially qualitative.

The observed $O_3$ profiles are simulated better by the new lightning flash-rate parameterisations (Run 2) than by the default PR92 parameterisations (Run 1) at Costa Rica (Figure 13c) and for all the southern hemispheric sites (Figure 13d–Figure 13h). At northern latitudes beyond Costa Rica, the PR92 scheme performs better at Hilo and Paramaribo (Figure 13a and b). The model with the PR92 scheme has a general tendency to overestimate ozone at most levels within 15°N–50°N, which is probably due to reasons not related to $LNO_x$, and the use of the new lightning flash-rate parameterisations worsens it (see Figure 15). The model describes the general shape of the observed profiles reasonably well. The model profiles with no $LNO_x$ included highlights the importance of $LNO_x$ on tropospheric ozone, particularly in the mid to upper troposphere.

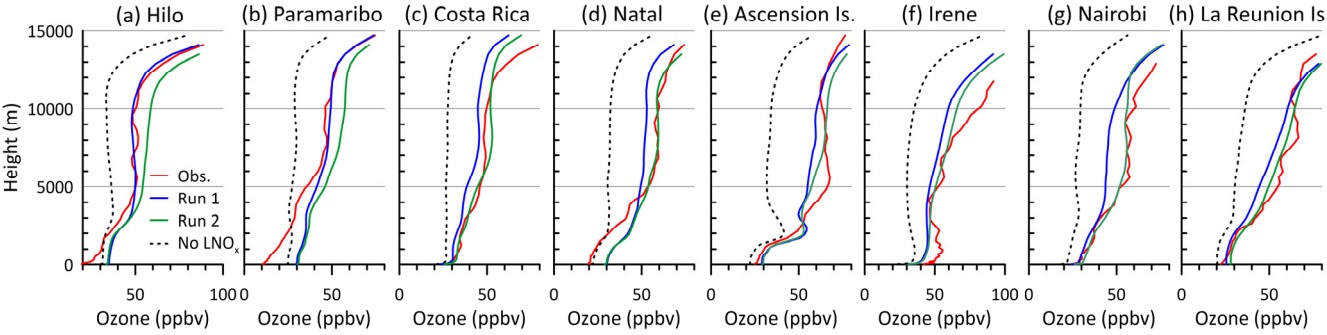

**Figure 13: Profiles of annual-mean ozone constructed using the SHADOZ ozonesonde measurements at eight sites for the year 2006, and the corresponding modelled profiles obtained using the default PR92 parameterisations (Run 1) and the new lightning flash-rate parameterisations from this study (Run 2). The modelled profiles without $LNO_x$ are also shown as a reference.**

The modelled zonal annual-mean tropospheric $O_3$ from the two Runs and the difference between the two are presented in Figure 14. In the lower troposphere, the modelled ozone is smaller over the Southern Hemisphere than over the Northern Hemisphere (Figure 14a and Figure 14b). The new flash-rate parameterisations result in $O_3$ increases everywhere (Figure 14c). Closer to the surface, the increase is approximately 2 ppb (by volume) in the Southern Hemisphere and 0.5–2 ppb in the Northern Hemisphere. The largest increases are nearly 8 ppb in the tropics at altitudes ~ 9 km.

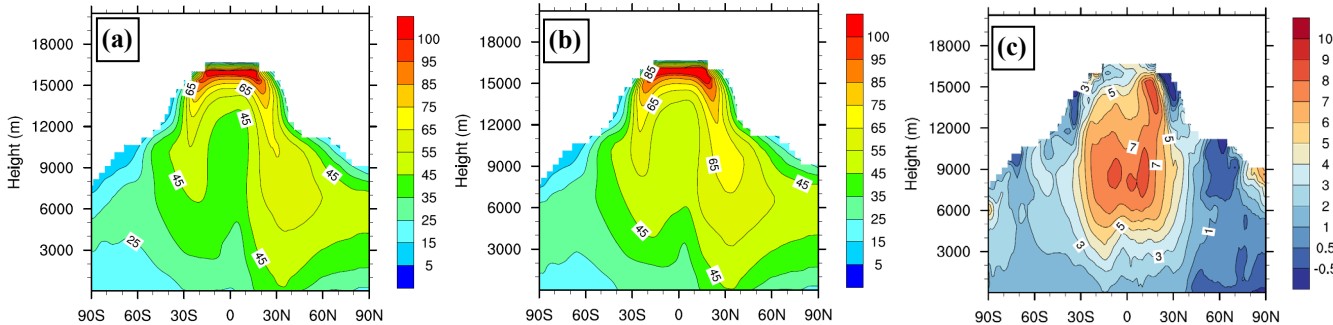

**Figure 14: Zonal annual-mean tropospheric ozone (ppb, by volume) modelled using (a) the default PR92 parameterisations (Run 1) and (b) the new lightning flash-rate parameterisations from this study (Run 2). The difference between Run 2 and Run 1 is shown in (c).**

The modelled zonal ozone distribution can be compared with observations. We use the monthly mean vertical ozone profile data for the year 2006, given as zonal means, from Bodeker Scientific database (http://www.bodekerscientific.com/data/monthly-mean-global-vertically-resolved-ozone) which combines measurements from several satellite-based instruments and ozone profile measurements from the global ozone-sonde network. The database
spans the period 1979 to 2016 with 5° latitude resolution and 70 altitude levels (1 to 70 km). Different 'Tiers' of data are provided, and we used the highest Tier 1.4 (vn1.0) data. For comparison with the model predictions, these monthly profile data were regridded to the model resolution. The modelled monthly tropospheric mask was zonal averaged and then applied to the monthly regridded data.

There is an agreement, both in magnitude and distribution, between the modelled zonal ozone in Figure 14a and b, and the
15 data plotted in Figure 15a. The model reproduces the observed lower levels of tropospheric ozone in the Southern Hemisphere. The observed high levels just below the tropopause within 10–40ºN are somewhat better reproduced by the new flash-rate parameterisations. Figure 15b and c represent the relative differences ($=[\bar{M} - \bar{O})/\bar{O}] \times 100\%$) between the annual averaged modelled and observed ozone for the PR92 scheme and the new parameterisations, respectively. On average, the model underestimation in the Southern Hemisphere has reduced with the new parameterisations, but there are areas such as
that within 10–20 ºS below 5 km where there is some overprediction. There is a clear improvement in the predicted ozone between 10ºS – 10ºN throughout the troposphere using the new scheme while that in the northern high latitudes below 6 km remains unaffected. In the Northern Hemisphere, the new scheme tends to overestimate ozone within 10–50 ºN below ~ 9 km; this is where there was already some overprediction by the PR92 scheme. In this region, even when $LNO_x$ is not included in the model (Figure 15d), the model-data differences are either small or there is some overestimation (near the
surface) of the observed ozone. This suggests that there are likely to be factors other than $LNO_x$ responsible for the predicted overestimation within 10–50 ºN. Additional factors that influence tropospheric ozone distribution in the model include dynamics, including inter-hemispheric mixing and stratosphere-to-troposphere exchange, precursor emissions, and how

chemical mechanisms are represented. Considering the above, we can say that the new flash-rate scheme leads the modelled tropospheric ozone in the right direction, which is also supported by the fact that it causes the tropospheric ozone burden to improve, as stated above. Getting ozone in the upper troposphere correct is climatically important as surface temperature is more sensitive to changes in ozone in the upper troposphere and near the tropopause than those in the lower atmosphere (Forster and Shine, 1997) and, similarly, radiative forcing due to tropospheric ozone is more sensitive to ozone abundance in the upper troposphere (Worden et al., 2011).

The volume-weighted tropospheric $O_3$ obtained from the PR92 scheme is 51.5 ppb over the globe, 48.8 ppb over the ocean and 56.6 ppb over land. With the new flash-rate scheme, these values increase by 4.1 ppb (8.0%), 4.4 ppb (9.1%) and 3.6 ppb (6.3%), respectively, and can be compared with the values 38.7, 36.2 and 43.4 ppb, respectively, obtained from the model simulation with no $LNO_x$ emissions.

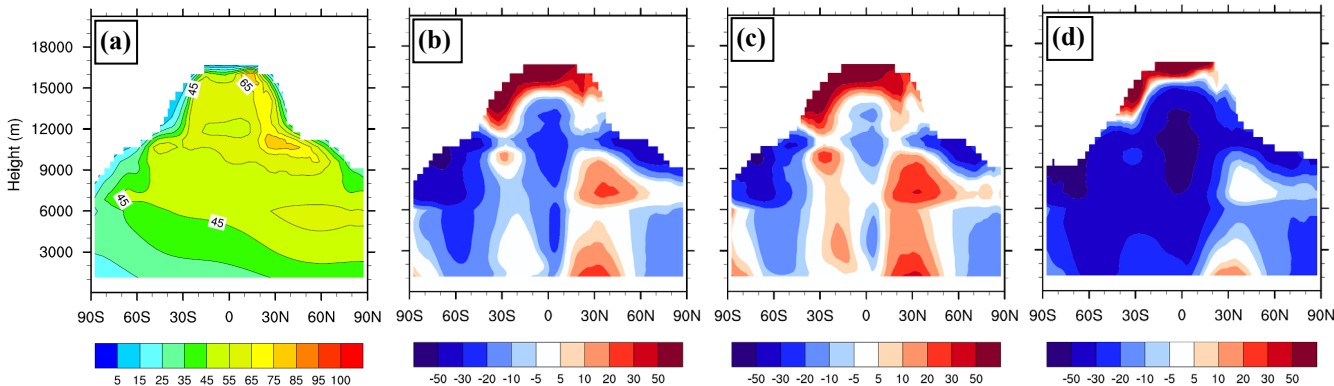

**Figure 15: Zonal distribution of tropospheric ozone concentration (ppbv) for the year 2006: (a) observed distribution based on the global monthly mean vertical ozone profile database available from Bodeker Scientific; (b) the relative difference (%) between the concentration modelled using the default PR92 lightning parameterisations (Run 1) and the observations; (c) the relative difference (%) between the concentration modelled using the new lightning parameterisations (Run 2) and the observations; and (d) the relative difference (%) between the concentration modelled without any $LNO_x$ and the observations.**

### 4.3 Hydroxyl (OH) radical

The hydroxyl radical (OH) is the dominant oxidizing (and cleansing) agent in the global troposphere and controls the atmospheric abundance and chemical lifetime of most natural and anthropogenic gases, such as methane ($CH_4$). The tropospheric abundance of OH is determined by a complex series of chemical reactions involving tropospheric ozone, methane, carbon monoxide (CO), non-methane volatile organic compounds (NMVOCs), and $NO_x$, and also the amount of solar radiation and humidity (Naik et al., 2013). Through these reactions, the amount of $LNO_x$ produced also impacts OH.

The modelled zonal total annual mean tropospheric OH in Figure 16 shows highest OH concentrations near the surface in the tropics, with values as high as $(25 - 30) \times 10^5$ molecules cm$^{-3}$ at ~ 20º N. The concentrations decrease with altitude, but there is a secondary maximum in the upper troposphere at around 13 km. There is an increase in the OH concentration using the new flash-rate parameterisations (Figure 16c), particularly in the upper troposphere in the tropics, by as much as $5 \times 10^5$ molecules cm$^{-3}$.

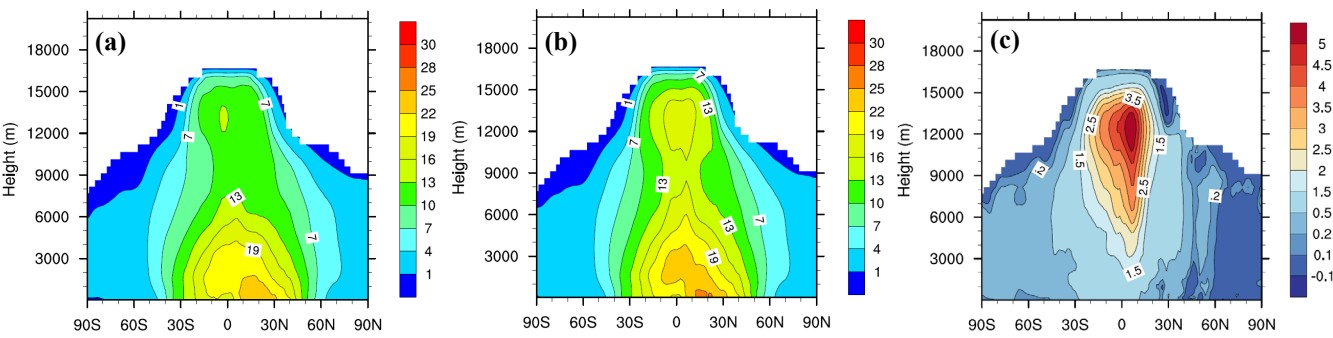

**Figure 16: Zonal annual mean tropospheric OH (× 10$^5$ molecules cm$^{-3}$) modelled using (a) the default PR92 parameterisations (Run 1) and (b) the new lightning flash-rate parameterisations from this study (Run 2). The difference between Run 2 and Run 1 is shown in (c).**

The annual mean relative OH difference (%) between Run 2 and Run 1 near the surface (Figure 17a) shows an increase in OH over the Southern Hemisphere oceans and Antarctica, and pockets of increase and slight decrease in the Northern Hemisphere. In the mid-troposphere, at a model height of 6.4 km (~ 450 hPa) (Figure 17b), there are large areas showing an increase in OH by up to 20–25% with the new flash-rate parameterisations, particularly in the tropics and Southern Hemisphere. The broad hemispheric differences in OH are qualitatively similar to those for O$_3$. With an increase in NO due to the new flash-rate parameterisation, OH increases (e.g. via the recycling of HO$_2$ by reaction with NO). In highly polluted air, NO$_2$ can be an OH sink (Lelieveld et al., 2016). Of course, transport would also influence these patterns, both horizontally and vertically.

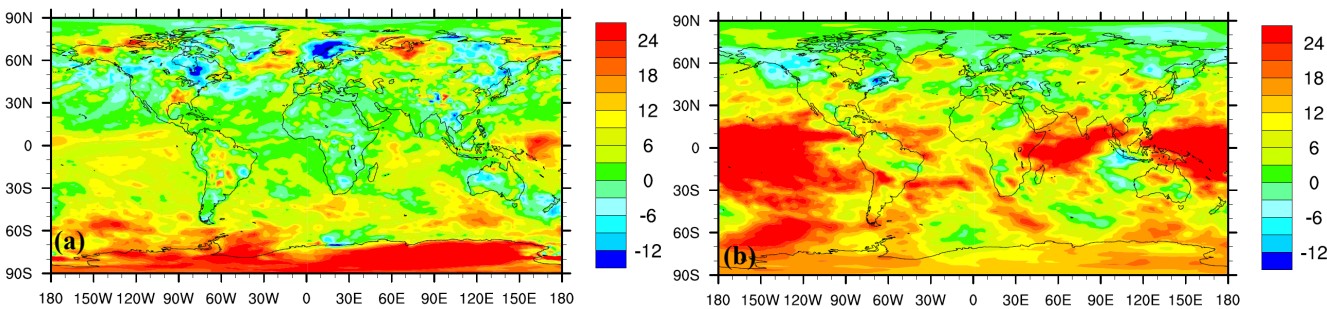

**Figure 17: Mean relative difference (%) between the hydroxyl radical (OH) predicted using the new lightning flash-rate parameterisations (Run 2) and the default PR92 parameterisations (Run 1): (a) at 20 m (the lowest model level) and (b) at 6400 m (~ 450 hPa).**

Overall, we find that, with the new flash-rate parameterisations, there is a 13% increase in the annual-average volume-weighted global tropospheric OH, from $10.6 \times 10^5$ to $12.0 \times 10^5$ molecules cm$^{-3}$. The increase over the ocean is by $1.6 \times 10^5$ (16.3%) and that over land by $0.9 \times 10^5$ molecules cm$^{-3}$ (7.6%). For comparison, the respective values obtained from the model simulation with zero LNO$_x$ emissions are $7.6 \times$, $7.3 \times$ and $8.1 \times 10^5$ molecules cm$^{-3}$.

The global amount can be compared with the ACCMIP multi-model mean of $11.1 \pm 1.6 \times 10^5$ molecules cm$^{-3}$ derived by Naik et al. (2013) for the year 2000. Recent observationally based values reported by Wolfe et al. (2019) for August 2016 are $12.6 \pm 2.9 \times 10^5$ for the Northern Hemisphere and $8.1 \pm 1.9 \times 10^5$ molecules cm$^{-3}$ for the Southern Hemisphere, and these for February 2017 are $8.8 \pm 2.1 \times 10^5$ and $11.4 \pm 2.8 \times 10^5$ molecules cm$^{-3}$, respectively. These can be compared with the corresponding modelled values $16.9 \times$, $7.3 \times$, $8.0 \times$ and $11.6 \times 10^5$ using the PR92 scheme, and $18.7 \times$, $8.5 \times$, $9.2 \times$ and $13.5 \times 10^5$ molecules cm$^{-3}$ using the new scheme. The LNO$_x$-induced increase in OH due to the new scheme adds to the model high bias in the OH burden in summer, whereas it reduces the magnitude of the bias in winter with the bias shifting from low to high. The model value in the Northern Hemisphere in August is considerably larger than the observation even with the PR92 scheme. It is known that the UKCA StratTrop configuration yields substantially larger OH in the Northern Tropics at low altitudes compared to observations and to the ACCMIP multi-model estimates (Archibald et al., 2020).

The surface methane concentrations were prescribed in the model and methane was allowed to undergo loss processes in the rest of the atmosphere. With an overall increase in OH using the new flash-rate parameterisations in ACCESS-UKCA, the global annual mean lifetime of CH$_4$ against loss by tropospheric OH ($\tau_{CH4\_OH}$, defined as the division of the global total atmospheric CH$_4$ burden and the globally integrated CH$_4$ loss rate by reaction with tropospheric OH) decreases by 6.7%, from 7.5 to 7.0 years. This value without the LNO emissions in the model is 9.2 years. The modelled methane lifetime is lower than the multi-model mean $9.7 \pm 1.5$ years reported by Naik et al. (2013), which could be due to a higher tropospheric burden of non-lightning related NO$_x$ in ACCESS-UKCA and/or a more intense photolysis.

## 4.4 Carbon monoxide (CO)

There is a decrease in the modelled total tropospheric carbon monoxide (CO) with the use of the new lightning parameterisations, as evident from the zonal annual-mean difference plot in Figure 18 (this CO reduction is coupled to the OH increase, via the reaction $OH + CO \rightarrow CO_2 + H$). In the lower troposphere, the decrease is by approximately 4–6 ppb (~ 7%) in the Southern Hemisphere and 2–4 ppb (~ 3%) in the Northern Hemisphere. This reduction gets a little larger in the mid to upper troposphere in the tropics. Overall, the reduction in the volume-weighted global CO is 4.5 ppb (5.6%). Over the ocean is 4.7 ppb (6.2%) and that over land 4.0 ppb (4.5%). The volume-weighted tropospheric CO obtained from the PR92 scheme is 80.3 ppb over the globe, 75.4 ppb over the ocean and 89.9 ppb over land. With the new flash-rate scheme, these values decrease by 4.5 ppb (5.6%), 4.7 ppb (6.2%) and 4.0 ppb (4.5%), respectively, and can be compared with the values 96.4, 91.8 and 105.4 ppb, respectively, obtained from the model simulation with no $LNO_x$ emissions.

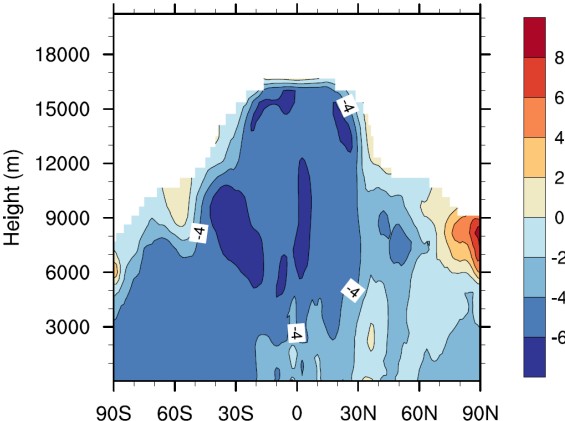

**Figure 18: The difference between the zonal total annual-mean tropospheric CO (ppb, by volume) modelled using the new lightning flash-rate parameterisations (Run 2) and the default PR92 parameterisations (Run 1).**

In Figure 19, we compare the modelled monthly-averaged CO with surface flask observations from the same GAW-WDCRG sites as in Figure 12 (data from Minamitorishima were missing for 4 months, so are not presented). With the use of the new lightning parameterisation, the relative change in the modelled yearly averaged CO at Ushuaia, Cape Grim, Mauna Loa and Mace Head is -8.1%, -9.8%, -3.8%, and -0.3%, respectively. The modelled ground-level CO is affected only very marginally by the flash-rate modification compared to the magnitude of the model-data differences, except at Ushuaia and at Cape Grim during the austral summer. Clearly, as in the case of ground-level $O_3$, the lightning changes alone do not reconcile the differences between the modelled CO and observations.

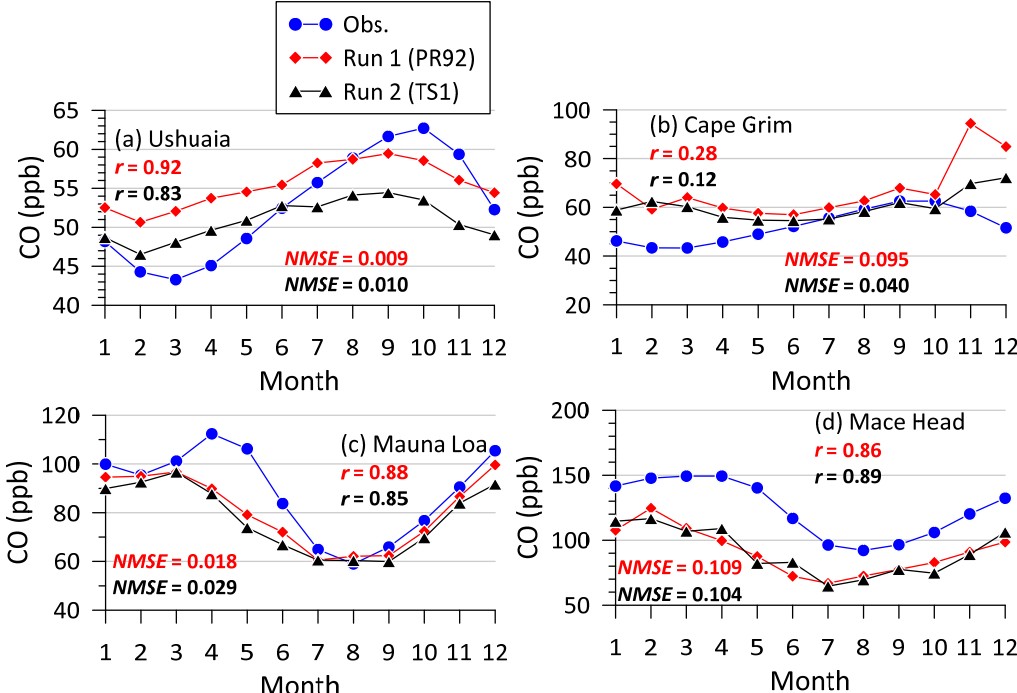

**Figure 19: Comparison of the modelled monthly-averaged CO concentrations (ppb, by volume) with observations at four oceanic/coastal ground stations for the year 2006. The two model runs are with the (a) default PR92 parameterisations (Run 1) and (b) new lightning flash-rate parameterisations (Run 2). The values of correlation coefficient (*r*) and normalised mean square error (NMSE) are also shown.**

## 5 Model simulations with meteorological nudging

As mentioned in Section 3.5, we also did Run 1 (PR92) and Run 2 (TS1) with meteorological nudging (which could impact convection indirectly in the model) with the same initial conditions as for the free running simulations. Broadly speaking, the results were not too different from the respective free running simulation results for the year 2006 reported above. The averaged global, land and ocean flash rates obtained from Run 1 with nudging were 32.87, 32.50 and 0.37 flashes s$^{-1}$, respectively, which are very similar to 32.92, 32.56 and 0.36 flashes s$^{-1}$, respectively, obtained from the free running Run 1 in Table 1. These obtained from Run 2 with nudging were 46.88, 36.80 and 10.08 flashes s$^{-1}$, respectively, which are on average 5% higher than 44.96, 35.88 and 9.08 flashes s$^{-1}$, respectively, obtained from the free running Run 2 in Table 1 (so a scaling factor of 0.95 would make the nudged model flash rates approximately match the free-running model flash rates). The total LNO$_x$ produced over the globe, land and ocean changed in the same proportions since the NO produced per flash was the same.

The annual-averaged global spatial distributions of the modelled flash rate with nudging for both Runs were very similar to the respective free running model plots shown in Figure 5c and Figure 5d, with the overall spatial correlation with the LIS/OTD satellite climatology (Figure 5a) slightly improved from 0.72 to 0.75, and some improvement in the model

performance over the Indian subcontinent but a slight deterioration over the southern US. Nudging would have an impact on tropospheric composition, which we have not presently investigated, but we estimate that with flash rate constrained the modelled tropospheric composition with and without nudging would, on average, be within ~5%.

**6 Conclusions**

We have critically examined parameterisations of lightning flash rate that are based on the cloud-top height approach. Testing of the widely used Price and Rind (1992) (PR92) parameterisations within the ACCESS-UKCA global chemistry-climate model for the year 2006 using the global LIS/OTD satellite data has revealed that while the parameterisation for land yields satisfactory predictions, with a globally averaged flash rate of 31.03 flashes $s^{-1}$ compared to the observed 34.92 flashes $s^{-1}$, the oceanic parameterisation severely underestimates the observed flash rate, yielding on average 0.33 flashes $s^{-1}$ compared to the observed 9.16 flashes $s^{-1}$. This leads to lightning-generated $NO_x$ ($LNO_x$) being underestimated proportionally over the ocean and thus influencing tropospheric composition. Any interannual variability in lightning was not investigated.

Following Boccippio's (2002) scaling relationships between thunderstorm electrical generator power and storm geometry as the basis, we derived alternative flash-rate parameterisations. While the new parameterisation for land performed slightly better than the corresponding PR92 one, giving a globally averaged flash rate of 34.23 flashes $s^{-1}$ compared to the observed 34.92 flashes $s^{-1}$, the new parameterisation for ocean performed more accurately, giving a globally averaged flash rate of 8.84 flashes $s^{-1}$ compared to the observed 9.16 flashes $s^{-1}$. We also tested an oceanic parameterisation by Michalon et al. (1999), which gives a global oceanic average of 11.31 flashes $s^{-1}$. With the new parameterisations, there was an increase in global $LNO_x$ from 4.8 to 6.6 Tg N $yr^{-1}$, with the new estimate comparable to $6.3 \pm 1.4$ Tg N $yr^{-1}$ obtained by Miyazaki et al. (2014) using an assimilation of multiple satellite datasets into a global CTM. There is a large uncertainty in the amount of NO produced per flash in the scientific literature. The model's use of 330 moles NO produced per flash is close to the average value 310 moles NO per flash determined by Miyazaki et al. (2014) using data assimilation, but requires better constraining.

The use of the new flash-rate parameterisations in ACCESS-UKCA demonstrated a considerable impact on the modelled tropospheric composition compared to the defaults PR92 parameterisations, mainly due to the change in the oceanic flash-rate component. In particular, the following impacts were observed:

- An increase in the mid- to upper-tropospheric $NO_x$ by as much as 40 ppt by volume (as $NO_2$) in the northern tropics. An overall increase in the global $NO_x$ by 8.7 ppt (15.7%) and by 9.9 ppt (28.0%) over the ocean. A better agreement of the modelled tropospheric $NO_2$ columns with the CAMS reanalysis data over the ocean.

- The tropospheric $O_3$ burden increased by 8.5%, from 284 to 308 Tg $O_3$, closer to a multi-model estimate of $337 \pm 23$ Tg $O_3$ (Young et al., 2013), the latter supported by measurement climatology. Overall, the distribution of the

modelled ozone in the troposphere improved somewhat compared to global observations in the Southern Hemisphere. There are considerable ozone biases in the model in the Northern Hemisphere that are not related to $LNO_x$.

- A 13% increase in the annual-average volume-weighted global tropospheric OH, from $10.6 \times 10^5$ to $12.0 \times 10^5$ molecules $cm^{-3}$.
- A decrease in the global annual mean methane lifetime against loss by tropospheric OH by 6.7%.
- An overall reduction in the global CO by 4.5 ppb by volume (5.6%).

The approach of parameterising lightning flash rate in terms of convective cloud-top height works well given its simplicity, and continues to be useful in accounting for $LNO_x$ in global models, although there were some significant spatial distributional differences in the modelled flash-rate density compared to the satellite data. The approach is also very sensitive due to an almost 5th power dependence on cloud-top height. With increased computational power in the future, it may be possible to understand and represent global $LNO_x$ in a better process-based manner through a cloud-resolving modelling framework with an explicit prediction of the electrical activity in storms.

Recent global chemistry-climate modelling studies using flash-rate parameterisations based on the convective cloud-top height show an increase in $LNO_x$ emissions in a future warming climate (e.g., Banerjee et al. 2014; Clark et al., 2017; Iglesias-Suarez et al., 2018), primarily as a result of increases in the depth of convection. This is also the case with a CAPE and precipitation rate based parameterisation (Romps et al., 2014). Conversely, it is found that a flash-rate scheme based on convective mass flux (Clark et al., 2017) and that based on upward cloud ice flux (Finney et al., 2018) predict a global decrease in future lightning flash density (under the RCP8.5 scenario). The study of Finney et al. (2018) argues that the ice flux method performs better than the PR92 cloud-top height method, and this should be reanalysed using the scheme proposed here which drastically alters the flash rate over the ocean. As well, there is an existing uncertainty as to which physical parameterisation approach best represents the reality and the feedbacks that are important for lightning under climate change. This will be difficult as parameters such as cloud ice content and/or updraught mass flux used in flux-based lightning schemes are poorly constrained by available observations.

**Data availability**

The ACCESS-UKCA global model output data (in NetCDF format) used for analysis and plotting, and the processed model lighting and composition data (in ASCII format) used for comparison with observations can be made available by contacting the corresponding author (Ashok Luhar: ashok.luhar@csiro.au). The observational datasets used in the present study were available from the following sources: the LIS/OTD lightning flash data from https://lightning.nsstc.nasa.gov/data/data_lis-otd-climatology.html, monthly mean vertical ozone profile data from http://www.bodekerscientific.com/data/monthly-mean-global-vertically-resolved-ozone, surface ozone and CO data from https://www.gaw-wdcrg.org, ERA-Interim global

reanalysis data from https://www.ecmwf.int/en/forecasts/datasets/reanalysis-datasets/era-interim, SHADOZ ozonesonde measurements from https://tropo.gsfc.nasa.gov/shadoz, OMI NO$_2$ column data from https://www.temis.nl/airpollution/no2.php, and CAMS global reanalysis data from https://ads.atmosphere.copernicus.eu/cdsapp#!/dataset/cams-global-reanalysis-eac4-monthly.

5 **Author contributions**

AKL devised the study, performed the flash-rate parameterisation formulation and model runs, analysed model output and data, and wrote the paper, with contributions and comments from all co-authors. IEG assisted with the flash-rate parameterisations and advised on various components of the paper, MTW performed some of the early model-data comparison for atmospheric composition and advised on the model setup, and NLA advised on the model configuration and 10 provided relevant technical details on UM-UKCA.

**Competing interests**

The authors declare that they have no conflict of interest.

**Acknowledgements**

This research was undertaken with the assistance of resources and services from the National Computational Infrastructure 15 (NCI), which is supported by the Australian government. Martin Dix of CSIRO is acknowledged for his help with model configuration issues. Fiona O'Connor and Mohit Dalvi of the U.K. Met Office are thanked for their assistance with the UKCA emission methodology and for answering questions about UM-UKCA at the early stages of model implementation. We would like to thank Bodeker Scientific, funded by the New Zealand Deep South National Science Challenge, for providing the Bodeker Scientific vertically resolved ozone database. Surface ozone and CO data from the World Data Centre 20 for Reactive Gases, tropospheric NO$_2$ column data from the OMI sensor from the Tropospheric Emission Monitoring Internet Service (TEMIS), ERA-Interim data from the European Centre for Medium-Range Weather Forecasts (ECMWF), and Copernicus' CAMS global reanalysis (EAC4) data were used in this research. Useful comments by the three anonymous referees, Fraser Dennison and a short comment by Declan Finney are much appreciated. We thank Kazuyuki Miyazaki for supplying the data corresponding to a diagram in Miyazaki et al. (2014).

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
