# Peer review of "Assessing and improving cloud-height based parameterisations of global lightning flash rate, and their impact on lightning-produced NOx and tropospheric composition in a chemistry-climate model"

_Atmospheric Chemistry and Physics, 2020_

## Referee Comment (RC1) · Anonymous Referee #3 · 2 Oct 2020

General remarks:

The paper by Luhar et al proposes a new CTH-based lightning scheme that considerably improves the maritime behaviour of the original CTH parameterization proposed by Price and Rind in 1992 (PR 1992).

The paper begins with is a relatively clear introduction (section 1). Then in section 2 the authors first describe the chemistry-climate model used followed by a description on how the lightning scheme and the NO per flash were implemented in the model. In

subsections 3.1/3.2/3.3 they describe three previous CTH-based schemes (including the one proposed by Boccippio in 2002 that inspired the authors' new lightning scheme) and the new lightning parameterization proposed (subsection 3.4). In subsection 3.5 the different model runs to be compared in the paper are commented. Subsection 3.6 is devoted to compare the lightning flash rates derived from four model runs with those from satellite observations.

The modelled LNOx, its vertical distribution and verification are described in section 3.7.1 (global LNOx), 3.7.2 (adopted vertical distribution of LNOx) and 3.7.3 (tropospheric NO2 verification), respectively. Finally, section 4 of the article is devoted to comment the impact of the new lightning scheme on some key chemical components of the troposphere with specific subsections for NOx (subsection 4.1), O3 (subsection 4.2), OH (subsection 4.3) and CO (subsection 4.4).

The paper is overall well written but requites some important clarifications. The figures need some improvement. In particular, the numbers inside Figures 10, 13, 14 and 17 are not readable and should be larger. Also the numbers in the vertical and horizontal axes of Figures 5, 6, 10, 11, 13, 14, 15 and 16 are small and not very visible. The numbers in the color bars should also be bigger.

Some more detailed comments:

Section 1

What do the authors mean in line 5 of page 4 with "... The performace of the PR92 flash-rate parameterizations has not yet been tested properly for their land and ocean components separately" ?.

There are already previous works indicating that the PR92 scheme exhibit large land-ocean biases. This has been already been pointed out by Finney et al 2014, 2016 and others as the author themselves state in the lines 3-5 of page 4. Please rephrase this sentence or make it clearer.

Section 2

What is the time step of the ACCESS-UKCA model used?.

The authors state that their ACCESS-UKCA setup includes some additional modifications compared to the base UM-UKCA v8.4 model. These changes seem to produce an increase (see line 20 of page 5) in the tropospheric O3 burden of about 12 %. Have the authors compared this increased O3 burden with observations?.

Section 2.1

What is the convection scheme used in the ACCESS-UKCA model?. This is important and should be clearly stated since any lightning scheme will be sensitive to the chosen convection parameterization. Please write it in the manuscript for the sake of clarity.

Did you use / implement the spatial calibration factor (c) introduced in PR92 and shown in equation (3)?. This is not clearly stated.

The authors should advise readers that the use of the method suggested by Price and Rind GRL 1993 to distinguish between CGs and ICs was only derived considering a number of thunderstorms in the US. However here the authors assume worldwide applicability. The authors should mention the restrictions and assumptions underlying such method. Also, it would be good if authors could say something about how the assumptions of the PR93 method can affect the results of the paper.

Section 2.2

The authors seem to assume that the energy of CG and IC flashes is the same. Is it so?. If yes, please state it clearly and add appropiate citations supporting this assumption (for instance Ridley et al 2005, Ott et al 2010 and / or others). It is also assumed that 330 moles NO / flash is produced independently of whether the flash is CG or IC. Why 330 ?.

In this paper the amount of NO per flash is prescribed to 330 moles NO / flash. What

is the underlying reason for choosing 330 moles NO / flash instead of the 310 moles NO / flash concluded by Miyazaki et al 2014?.

By using equation (15) in Price et al JGR 1997 the authors could estimate (assuming the energy per flash) the number of NO molecules produced per joule. This an interesting magnitude to show and it is possible since they have computed the amount of global LNOx (Table 4, equation (21)) and are assuming that the energies of CG and IC flashes are the same, and that aproximately 75 % of predicted total flashes per second (Table 1) are CGs while 25 % are ICs.

In line 16 of page 7, the authors comment a little bit how the produced amount of LNOx is vertically distributed. It is mentioned that it is distributed evenly vertically from 500 hPa (aprox 6 km) to the cloud top for IC flashes, and from surface to 500 hPa for CG flashes. What is the rationale and / or the physical, chemical and transport reasons (and / or possible observations) for choosing / supporting such vertical distribution ?. This is a bit obscure to me.

---

## Referee Comment (RC2) · Anonymous Referee #3 · 5 Oct 2020

Section 3.1

In line 15 of page 9, it is said "... by substituting Eq. (8) into Eq(13) ...", wouldn't it be the opposite?.

Section 3.5

According to the authors ACCESS-UKCA was setup as a free running simulation for 2 years (2005 and 2006), and the simulation was started using the model initial conditions taken from a nudged model run (see line 10 in page 13). I think the authors should

be a bit more specific on this technical matter. I underdtand that the nudging some-how guarantees / ensures that the basic dynamics in the lower-middle atmosphere is identical in simulations in which other changes (implementation of a lightning scheme for instance) are made. Is the nudging applied to all altitudes (pressure levels of the model)?. Also, it is not clearly indicated whether lightning was included (or not) in the first free running simulation. Was it?.

I consider that to "see" the influence of lightning only in a CTM one should proceed as the following: First run your code in a free-running dynamic mode without considering lightning. Then run a second model simulation also without lightning but now using the nudge, that is, the horizontal wind and temperature fields in the tropo-stratosphere are nudged at each model time step of the first free-running dynamic ACCESS-UKCA run. Then, what I would do, is to run a third nudged ACCESS-UKCA simulation (with the lightning scheme on) that is nudged to the first free-running dynamics ACCESS-UKCA run. Finally, I would repeat the third simulation for each of your lightning schemes (PR92 and your new one) and will always compare their output with the results of the second nudged ACCESS-UKCA runs. In this way you will ensure that you are really carrying out comparisons between simulations of the atmosphere with and without lightning that are not biased by dynamical effects.

It is not completely clear to me if your "nudged model run" is really free of dynamical effects. Please comment on this and try to be more specific.

Section 3.6

As a remark, by looking at Table 1 I see that the output of RUN 1 (PR92) gives quite low global lightning flash frequency (32.92 flashes / s). Note that the UKCA-UM model was already used by Finney et al 2016 and applied a scaling factor of 1.44 to match PR92 global flash frequency to LIS/OTD observations. In your case the scaling factor would be 1.40.

Could you please explain a bit the underlying reasons for your model runs (including

[Figure]

RUN 2) to underpredict in spring in the SH and NH and overpredict in autumn in the SH (see Figures 3 a/b) ?.

Both PR92 and the new lightning scheme proposed fail in accurately describing the tropical oceanic flash rate (see Fig. 4c). There is a considerable overestimation of RUN 1 (PR92) and RUN 2 (new lightning scheme). What is the reason for this?. This has consequences on the simulation results shown in Fig. 5a (observations) and Fig. 5d (new scheme) where the tropical oceanic overestimated flash rate is apparent. Please comment a bit on this behaviour.

Regarding land, note that North America, the Indian and Australian continents are not very much well described either in RUN 1 and RUN 2 (new scheme). Please give reasons for this.

In commenting the use of scaling factor for flash frequency (line 1 and 2 of page 20) you should also cite the works by Tost et al 2007, Finney et al 2016 and Clark et al 2017 (among others) that applied such scaling factors in different models.

Regarding scaling for NO produced per flash, authors have prescribed an amount of 330 moles NO/flash which immediately conditions the desired lightning generated NOx (LNOx) as can be clearly seen from equation (21). Any comment on this ?.

The authors are assuming that all lightning flashes produce 330 moles NO / flash (no matter if CG or IC and independently of occurring in land or ocean). However, it is known that CG strokes over water usually carry more charge into them which leads to a higher transported current. This is an indication that, on average, CGs over water are more energetic than CGs over land and, consequently, CGs over ocean would produce a larger LNOx (see the paper by Nag and Cummings in GRL 2017). The latter is an indication of different land / ocean convection regimes. This is not considered by any lightning scheme (quantifiying the occurrence rate, not the energy). Authors should add comments on these deficiencies so that readers can have a fair perspective of the many limitations of lightning schemes (any).

Section 3.7

Subsection 3.7.1

In my view the lack of scaling flash frequencies (and the fact of using a prescribed $P\_NO = 330$ moles NO per flash) artificially magnifies the difference between the PR92 LNOx (4.8 Tg N / yr) and the one resulting from the new lightning scheme (RUN 2) leading to 6.6 Tg N / yr. If authors would have scaled (to match observations) the flash frequencies of each tested lightning scheme (especially the one of PR92), the resulting LNOx of PR92 and TS1 would have been much closer.

In connection with this, I miss a deep discussion on the reasons for selecting 330 moles NO / flash. For example, there are recent papers (not cited by the authors) by Bucsela et al JGR-Atm 2019 and Allen et al JGR-Atm 2019 where, based on OMI + WWLLN observations, find that LNOx can be 180 moles NO / flash +- 100 in midlatitudes summertime NH. Complementarily, the paper by Allen et al 2019 finds that LNOx can range between 70 and 270 moles NO / flash in the tropics.

I disagree with the sentence in lines 11-12 of page 22 that the new flash-rate parameterization (Fig. 6b) agrees better with annual LNOx distribution obtained by Miyazaki et al 2014 (Fig. 6c). There are large land geographical regions (North America, Australia, India, EuroAsia) where the predicted LNOx by PR92 and the new scheme are pretty similar and very different with respect the LNOx distribution derived by Miyazaki et al 2014 (Fig. 6c). This is mainly due to the very different flash densities of both PR92 and RUN 2 (Fig. 5c and 5d) compared to observations (Fig. 5a). So, the global flash frequency (and LNOx) could be similar to considered observations (LIS / OTD and Miyazaki's 2014) but, to me, a more demanding comparison would require detailed comparison of flash frequencies (and LNOx) per continental region (North America, South America, Africa, EuroAsia, ...). Could the authors provide a Table showing such comparison?.

Subsection 3.7.2

The vertical distribution of LNOx is crucial. The authors compare their chosen vertical distribution with those of Pickering et al 1998 and Ott et al 2010. The paper adopts an alternative vertical distribution closer to Ott's. However I miss a full discussion explaining / supporting the reasons that moved the authors to use the vertical LNOx distribution (blue dots) shown in Fig. 7.

Please comment and justify your election of vertical distribution. Do you have supportive observations?. Why do you use these profiles?.

The relative energy of global ICs with respect to global CGs has consequences and / or conditions the LNOx vertical distributions. For instance, the vertical LNOx introduced by Pickering et al 1998 is consistent with their election of IC flashes being 10 % as energetic as CG flashes. In fact, according to Pickering et al 1998, if global IC flashes contained less than 10 % energy as CG flashes, the upper troposphere (UT) peak (upper part of the "C-shaped" distributions) in the mass profiles might not be as pronounced. Consequently, if ICs are equally energetic as CGs (as authors have assumed) the UT peak would be even more significant and this is not consistent with the vertical distribution used by the authors that rather seems to be a kind of mean between Pickering's and Ott's distribution. But, again, what are your physical / chemical / transport reasons supporting such profiles?.

Subsection 3.7.3

While I understand the authors' reasoning for tropospheric NO2 verification, I do not fully agree with your conclusions of this section.

As I see it, the conflict in your procedure starts in line 6 of page 25. Here you indicate that since N_v_trop_180 is not available from observations, you take the average of the curves (in Figure 8) showing the predicted N_v_trop_180, that is, the mean tropospheric NO2 column vs latitude resulting from RUN 1 (PR92) and RUN 2 (new scheme) over the reference longitude of 180 degrees in 2006. Doing this somehow "contaminates" the reference, that is, the CAMS data. This "contamination" leads to curves like

the ones shown in Fig. 9 where, inevitably, RUN 1 and RUN 2 for global, land and oceanic scenarios are strangely close to the CAMS values (considered as reference).

Do Fig. 9 show total NO2 columns or only the lightning contribution to the zonal annual-mean tropospheric NO2 column ?. If total, please state it clearly.

I miss comparison of your NO2 values (shown in Fig. 9) with NO2 values reported in Bucsela et al 2019 (see Fig. 3(a) there) from OMI + WWLLN observations in northern midlatitude regions.

Please elaborate on this a bit.

Section 4 / Impact on chemical tropospheric composition

Let me start by indicating that in this section I miss a more detailed discussion on explicit chemical reactions and species in the context of the production / loss of the lightning affected species (NOx, O3, OH and CO) in the different geopraphical regions. As mentioned in line 32 of section 2, the model includes 306 chemical reactions and 86 species. This chemical set (plus the aerosol chemistry) is quite rich so that key chemical processes could have been pointed out. This is not really done.

Please try to indicate the key processes that, according to the model's reaction set, play the most important role(s) for the formation / loss of each of the species investigated. This is very important and illuminating for the readers.

Section 4.1 (NOx)

As said above I think that the comparison between modelled tropospheric NO2 columns and observations shown in section 3.7.3 is not completely convincing.

Here you compare the total tropospheric NO2 colums resulting from PR92 and the new lightning scheme and its difference. I think it would have been clearer for readers to show only the corresponding lightning contributions to the tropospheric NO2 column.

Section 4.2 (O3)

Could you please indicate the explicit chemical mechanisms that (according to the adopted chemical set) are controlling the balance of O3 at 20 m and at 6400 m due to lightning activity ?. What are the key chemical processes controlling ozone population at the two considered reference altitudes?. Are they the same or different?. This is an interesting information not commented in the paper.

Why, according to the authors, the new lightning scheme is not really able to account for the O3 observations in Fig. 12(c) and Fig. 12 (e)?.

Section 4.3 (OH)

Could you please show only the lightning contribution to the total OH tropospheric column ?. It is also important to show readers what are the crucial chemical reactions due to the increase of OH at 20 m and 6400 m.

The authors openly admit that the UKCA StratTop configuration produces an overestimation of OH. It would be interesting for readers if the authors could dig into their chemical scheme and indicate what chemical processes could be playing a role (or could somehow explain) the modelled overestimation.

Please comment.

Section 4.4 (CO)

Are the authors showing in Fig. 17 the total annual-mean tropospheric CO or only the one due to lightning ?.

Recommendation:

This paper reports on a improved CTH-based lightning scheme with the maritime lightning flash frequency being more realistic that the one of the PR92 lightning parameterization. The paper could be published in ACP but only after the authors have appropriately answered the questions and comments that I have addressed. There a number of points that need clarification and improvement before this manuscript can

be accepted.

Please also note the supplement to this comment:
https://acp.copernicus.org/preprints/acp-2020-885/acp-2020-885-RC2-
supplement.pdf

**Supplement:**

**Comments to the Authors:**

Review of MS with titled "*Assesing and improving cloud-height based parameterizations of global lightning flash rate, and their impact on lightning-produced NOx and tropospheric composition*" by A. Luhar *et al.*, submitted for publication in Atmospheric Chemistry and Physics.

**General remarks:**

The paper by Luhar et al proposes a new CTH-based lightning scheme that considerably improves the maritime behaviour of the original CTH parameterization proposed by Price and Rind in 1992 (PR 1992).

There is a relatively clear introduction (**section 1**). Then in **section 2** the authors first describe the chemistry-climate model used followed by a description on how the lightning scheme and the NO per flash were implemented in the model. In **subsections 3.1/3.2/3.3** they describe three previous CTH-based schemes (including the one proposed by Boccippio in 2002 that inspired the authors' new lightning scheme) and the new lightning parameterization proposed (**subsection 3.4**). In **subsection 3.5** the different model runs to be compared in the paper are commented. **Subsection 3.6** is devoted to compare the lightning flash rates derived from four model runs with those from satellite observation. The modelled LNOx, its vertical distribution and verification are described in **section 3.7.1** (global LNOx), **3.7.2** (adopted vertical distribution of LNOx) and **3.7.3** (tropospheric NO2 verification), respectively. Finally, **section 4** of the article is devoted to comment the impact of the new lightning scheme on some key chemical components of the troposphere with specific subsections for NOx (subsection 4.1), O3 (subsection 4.2), OH (subsection 4.3) and CO (subsection 4.4).

The paper is overall well written but requites some clarifications. The figures need some improvement. In particular, the numbers inside Figures 10, 13, 14 and 17 are not readable and should be larger. Also the number in the vertical and horizontal axes of Figures 5, 6, 10, 11, 13, 14, 15 and 16 are small and not very visible. The numbers in the color bars should also be bigger.

Some more **detailed comments:**

**Section 1**

What do the authors mean in line 5 of page 4 with "*... The performace of the PR92 flash-rate parameterizations has not yet been tested properly for their land and ocean components separately*" ?.

There are previous works indicating that the PR92 scheme exhibit large land-ocean biases. This has been already been pointed out by Finney et al 2014, 2016 and others as the author themselves state in the lines 3-5 of page 4. Please rephrase this sentence or make it clearer.

**Section 2**

What is the time step of the ACCESS-UKCA model used?.

The authors state that their ACCESS-UKCA setup includes some additional modifications compared to the base UM-UKCA v8.4 model. These changes seem to produce an increase (see line 20 of page 5) in the tropospheric O3 burden of about 12 %. Have the authors compared this increased O3 burden with observations?.

**Section 2.1**

What is the convection scheme used in the ACCESS-UKCA model?. This is important and should be clearly stated since any lightning scheme will be sensitive to the chosen convection parameterization. Please write it in the manuscript for the sake of clarity.

Did you use / implement the spatial calibration factor (c) introduced in PR92 and shown in equation (3)?. This is not clearly stated.

The authors should advise readers that the use of the method suggested by Price and Rind GRL 1993 to distinguish between CGs and ICs was only derived considering a number of thunderstorms in the US. However here the authors assume worldwide applicability. The authors should mention the restrictions and assumptions underlying such method. Also, it would good if authors could say something about how the assumptions of the PR93 method could affect the results of the paper.

**Section 2.2**

The authors seem to assume that the energy of CG and IC flashes is the same. Is it so?. If yes, please state it clearly and add appropiate citations supporting this assumption (for instance Ridley et al 2005, Ott et al 2010 and / or others. It is also assumed that 330 moles NO / flash is produced independently of whether the flash is CG or IC. Why 330 ?.

In this paper the amount of NO per flash is prescribed to 330 moles NO / flash. What is the underlying reason for choosing 330 moles NO / flash **instead of** the 310 moles NO / flash concluded by Miyazaki et al 2014?.

By using equation (15) in Price et al JGR 1997 the authors could estimate (assuming the energy per flash) the number of NO molecules produced per joule. This an interesting magnitude to show and it is possible since they have computed the amount of global LNOx (Table 4, equation (21)) and are assuming that the energies of CG and IC flashes are the same, and that aproximately 75 % of predicted total flashes per second (Table 1) are CGs while 25 % are ICs.

**In line 16 of page 7**, the authors comment a little bit how the produced amount of LNOx is vertically distributed. It is mentioned that it is distributed evenly vertically from 500 hPa (aprox 6 km) to the cloud top for IC flashes, and from surface to 500 hPa for CG flashes. What is the rationale and / or the physical, chemical and transport reasons (and / or possible observations) for choosing / supporting such vertical distribution ?. This is a bit obscure to me.

**Section 3.1**

In line 15 of page 9, it is said "... by substituting Eq. (8) into Eq(13) ...", wouldn't it be the opposite?.

**Section 3.5**

According to the authors ACCESS-UKCA was setup as a free running simulation for 2 years (2005 and 2006), and the simulation was started using the model initial conditions taken from a nudged model run **(see line 10 in page 13)**. I think the authors should be a bit more specific on this technical matter. I underdtand that the nudging somehow guarantees / ensures that the basic dynamics in the lower-middle atmosphere is identical in simulations in which other changes (implementation of a lightning scheme for instance) are made. Is the nudging applied to all altitudes (pressure levels of the model)?. Also, it is not clearly indicated whether lightning was included (or

not) in the first free running simulation. Was it?.

I consider that to "see" the influence of **lightning only** in a CTM one should proceed as the following: **First** run your code in a free-running dynamic mode **without** considering lightning. Then run a **second** model simulation **also without lightning** but now using the nudge, that is, the horizontal wind and temperature fields in the tropo-stratosphere are nudged at each model time step of the first free-running dynamic ACCESS-UKCA run. **Then**, what I would do, is to run a third nudged ACCESS-UKCA simulation (**with the lightning scheme on**) that is nudged to the first free-running dynamics ACCESS-UKCA run. **Finally**, I would **repeat the third** simulation for each of your lightning schemes (PR92 and your new one) and **will always compare** their output with the results of the second nudged ACCESS-UKCA runs. In this way you will ensure that you are really carrying out comparisons between simulations of the atmosphere with and without lightning that are not biased by dynamical effects.

It is not completely clear to me if your "nudged model run" is really free of dynamical effects. Please comment on this and try to be more specific.

**Section 3.6**

As a remark, by looking at Table 1 I see that the output of RUN 1 (PR92) gives quite low global lightning flash frequency (32.92 flashes / s). Note that the UKCA-UM model was already used by Finney et al 2016 and applied a scaling factor of 1.44 to match PR92 global flash frequency to LIS/OTD observations. In your case the scaling factor would be 1.40.

Could you please explain a bit the underlying reasons for your model runs (including RUN 2) to underpredict in spring in the SH and NH and overpredict in autumn in the SH (see Figures 3 a/b) ?.

Both PR92 and the new lightning scheme proposed fail in accurately describing the tropical oceanic flash rate (see Fig. 4c). There is a considerable overestimation of RUN 1 (PR92) and RUN 2 (new lightning scheme). **What is the reason for this?.** This has consequences on the simulation results shown in Fig. 5a (observations) and Fig. 5d (new scheme) where the tropical oceanic overestimated flash rate is apparent. Please comment a bit on this behaviour.

Regarding land, note that North America, the Indian and Australian continents are not very much well described either in RUN 1 and RUN 2 (new scheme). Please give reasons for this.

In commenting the use of scaling factor for flash frequency (line 1 and 2 of page 20) **you should also cite** the works by Tost et al 2007, Finney et al 2016 and Clark et al 2017 (among others) that applied such scaling factors in different models.

Regarding scaling for NO produced per flash, **authors have prescribed** an amount of 330 moles NO/flash which immediately conditions the desired lightning generated NOx (LNOx) as can be clearly seen from equation (21). Any comment on this ?.

The authors are assuming that all lightning flashes produce 330 moles NO / flash (no matter if CG or IC and independently of occurring in land or ocean). **However**, it is known that CG strokes over water usually carry more charge into them which leads to a higher transported current. This is an indication that, on average, CGs over water are more energetic than CGs over land and, consequently, CGs over ocean would produce a larger LNOx (see the paper by Nag and Cummings in GRL 2017). The latter is an indication of different land / ocean convection regimes. This is not considered by any lightning scheme (quantifiying the occurrence rate, not the energy). Authors should add comments on these deficiencies so that readers can have a fair perspective of the many

limitations of lightning schemes (any).

**Section 3.7**

**Subsection 3.7.1**

In my view the lack of scaling flash frequencies (and the fact of using a prescribed P_NO = 330 moles NO per flash) artificially magnifies the difference between the PR92 LNOx (4.8 Tg N / yr) and the one resulting from the new lightning scheme (RUN 2) leading to 6.6 Tg N / yr. If authors would have scaled (to match observations) the flash frequencies of each tested lightning scheme (especially the one of PR92), the resulting LNOx of PR92 and TS1 would have been much closer.

In connection with this, I miss a deep discussion on the reasons for selecting 330 moles NO / flash. For example, there are recent papers (not cited by the authors) by Bucsela et al JGR-Atm 2019 and Allen et al JGR-Atm 2019 where, based on OMI + WWLLN observations, find that LNOx can be 180 moles NO / flash +- 100 in midlatitudes summertime NH. Complementarily, the paper by Allen et al 2019 finds that LNOx can range between 70 and 270 moles NO / flash in the tropics.

**I disagree** with the sentence in lines 11-12 of page 22 that the new flash-rate parameterization (Fig. 6b) agrees better with annual LNOx distribution obtained by Miyazaki et al 2014 (Fig. 6c). There are large land geographical regions (North America, Australia, India, EuroAsia) where the predicted LNOx by PR92 and the new scheme are pretty similar and very different with respect the LNOx distribution derived by Miyazaki et al 2014 (Fig. 6c). This is mainly due to the very different flash densities of both PR92 and RUN 2 (Fig. 5c and 5d) compared to observations (Fig. 5a). So, the global flash frequency (and LNOx) could be similar to considered observations (LIS / OTD and Miyazaki's 2014) but, to me, a more demanding comparison would require detailed comparison of flash frequencies (and LNOx) per continental region (North America, South America, Africa, EuroAsia, ...). **Could the authors provide a Table showing such comparison?.**

**Subsection 3.7.2**

The vertical distribution of LNOx is crucial. The authors compare their chosen vertical distribution with those of Pickering et al 1998 and Ott et al 2010. The paper adopts an alternative vertical distribution closer to Ott's. However I miss a full discussion explaining / supporting the reasons that moved the authors to use the vertical LNOx distribution (blue dots) shown in Fig. 7.

Please comment and justify your election of vertical distribution. **Do you have supportive observations?. Why do you use these profiles?.**

The relative energy of global ICs with respect to global CGs has consequences and / or conditions the LNOx vertical distributions. For instance, the vertical LNOx introduced by Pickering et al 1998 is consistent with their election of IC flashes being 10 % as energetic as CG flashes. In fact, according to Pickering et al 1998, if global IC flashes contained less than 10 % energy as CG flashes, the upper troposphere (UT) peak (upper part of the "C-shaped" distributions) in the mass profiles might not be as pronounced. Consequently, if ICs are equally energetic as CGs (as authors have assumed) the UT peak would be even more significant and this is not consistent with the vertical distribution used by the authors that rather seems to be a kind of mean between Pickering's and Ott's distribution. **But, again, what are your physical / chemical / transport reasons supporting such profiles?.**

**Subsection 3.7.3**

While I understand the authors' reasoning for tropospheric NO2 verification, I do not fully agree with your conclusions of this section.

As I see it, the conflict in your procedure starts in line 6 of page 25. Here you indicate that since N_v_trop_180 is not available from observations, you take the average of the curves (in Figure 8) showing the predicted N_v_trop_180, that is, the mean tropospheric NO2 column vs latitude resulting from RUN 1 (PR92) and RUN 2 (new scheme) over the reference longitude of 180 degrees in 2006. Doing this somehow "contaminates" the reference, that is, the CAMS data. This "contamination" leads to curves like the ones shown in Fig. 9 where, inevitably, RUN 1 and RUN 2 for global, land and oceanic scenarios are strangely close to the CAMS values (considered as reference).

Do Fig. 9 show total NO2 columns **or only** the lightning contribution to the zonal annual-mean tropospheric NO2 column ?. If total, please state it clearly.

I miss comparison of your NO2 values (shown in Fig. 9) with NO2 values reported in Bucsela et al 2019 (see Fig. 3(a) there) from OMI + WWLLN observations in northern midlatitude regions.

Please elaborate on this a bit.

**Section 4 / Impact on chemical tropospheric composition**

Let me start by indicating that in this section I miss a more detailed discussion on explicit chemical reactions and species in the context of the production / loss of the lightning affected species (NOx, O3, OH and CO) in the different geographical regions. As mentioned in line 32 of section 2, the model includes 306 chemical reactions and 86 species. This chemical set (plus the aerosol chemistry) is quite rich so that key chemical processes could have been pointed out. This is not really done.

Please try to indicate the key processes that, according to the model's reaction set, play the most important role(s) for the formation / loss of each of the species investigated. This is very important and illuminating for the readers.

**Section 4.1 (NOx)**

As said above I think that the comparison between modelled tropospheric NO2 columns and observations shown in section 3.7.3 is not completely convincing.

Here you compare the total tropospheric NO2 colums resulting from PR92 and the new lightning scheme and its difference. I think it would have been clearer for readers to show only the corresponding lightning contributions to the tropospheric NO2 column.

**Section 4.2 (O3)**

Could you please indicate the explicit chemical mechanisms that (according to the adopted chemical set) are controlling the balance of O3 at 20 m and at 6400 m due to lightning activity ?. What are the key chemical processes controlling ozone population at the two considered reference altitudes?. Are they the same or different?. This is an interesting information not commented in the paper.

Why, according to the authors, the new lightning scheme is not really able to account for the O3 observations in Fig. 12(c) and Fig. 12 (e)?.

**Section 4.3 (OH)**

Could you please show only the lightning contribution to the total OH tropospheric column ?. It is also important to show readers what are the crucial chemical reactions due to the increase of OH at 20 m and 6400 m.

The authors openly admit that the UKCA StratTop configuration produces an overestimation of OH. It would be interesting for readers if the authors could dig into their chemical scheme and indicate what chemical processes could be playing a role (or could somehow explain) the modelled overestimation.

Please comment.

**Section 4.4 (CO)**

Are the authors showing in Fig. 17 **the total** annual-mean tropospheric CO or only the one due to lightning ?.

**Recommendation:**

This paper reports on a improved CTH-based lightning scheme with the maritime lightning flash frequency being more realistic that the one of the PR92 lightning parameterization. The paper could be published in ACP but only after the authors have appropriately answered the questions and comments that I have addressed. There a number of points that need clarification and improvement before this manuscript can be accepted.

---

## Referee Comment (RC3) · Anonymous Referee #1 · 6 Oct 2020

General Comments: Well written manuscript that is nearly ready for publication.

Scientific Comments:

Abstract reads well. My only quibble is that PR92's deficiencies over water have been well known for years. Perhaps you should go with . . . via the Price and Rind (1992) (PR92) formulation, whose water parameterization is known to greatly underestimate flashes.

P5 L19: Be clear that the k referred to in line 19 is the same as the one referred to
in equation (2). This could be done by removing the line 19 bullet and including this sentence in the preceding paragraph.

P6L23: What is the justification for parameterizing the thickness of the cold cloud region in terms of latitude when you could use temperature profiles from the driving model? What errors are induced by assuming a simple dependence on latitude for the cold-cloud region?

P12L25: How different would NO and NL be if you used the same value of k1 for land and ocean points, i.e., would the difference and values be comparable to that shown in Michalon.

P13 L5: When specifying flash rates, is continental-marine synonymous with land-ocean or is the convection in grid boxes within x km of the coast classified as continental in character? Or to put it another way: Should there be a transition zone over the ocean where flashes are still parameterized as if they are continental in character?

P16 L10-11 How sensitive is the conclusion (that Run 2 does the best) to the choice of meteorological model? Or to put it another way, how likely is it that the Michalon approach would do better if the cloud top heights were taken from a different model?

P18 L21: What do you mean by "significant spatial differences"? High-biases? Low-biases?

P23 L17: What is your rationale for distributing IC and CG emissions as you do? Is it motivated by the results of Pickering, Ott. Other? Wouldn't sub-grid scale mixing lead to some overlap in altitude between where NO from IC and CG flashes is deposited? How did you choose 500 hPa as the dividing line?

P28 L13: Several of the sites you selected are in regions where the impact of NOx from lightning is minimal. Please add a comparison or two to profiles from locations within the SHADOZ network that may be more affected by lightning-NOx. You could use boundary layer values from the profiles if you wish to retain your focus on the surface.

P28L15: Is LNOx an important source of NOx at any of these sites? Are differences between the models larger in the winter or summer? I would assume the latter; however, it isn't clear at Ushuaia for e.g..

P33 L3: You should state that the LNOx-induced increase in OH due to the new scheme exacerbates the model high bias in OH burden. – but a softer adjective than "exacerbate" is fine.

P36 L6: Would it make sense to give the locations of all of the data sets here as opposed to referring back to the manuscript or the acknowledgements section.

Technical Details:

P2L11 NO and NO2 have already been defined.

P5L18 mb — hPa

P6 L12: as a function of and H — as a function of H

P6 L14: (discusses later) — (discussed later)

P10 L10: when used in global models — when used over water in global models

P28 L8: global ozone — global annual mean ozone

P31 L1 The three "by"s are not necessary and can be removed.

P32 L8: At mid-troposphere — In the mid-troposphere

P34 L2 The NMSE and r values suggest a mixed result. — Be specific as to what you mean here.

P36 L1 "perhaps currently not well constrained" — poorly constrained

---

## Short Comment (SC1) · 30 Oct 2020

Thank you to the authors for their work. I appreciate their bringing together these details regarding the cloud-top height scheme, and its importance in atmospheric chemistry models.

I have a comment relating to p3(L26-29) and p25(L29-31).

From the evaluation of Clark et al. (2017), I would say that there is not much between the PR92 scheme and the Yoshida et al. (2009) scheme based on cold-cloud depth

(CCD). The CCD scheme does, however, show a much smaller increase in lightning activity in the climate change projections.

The CCD scheme incorporates the freezing level, and indirectly relates to the climate change effects on cloud structure and cloud ice. Therefore, I see it as an important lightning scheme that includes the popular cloud-top height variable but also doesn't ignore potential changes in cloud structure under climate change.

Have the authors considered the scheme? And I would like to suggest that this alternative approach to modifying the cloud-top height-based lightning scheme is at least presented and discussed in their paper.

Clark et al. (2017) GRL. https://doi.org/10.1002/2017GL073017

Yoshida et al. (2009) JGR-Atmos. https://doi.org/10.1029/2008JD010370
* * *

---

## Referee Comment (RC4) · Anonymous Referee #2 · 2 Nov 2020

Luhar et al. 2020 implement an alternative lightning flash rate parameterization following Boccippio 2002 in the ACCESS-UKCA global chemistry climate model. The new parameterization is evaluated by comparison to satellite observations of lightning and showed that it yields a better agreement than the default lightning parameterization (PR92). This study then assesses the impact of the new parameterization on the model simulation of NOx, O3, OH and CO. Not surprisingly, the results demonstrate that the relatively small amount of NOx emitted by lightning, leads to a disproportionately large impact on middle-to-upper tropospheric chemistry. Accurate representation of lightning

and lighting NOx thus is essential to accurate chemistry and climate models.

I have two major comments. First, a model run utilizing the parameterization following Boccippio 2002 (referred BO02 hereafter) is missing. Table 1 lists out the four parameterizations discussed in this study. The new alternative parameterization proposed in this study is quite similar to BO02. As I read it, the only significant difference is changing the linear coefficient by a factor of 2 for lightning parameterization over ocean (Fo). Sec 3.6 shows that switching from PR 92 to this new parameterization improves model's performance on producing lightning flash rates. However, it's unclear that whether this improvement can be achieved by just switching to BO02 or not. This addition of a model run using BO02 is needed to demonstrate that the modification on BO02 suggested in this study is essential.

Second, in Sec 4.2 and 4.4, the authors incorporated ground-base in-situ observations of O3 and CO and compared the model's simulations against the observations. They concluded that the model using the new parameterization outperforms the one using PR92 and yields better agreements of O3 and CO with in-situ observations. However, the results shown from Figure 12 and Figure 18 are not convincing enough to support the conclusions. The lightning parameterizations only lead to marginal changes in monthly averaged O3 and CO, and these effects are not obviously responsible for reconciling the difference between model and observation.

Overall, this paper appears an incremental improvement but it does offer some new insights and should be published after attention to these comments and the ones below.

Other specific comments:

Page 4 Line 5-6: All cited papers listed above evaluate performances of PR92 over either land or ocean, or both. This statement doesn't hold with respect to all existing studies.

Page 5 Line 4: What's the chemical timestep? I understand the model timestep of 60

min as too large to solve chemistry properly. But if I am incorrect, some extra words to explain would be helpful.

Page 6 Line 10: The threshold of 5km for cloud thickness looks arbitrary, authors should discuss this in relation to the estimate of lightning flashes.

Page 8 Line 12: Based on the discussion of electrical dipole, cloud thickness seems to be a better parameter representing dipole separation and size of charge centers. It is not obvious to me that it links to cloud-top height.

Page 25 Line 11-13: The argument of better agreement over ocean using model run 2 is not convincing in terms of large uncertainty in the calculation. Nv,trop from CAMS is calculated using the average of the two curves as Nv,trop,180, which leads to over-estimate compared to Nv,trop calculated using model run1 (PR92) and underestimate compared to Nv,trop calculated using model run 2 (TS1) over the tropical region. Note in Figure 9b, the column NO2 ($\sim0.5X1015$) within the latitudes $\pm$ 30$^\circ$ is comparable to column NO2 ($\sim0.3X1015$) over the reference longitude shown in Figure 8. The better agreement between model run2 and CAMS shown in Figure 9b may be predominantly attributed to the uncertainty introduced in Nv,trop,180. To make the result more convincing, two model runs should be compared to two CAMS column NO2 datasets calculated using Nv,trop,180 from model run1 and run2, respectively.

Figure 14: It's very hard to conclude that the new parameterizations lead to modelled ozone closer to the observations from this figure. A better visualization is suggested, for instance, set zero to white color, use relative difference plot etc.

[Figure]

---

## Author Comment (AC1) · 13 Mar 2021

**Reply by the authors to Referee #1's comments on**
**"Assessing and improving cloud-height based parameterisations of global lightning flash rate, and their impact on lightning-produced NO$_x$ and tropospheric composition" (#acp-2020-885)**

**Anonymous Referee #1 (RC3)**

We are grateful to the Referee for taking the time to read our manuscript and making a number of valuable comments. In the following, we provide our responses to these comments (the Referee's comments are shown in blue). The locations of the changes made refer to those in the non-tracked version of the revised manuscript.

General Comments: Well written manuscript that is nearly ready for publication.

Thank you.

Scientific Comments:

Abstract reads well. My only quibble is that PR92's deficiencies over water have been well known for years. Perhaps you should go with . . . via the Price and Rind (1992) (PR92) formulation, whose water parameterization is known to greatly underestimate flashes.

**Response:** We agree with the Reviewer.

**Changes in manuscript:** The sentence in the abstract has been modified as follows:

"… via the Price and Rind (1992) (PR92) parameterisations for land and ocean, where the oceanic parameterisation is known to greatly underestimate flash rates."

Also, "as expected" inserted in the sentence to read "… the oceanic parameterisation, as expected, underestimates the observed flash-rate density severely,"

P5 L19: Be clear that the k referred to in line 19 is the same as the one referred to in equation (2). This could be done by removing the line 19 bullet and including this sentence in the preceding paragraph.

**Response:** Point taken.

**Changes in manuscript:** The suggested change made.

P6L23: What is the justification for parameterizing the thickness of the cold cloud region in terms of latitude when you could use temperature profiles from the driving model? What errors are induced by assuming a simple dependence on latitude for the cold-cloud region?

**Response:** In the ACCESS-UKCA model, the method of apportioning the total number of flashes into cloud-to-ground (CG) and intracloud (IC) flashes is based on the commonly used empirical parametrisation of Price and Rind (1993) which was developed using cloud height and IC/CG ratio data for 139 individual thunderstorms. In the Price and Rind (1993) parameterisation, the ratio $z_R$ = IC/CG increases as a sole function of the thickness ($dH$) of the cold cloud region in thunderstorms (from 0°C to cloud top), and $dH$ is parameterised as a decreasing function of latitude.

ACCESS-UKCA uses the Price and Rind (1993) parameterisation in its entirety, including the thickness of the cold cloud region as a function of latitude. We find that, on average, 24% of the modelled flashes are CG flashes, which is the same amount obtained by Price and Rind (1994) using the same parameterisation.

As stated by the Referee, another possible way to compute the thickness of the cold cloud region is to directly use the temperature profiles from the model. There may be some sensitivity of the ratio $z_R$ = IC/CG to which of the two ways $dH$ is calculated and its impact on LNO$_x$. However, we think that this sensitivity would be relatively small in our study considering the following.

Firstly, in the present model setup, the CG and IC proportions are only used in estimating the vertical LNO$_x$ profile (the amount of LNO$_x$ produced per CG and IC flash is taken to be the same). Therefore, the total LNO$_x$ production is unaffected.

Secondly, Price and Rind (1993) observed that the correlation between observations of the thickness of the cold cloud region and the $dH$ parameterisation based on latitude was quite high (= 0.90).

Thirdly, our estimate of CG flashes being 24% of the total is similar to the value 20% obtained by Barthe and Barth (2008) who directly computed $dH$ as the thickness between the average height of the 0°C isotherm across the model-generated cloud and the average height of where the computed total hydrometeor mixing ratio decreases to $10^{-5}$ kg kg$^{-1}$ at the top of the cloud.

**Changes in manuscript:** The last paragraph of Section 2.1 (P7L12–23) is expanded to read:

"We use Eq. (4) along with Eq. (3) to calculate the total flash rate $f_{L,O}$ which is then apportioned into cloud-to-ground (CG) and intracloud (IC) flash rates using an empirical parameterisation for the ratio $z_R$ = IC/CG developed by Price and Rind (1993) (PR93) based on thunderstorm observations in the western United States. In this parameterisation, $z_R$ increases as a function of the thickness ($dH$) of the cold cloud region in thunderstorms (from 0°C to cloud top), and $dH$ is parameterised as a decreasing function of latitude. The PR93 parameterisation has been used frequently, with further validation for case studies reported by Pickering et al. (1998) and Fehr et al. (2004). Allen and Pickering (2002) and Grewe et al. (2001) used it in global atmospheric chemistry models, with the former evaluating it for cases in the US. The averaged values of $z_R$ and the CG to total flash ratio obtained from the PR93 parameterisation in the present study are 3.14 and 0.24, respectively. These values are comparable to $z_R \sim 4$ and the CG to total flash ratio $\sim 0.2$ obtained by Barthe and Barth (2008) using $dH$ calculated directly from modelled cloud temperature and total hydrometeor mixing ratio in the PR93 parameterisation. Using IC/CG measurements, Bond et al. (2002) derived a parameterisation for $z_R$ as a linearly decreasing function of latitude and obtained $z_R$ = 3.76 and the CG to total flash ratio = 0.21 over the tropics (35°N–35°S)"

P12L25: How different would NO and NL be if you used the same value of k1 for land and ocean points, i.e., would the difference and values be comparable to that shown in Michalon.

**Response:** This sentence was meant to be qualitative in nature. It is difficult to establish an exact equivalence between the $N_O$ and $N_L$ values used by Michalon et al. (1999) in deriving their oceanic flash rate ($F_O$) expression and the $k_1$ values used in our parameterisations, because the power law dependence on cloud-top height is different in these expressions. But neglecting the differences in this dependence and assuming the same logic as Michalon et al.'s in deriving their oceanic flash rate, the different $k_1$ values for land ($k_{1L}$) and ocean ($k_{1O}$) can be linked via $k_{1O}/k_{1L} = (N_O/N_L)^{2/3}$. So, if $k_{1O}$ is assumed to be the same as $k_{1L}$ (= $1.612 \times 10^{-5}$) then $N_O = N_L$. On the other hand, for $k_{1O} = 0.7 \times 10^{-5}$ as calculated for the oceanic component in the new parameterisation, $(N_O/N_L)^{2/3} = k_{1O}/k_{1L} = 0.43$. So, for $N_L$ = 600 per mg used by Michalon et al.

we get $N_O$ = 170 per mg compared with $N_O$ = 50 per mg used by Michalon et al. Thus, the differences in $k_1$ and $N_O/N_L$ do not match.

**Changes in manuscript:** Considering the above, we simply modify the sentence (P14L22–24) as "…for land and ocean can be interpreted in terms of $N_O$ and $N_L$ being different with $N_O < N_L$."

P13 L5: When specifying flash rates, is continental-marine synonymous with land-ocean or is the convection in grid boxes within x km of the coast classified as continental in character? Or to put it another way: Should there be a transition zone over the ocean where flashes are still parameterized as if they are continental in character?

**Response:** We use 'continental-marine' and 'land-ocean' synonymously. In the model, any grid cell that has a non-zero land surface fraction is considered land for the purposes of lightning $NO_x$ calculation. Conversely, only grid cells with 100% water surface coverage are considered ocean.

**Changes in manuscript:** This is made clear in the 2nd paragraph of Section 2.1 (P7L9–11).

P16 L10-11 How sensitive is the conclusion (that Run 2 does the best) to the choice of meteorological model? Or to put it another way, how likely is it that the Michalon approach would do better if the cloud top heights were taken from a different model?

**Response:** We do not have an alternative meteorological model or a convection scheme to test the sensitivity of the calculated lightning flash rates. The atmospheric component of the ACCESS-UKCA model used is the UK Met. Office Unified Model (UM). The convective cloud bottom level and top level used in the flash rate calculation are diagnosed from the UM convection scheme. Evaluation of the distribution of cloud depths simulated by the UM has been reported in studies such as Klein et al. (2013) and Hardiman et al. (2015) (referred to in the paper).

Some sensitivity of flash-rate parameterisations to meteorology can be examined by doing a model run with meteorological nudging. We have done additional model simulations with meteorological nudging, and summarise the results in a new Section 5 titled "Model simulations with meteorological nudging." In short, the average flash rates obtained from Run 2 with nudging were approximately 5% higher than those from the free running Run 2 in Table 1 (so a scaling factor of 0.95 would make the nudged model flash rates approximately match the free running model flash rates).

The Michalon et al. approach may or may not do better if the cloud top heights were taken from a different model. Our oceanic parameterisation is based on flash rate vs. cloud-top height observations as presented in Figure 1, and is, by design, able to represent the flash rate data in Figure 1 better than does the Michalon et al. parameterisation. Also, our parameterisation is consistent with Boccippio's (2002) scaling relationships.

One commonly used approach to get the modelled flash rate right is to first calculate a scaling factor that is the ratio of the observed globally averaged flash rate to the modelled globally averaged flash rate, and then apply this scaling factor to the modelled flash rate for subsequent model runs. This approach is useful as long as the relative global spatial distribution of flash rate is represented realistically. Otherwise, for example with the PR92 approach, the use of global scaling would lead to an overestimation of flash rate (and $LNO_x$) over land and underestimation over the ocean, although the global average flash rate (and $LNO_x$) may be correct.

**Changes in manuscript:** New Section 5 on "Model simulations with meteorological nudging" included. Some discussion on scaling is already given in the last paragraph of Section 3.6 (P22L6–17).

P18 L21: What do you mean by "significant spatial differences"? High-biases? Low-biases?

**Response:** Point taken.

**Changes in manuscript:** The sentence is modified (P21L1–5) to read "However, as shown in Figure 5d, ACCESS-UKCA with the new flash-rate parameterisations (Run 2) simulates the oceanic distribution of flash density much better than the PR92 scheme, although it is clear that there are some significant spatial differences (e.g. low bias over western Indian Ocean near southern Africa, and high bias over equatorial Indian Ocean and the Pacific) compared to the corresponding observations and climatology."

P23 L17: What is your rationale for distributing IC and CG emissions as you do? Is it motivated by the results of Pickering, Ott. Other? Wouldn't sub-grid scale mixing lead to some overlap in altitude between where NO from IC and CG flashes is deposited? How did you choose 500 hPa as the dividing line?

**Response:** In our model, once the amount of $LNO_x$ at a grid point location at a time step has been computed, it is distributed evenly in the vertical in log-pressure coordinate from 500 hPa to the cloud top for intra-cloud (IC) flashes, and from 500 hPa to surface for cloud-to-ground (CG) flashes. The method is motivated by the data analysis of Price and Rind (1993). Their observations from 139 thunderstorms cover cold cloud thickness (i.e. the cloud top height minus the freezing level) values ranging between 5.5–15 km and freezing level values between 2.7–5 km. The ratio $z_R$ = IC/CG increases from 0 to 4.6 with cold cloud thickness from 5.5 to 15 km, but remains relatively constant with freezing level. We take the level below which the CG generated $LNO_x$ is distributed as the observed minimum freezing level plus half of the minimum cold cloud thickness, i.e. $(2.7+5.5/2) \approx 5.5$ km. The selected 500 hPa level is closest to this 5.5 km value.

The use of the log-pressure (rather than linear pressure) coordinate yields a vertical distribution of $LNO_x$ that has more $LNO_x$ released at higher levels.

The non-uniform shape of the averaged modelled vertical distributions of $LNO_x$ is largely caused by the averaging of the $LNO_x$ profile from every time step over spatial and temporal variations in the cloud-top height.

Our averaged model profiles of $LNO_x$ in Figure 7 compare better with Ott et al. (2010). But we believe more observations are required to better understand the nature of $LNO_x$ distribution in the vertical and to constrain it.

**Changes in manuscript:** A new paragraph added just prior to Section 3 (P9L1–11).

P28 L13: Several of the sites you selected are in regions where the impact of NOx from lightning is minimal. Please add a comparison or two to profiles from locations within the SHADOZ network that may be more affected by lightning-NOx. You could use boundary layer values from the profiles if you wish to retain your focus on the surface.

**Response:** The selection of the ground-based sites, viz. Ushuaia, Cape Grim, Mauna Loa, Minamitorishima and Mace Head, was based on these sites being either oceanic or coastal and covering a range of latitudes. This emphasis on oceanic/coastal location was given because of the

relatively large difference between the PR92 oceanic flash rate parameterisation and the new one which would directly impact the modelled ozone at such locations.

But it was clear that $LNO_x$ impacts surface ozone very little compared to that at higher altitudes. And in that regard, we thank the Reviewer for the suggestion about the use of the SHADOZ (Southern Hemisphere ADditional OZonesondes) ozone profile data (https://tropo.gsfc.nasa.gov/shadoz/). We have now processed the SHADOZ data and made a comparison with the modelled ozone profiles at eight sites for the year 2006.

**Changes in manuscript:** In the Figure 12 description, we add "Apart from data availability and covering a range of latitudes, the site selection was based on these sites being either oceanic or coastal so that the relatively large difference between the PR92 oceanic flash rate parameterisation and the new one could be examined against the observations."

We have now included a comparison with the SHADOZ observations in Section 4.2 (P34L7–P35L17) and a new Figure 13. The comparison supports the results from the new oceanic flash-rate parameterisation for all southern hemispheric sites.

P28L15: Is LNOx an important source of NOx at any of these sites? Are differences between the models larger in the winter or summer? I would assume the latter; however, it isn't clear at Ushuaia for e.g..

**Response:** As mentioned in the response above, only oceanic/coastal sites were selected because the two flash rate parameterisations mostly differ in their oceanic treatment.

There were no $NO_x$ measurements at these sites, and here (in Figure 12) we compared ozone. There are small, but noticeable, differences in the modelled ozone as a result of the differences in the two flash rate parameterisations at these ground-based sites, which suggests that there is an impact of $LNO_x$, but it is much smaller compared to that the at higher altitudes (e.g. mid troposphere) (which is clear from the SHADOZ comparison in Figure 13 and the zonal difference plot in Figure 14c).

Generally, factors such as model's transport and chemical mechanisms, and input precursor emissions and their distributions are probably more influential in governing ozone model-data differences than $LNO_x$ near the Earth's surface.

There is no clear indication from the plots (Figure 12) if the differences in ozone between the two models are larger in the winter or summer, except for Ushuaia where the differences are larger during winter to spring. It is difficult to link the seasonal differences between the two modelled ozone variations linearly to those in $LNO_x$ due to the complexity of ozone chemistry coupled with transport.

**Changes in manuscript:** In the Figure 12 description, we add (P33L13–25) "Mauna Loa is located at an elevation of 3397 m on an island which is smaller in size than the grid resolution of the model and therefore it is difficult to correspond the sampling height to a particular vertical model level. We used the modelled concentrations from the bottom model level for all sites. The two model simulations describe the observed monthly variations reasonably well, except at Mauna Loa and Mace Head (the relatively large disagreement at Mauna Loa is likely due to the model resolution issues). There are small, but noticeable, differences in the modelled ozone from the two simulations. The relative change in the modelled yearly averaged $O_3$ at these ground-based sites with the use of the new lightning parameterisation is small, at 5.9%, 1.3%, -1.9%, 5.9% and 0%, respectively. There is some improvement in the modelled seasonal variation at Ushuaia, Cape Grim and Minamitorishima with the new $LNO_x$ scheme, but for the other two sites the model-data differences are much larger than those due to the $LNO_x$ changes. Generally, factors such as model's transport and chemical mechanisms, and input precursor emissions and their distributions are

probably more influential in governing ozone model-data differences than LNO$_x$ near the Earth's surface. There is no clear indication if the differences in ozone between the two models are larger in the winter or summer, except for Ushuaia where the differences are larger during winter to spring."

P33 L3: You should state that the LNOx-induced increase in OH due to the new scheme exacerbates the model high bias in OH burden. – but a softer adjective than "exacerbate" is fine.

**Response:** Point taken.

**Changes in manuscript:** Added (P39L15–17) "The LNO$_x$-induced increase in OH due to the new scheme adds to the model high bias in the OH burden in summer, whereas it reduces the magnitude of the bias in winter with the bias shifting from low to high."

P36 L6: Would it make sense to give the locations of all of the data sets here as opposed to referring back to the manuscript or the acknowledgements section.

**Response:** Point taken.

**Changes in manuscript:** The locations/sources of the datasets used are now provided under "Data availability" (P43).

Technical Details:

**Changes in manuscript:** Correction made.

**Changes in manuscript:** Correction made.

**Changes in manuscript:** Correction made.

**Changes in manuscript:** Correction made.

**Changes in manuscript:** Correction made.

**Changes in manuscript:** Correction made.

P31 L1 The three "by"s are not necessary and can be removed.

**Changes in manuscript:** Correction made.

P32 L8: At mid-troposphere — In the mid-troposphere

**Changes in manuscript:** Correction made.

P34 L2 The NMSE and r values suggest a mixed result. — Be specific as to what you mean here.

**Changes in manuscript:** The sentence is deleted, and we add (P40L16–21) "With the use of the new lightning parameterisation, the relative change in the modelled yearly averaged CO at Ushuaia, Cape Grim, Mauna Loa and Mace Head is -8.1%, -9.8%, -3.8%, and -0.3%, respectively. The modelled ground-level CO is affected only very marginally by the flash-rate modification compared to the magnitude of the model-data differences, except at Ushuaia and at Cape Grim during the austral summer. Clearly, as in the case of ground-level $O_3$, the lightning changes alone do not reconcile the differences between the modelled CO and observations."

P36 L1 "perhaps currently not well constrained" — poorly constrained

**Changes in manuscript:** Correction made.

---

## Author Comment (AC2) · 13 Mar 2021

**Reply by the authors to Referee #2's comments on**
**"Assessing and improving cloud-height based parameterisations of global lightning flash rate, and their impact on lightning-produced NO$_x$ and tropospheric composition"** (#acp-2020-885)

**Anonymous Referee #2 (RC4)**

We are grateful to the Referee for taking the time to read our manuscript and making a number of valuable comments. In the following, we provide our responses to these comments (the Referee's comments are shown in blue). The locations of the changes made refer to those in the non-tracked version of the revised manuscript.

Luhar et al. 2020 implement an alternative lightning flash rate parameterization following Boccippio 2002 in the ACCESS-UKCA global chemistry climate model. The new parameterization is evaluated by comparison to satellite observations of lightning and showed that it yields a better agreement than the default lightning parameterization (PR92). This study then assesses the impact of the new parameterization on the model simulation of NOx, O3, OH and CO. Not surprisingly, the results demonstrate that the relatively small amount of NOx emitted by lightning, leads to a disproportionately large impact on middle-to-upper tropospheric chemistry. Accurate representation of lightning and lighting NOx thus is essential to accurate chemistry and climate models.

I have two major comments. First, a model run utilizing the parameterization following Boccippio 2002 (referred BO02 hereafter) is missing. Table 1 lists out the four parameterizations discussed in this study. The new alternative parameterization proposed in this study is quite similar to BO02. As I read it, the only significant difference is changing the linear coefficient by a factor of 2 for lightning parameterization over ocean (Fo). Sec 3.6 shows that switching from PR 92 to this new parameterization improves model's performance on producing lightning flash rates. However, it's unclear that whether this improvement can be achieved by just switching to BO02 or not. This addition of a model run using BO02 is needed to demonstrate that the modification on BO02 suggested in this study is essential.

**Response:** We thank the Referee and agree with the suggestion made. We have now included results from an additional simulation (Run 5) corresponding to Boccippio's (Bo02) flash-rate parameterisations (Eqs. (9) and (10) in our paper). As hinted by the Referee, the Bo02 parameterisations and the new/alternate parameterisations (Eqs. (18) and (20)) used in Run 2 (TS1), which are based on the Bo02 approach, only differ in the values of their linear coefficients.

It is clear from Table 1 that Run 5 (Bo02) leads to lightning flash frequencies over the ocean that are twice as large as the observations.

**Changes in manuscript:** As mentioned above, Run 5 (Bo02) has been added in Section 3.5 and the results discussed in the paper.

Second, in Sec 4.2 and 4.4, the authors incorporated ground-base in-situ observations of O3 and CO and compared the model's simulations against the observations. They concluded that the model using the new parameterization outperforms the one using PR92 and yields better agreements of O3 and CO with in-situ observations. However, the results shown from Figure 12 and Figure 18 are not convincing enough to support the conclusions. The lightning parameterizations only lead to

marginal changes in monthly averaged O3 and CO, and these effects are not obviously responsible for reconciling the difference between model and observation.

**Response:** We agree with the Reviewer. The focus here needs to be more on how the flash-rate modification impacts ground-level $O_3$ and CO, and less on the model-data comparison. We recognise that lightning $NO_x$ alone cannot explain the model-data differences and that there would be other factors at play that are responsible for the large differences between the modelled values and observations at some of the sites. The relevant text in the paper has been modified accordingly.

**Changes in manuscript:** In Section 4.2 (P33L13–23), we add "Mauna Loa is located at an elevation of 3397 m on an island which is smaller in size than the grid resolution of the model and therefore it is difficult to correspond the sampling height to a particular vertical model level. We used the modelled concentrations from the bottom model level for all sites. The two model simulations describe the observed monthly variations reasonably well, except at Mauna Loa and Mace Head (the relatively large disagreement at Mauna Loa is likely due to the model resolution issues). There are small, but noticeable, differences in the modelled ozone from the two simulations. The relative change in the modelled yearly averaged $O_3$ at these ground-based sites with the use of the new lightning parameterisation is small, at 5.9%, 1.3%, -1.9%, 5.9% and 0%, respectively. There is some improvement in the modelled seasonal variation at Ushuaia, Cape Grim and Minamitorishima with the new $LNO_x$ scheme, but for the other two sites the model-data differences are much larger than those due to the $LNO_x$ changes. Generally, factors such as model's transport and chemical mechanisms, and input precursor emissions and their distributions are probably more influential in governing ozone model-data differences than $LNO_x$ near the Earth's surface."

In Section 4.4 (P40L16–21), we say "With the use of the new lightning parameterisation, the relative change in the modelled yearly averaged CO at Ushuaia, Cape Grim, Mauna Loa and Mace Head is -8.1%, -9.8%, -3.8%, and -0.3%, respectively. The modelled ground-level CO is affected only very marginally by the flash-rate modification compared to the magnitude of the model-data differences, except at Ushuaia and at Cape Grim during the austral summer. Clearly, as in the case of ground-level $O_3$, the lightning changes alone do not reconcile the differences between the modelled CO and observations."

We now also present a comparison with the SHADOZ ozonesonde measurements following a suggestion by Referee #1 (P34L7–P35L17).

Overall, this paper appears an incremental improvement but it does offer some new insights and should be published after attention to these comments and the ones below.

**Response:** Thank you for your comment.

Other specific comments:

Page 4 Line 5-6: All cited papers listed above evaluate performances of PR92 over either land or ocean, or both. This statement doesn't hold with respect to all existing studies.

**Response:** The intended emphasis in the statement was on 'properly', that is to say 'fully'. But following the Referee's comment, we have deleted the sentence.

**Changes in manuscript:** The sentence has been deleted.

 What's the chemical timestep? I understand the model timestep of 60 min as too large to solve chemistry properly. But if I am incorrect, some extra words to explain would be helpful.

**Response:** The model dynamical timestep is 20 minutes, the UKCA chemical solver is called every hour. It is a symbolic backward Euler solver with Newton-Raphson iteration, and runs to convergence, halving the step when required. Further information on the chemical solver used and its performance is given by Esenturk et al. (2018, Geosci. Model Dev., https://doi.org/10.5194/gmd-11-3089-2018).

**Changes in manuscript:** The above text is added and the reference Esenturk et al. (2018) included in the 2$^{nd}$ paragraph of Section 2 (P5L11–13).

 The threshold of 5km for cloud thickness looks arbitrary, authors should discuss this in relation to the estimate of lightning flashes.

**Response:** In the model's lightning scheme, a threshold convective cloud scale needs to be specified for it to constitute a thunderstorm. We use a minimum convective cloud thickness (i.e. the height of cloud top minus the height of cloud base) of 5 km for the lightning $NO_x$ to be activated. The cloud base and top are diagnosed on a time-step basis from the physical model's convection scheme. The selected threshold of 5 km is consistent with observations of the vertical scale of thunderstorms presented by several researchers, viz. Price and Rind (1992, 1993), Molinié and Pontikis (1995), and Ushio et al. (2001), which have a minimum value of approximately 5 km. Boccippio (2002) considered the Price and Rind (1992) data for cloud tops greater than 6 km.

While prescribing a minimum convective cloud thickness of 5 km for lightning is somewhat arbitrary, having no such threshold value would be unrealistic because then it would be implicitly assumed that a convective cloud always translates to a thunderstorm, and this would lead to unrealistically high flash rates. For example, removing this constraint in our base model (with the PR92 lightning scheme) increased the average global flash rate by 44%.

**Changes in manuscript:** Part of above discussion added to 2$^{nd}$ paragraph of Section 2.1 (P6L20–26). Also, please refer to the last paragraph of Section 3.6 (P22L6–17) on applying a scaling factor to modelled flash rate.

 Based on the discussion of electrical dipole, cloud thickness seems to be a better parameter representing dipole separation and size of charge centers. It is not obvious to me that it links to cloud-top height.

**Response:** In the conceptual picture of a thunderstorm as an electrical dipole used in developing the scaling relationships for the electrical power generated by the thunderstorm, it is assumed that the two cloud charges are spherical (each with radius $R$) and the dipole separation is $2R$. To derive an operationally useful and empirically testable scaling relationship, it is further assumed that the dipole separation varies as cloud-top height. Boccippio (2002) justifies this approximation by observations that in many storms the lower negative charge region remains relatively constant in height and that most upper positive charge is carried on small ice crystals with negligible terminal velocity. Thus, cloud-top height can be taken as a linear approximation of dipole separation.

**Changes in manuscript:** In the paper just above Eq. (5), we add "This assumption is based on observations that in many storms the lower negative charge region remains relatively constant in height and that most upper positive charge is carried on small ice crystals with negligible terminal velocity."

Page 25 Line 11-13: The argument of better agreement over ocean using model run 2 is not convincing in terms of large uncertainty in the calculation. Nv,trop from CAMS is calculated using the average of the two curves as Nv,trop,180, which leads to overestimate compared to Nv,trop calculated using model run1 (PR92) and underestimate compared to Nv,trop calculated using model run 2 (TS1) over the tropical region. Note in Figure 9b, the column NO2 (_0.5X1015) within the latitudes ± 30° is comparable to column NO2 (_0.3X1015) over the reference longitude shown in Figure 8. The better agreement between model run2 and CAMS shown in Figure 9b may be predominantly attributed to the uncertainty introduced in Nv,trop,180. To make the result more convincing, two model runs should be compared to two CAMS column NO2 datasets calculated using Nv,trop,180 from model run1 and run2, respectively.

**Response:** Thank you for this comment. A similar comment was also raised by Referee #3.

With regards to comparing the tropospheric NO2 columns, since we did not have $N_{v,trop,180}$ directly from observations, we used the model generated latitudinal variation of $N_{v,trop,180}$ in the derivation of the 'observed' $N_{v,trop}$. The quantity $N_{v,trop}$ thus obtained was then used to compare with the modelled $N_{v,trop}$. But, as the Referee has rightly pointed out, this approach influences the model-data comparison because the data then partially depend on the model results which in turn biases the comparison in favour of a better model performance.

The Referee's suggestion that $N_{v,trop,180}$ calculated separately for model Run1 and Run 2 should be used (rather than the average of the two) would not alleviate the core issue because $N_{v,trop,180}$ going into the determination of the observed $N_{v,trop}$ would still be dependent on the model results.

We have now used a much more justifiable approach whereby we calculate $N_{v,trop,180}$ directly from the Ozone Monitoring Instrument (OMI) satellite data of tropospheric NO2 columns (http://www.temis.nl/airpollution/no2.html; Boersma et al., 2017, 2018) and use this in the CAMS reanalysis data to obtain $N_{v,trop}$. With this, the model performance does not turn out to be as strong as before (as expected), but there is no change in the overall conclusion from the model-data comparison.

**Changes in manuscript:** The quantity $N_{v,trop,180}$ is now calculated using the Ozone Monitoring Instrument (OMI) satellite data of tropospheric NO2 columns (http://www.temis.nl/airpollution/no2.html; Boersma et al., 2017, 2018) and the model-data comparison is revised accordingly.

The pertinent Section 3.7.3 has been fully revised, including Figures 8 and 9 and Table 5, and additional references of Boersma et al. (2017, 2018). We have also changed the section heading from "Tropospheric NO2 verification" to "Modelled tropospheric total column NO2 and validation".

References of Boersma et al. (2017, http://temis.nl/qa4ecv/no2col/QA4ECV_NO2_PSD_v1.1.compressed.pdf; 2018, Atmos. Meas. Tech., https://doi.org/10.5194/amt-11-6651-2018) added.

Figure 14: It's very hard to conclude that the new parameterizations lead to modelled ozone closer to the observations from this figure. A better visualization is suggested, for instance, set zero to white color, use relative difference plot etc.

**Response:** Point taken. We have redrawn the plots (now Figure 15) with the range around zero set to white colour. Also, these are now relative difference plots (rather than absolute difference). An

additional Figure 15d is given showing the relative difference between the concentration modelled without any $LNO_x$ and the observations. The text has been changed to describe the modified plots.

**Changes in manuscript:** As above.

---

## Author Comment (AC3) · 13 Mar 2021

**Reply by the authors to Referee #3's comments on**
**"Assessing and improving cloud-height based parameterisations of global lightning flash rate, and their impact on lightning-produced NO$_x$ and tropospheric composition" (#acp-2020-885)**

**Anonymous Referee #3 (RC1)**

We are grateful to the Referee for taking the time to read our manuscript and making an extensive number of valuable comments. In the following, we provide our responses to these comments (the Referee's comments are shown in blue). The locations of the changes made refer to those in the non-tracked version of the revised manuscript. Details of only those references that are cited here but not in the revised paper are given here; details of all other references are given in the revised paper.

1)

General remarks:

The paper by Luhar et al proposes a new CTH-based lightning scheme that considerably improves the maritime behaviour of the original CTH parameterization proposed by Price and Rind in 1992 (PR 1992).

The paper begins with is a relatively clear introduction (section 1). Then in section 2 the authors first describe the chemistry-climate model used followed by a description on how the lightning scheme and the NO per flash were implemented in the model. In subsections 3.1/3.2/3.3 they describe three previous CTH-based schemes (including the one proposed by Boccippio in 2002 that inspired the authors' new lightning scheme) and the new lightning parameterization proposed (subsection 3.4). In subsection 3.5 the different model runs to be compared in the paper are commented. Subsection 3.6 is devoted to compare the lightning flash rates derived from four model runs with those from satellite observations.

The modelled LNOx, its vertical distribution and verification are described in section 3.7.1 (global LNOx), 3.7.2 (adopted vertical distribution of LNOx) and 3.7.3 (tropospheric NO2 verification), respectively. Finally, section 4 of the article is devoted to comment the impact of the new lightning scheme on some key chemical components of the troposphere with specific subsections for NOx (subsection 4.1), O3 (subsection 4.2), OH (subsection 4.3) and CO (subsection 4.4).

The paper is overall well written but requites some important clarifications. The figures need some improvement. In particular, the numbers inside Figures 10, 13, 14 and 17 are not readable and should be larger. Also the numbers in the vertical and horizontal axes of Figures 5, 6, 10, 11, 13, 14, 15 and 16 are small and not very visible. The numbers in the color bars should also be bigger.

**Response:** Thank you for the comment. As suggested, we have increased the font size of the numbers in the figures to improve readability.

**Changes in manuscript:** As above.

2)

Some more detailed comments:

Section 1

What do the authors mean in line 5 of page 4 with "... The performace of the PR92 flash-rate parameterizations has not yet been tested properly for their land and ocean components separately"?.

There are already previous works indicating that the PR92 scheme exhibit large landocean biases. This has been already been pointed out by Finney et al 2014, 2016 and others as the author themselves state in the lines 3-5 of page 4. Please rephrase this sentence or make it clearer.

**Response:** Point taken. There was a similar comment by Referee #1. This particular sentence is not necessary since the preceding sentence already cites relevant references and makes it clear that the PR92 parameterisation underestimates flash rate over the ocean.

**Changes in manuscript:** The sentence has been deleted.

3)

Section 2

What is the time step of the ACCESS-UKCA model used?.

**Response:** The model dynamical timestep is 20 minutes, the UKCA chemical solver is called every hour. It is a symbolic backward Euler solver with Newton-Raphson iteration, and runs to convergence, halving the step when required. Further information on the chemical solver used and its performance is given by Esentürk et al. (2018, Geosci. Model Dev., https://doi.org/10.5194/gmd-11-3089-2018).

**Changes in manuscript:** The above text is added and the reference Esentürk et al. (2018) included in the 2nd paragraph of Section 2 (P5L11–13).

4)

The authors state that their ACCESS-UKCA setup includes some additional modifications compared to the base UM-UKCA v8.4 model. These changes seem to produce an increase (see line 20 of page 5) in the tropospheric O3 burden of about 12 %. Have the authors compared this increased O3 burden with observations?.

**Response:** The ozone burden is commented upon in Section 4.2, giving a comparison with the multi-model mean value reported by Young et al. (2013) which they state is consistent with measurement climatologies.

**Changes in manuscript:** The sentence has been modified to read (P6L4–5) "The above changes lead to an increase in the modelled tropospheric ozone burden by about 12% (the first two changes by ~ 7% and the last by ~ 5%) to 284 Tg $O_3$ and this increase is towards the global modelling average (see Section 4.2)."

5)

Section 2.1

What is the convection scheme used in the ACCESS-UKCA model?. This is important and should be clearly stated since any lightning scheme will be sensitive to the chosen convection parameterization. Please write it in the manuscript for the sake of clarity.

**Response:** The convection scheme used in ACCESS-UKCA (vn8.4) is summarised by Walters et al. (2014). It is a mass flux scheme based on Gregory and Rowntree (1990) with various extensions

to include downdraughts and convective momentum transport. It consists of three stages: (a) diagnosis to determine whether convection is possible from the boundary layer, (b) a call to the shallow or deep convection scheme for all points diagnosed deep or shallow by the first step, and (c) a call to the mid-level convection scheme for all grid points.

The convective could base ($H_b$) is taken to be the air parcel ascent start level and the cloud top ($H$) is set to be the top of the ascent.

**Changes in manuscript:** The above is clarified in the 1ˢᵗ paragraph of Section 2.1 (P6L11–19).

6)

Did you use / implement the spatial calibration factor (c) introduced in PR92 and shown in equation (3)?. This is not clearly stated.

**Response:** Yes, we used the spatial calibration factor (Eq. (3)) which appears in Eq. (4).

**Changes in manuscript:** The above is clarified in the last paragraph of Section 2.1 (P7L12–14).

7)

The authors should advise readers that the use of the method suggested by Price and Rind GRL 1993 to distinguish between CGs and ICs was only derived considering a number of thunderstorms in the US. However here the authors assume worldwide applicability. The authors should mention the restrictions and assumptions underlying such method. Also, it would be good if authors could say something about how the assumptions of the PR93 method can affect the results of the paper.

**Response:** We now mention that the Price and Rind (1993) (PR93) parameterisation for the ratio $z_R$ = IC/CG which is used to partition the total flash rate into the CG and IC flash rates was based on thunderstorm observations in the western United States.

Regarding its worldwide applicability, one problem, as far as we know, is that there are no satellite measurements of cloud-to-ground flashes covering the whole globe that can be used for testing this or any other parameterisation.

However, the PR93 parameterisation has been used frequently and perhaps the best that exists. Further validation of this comes from studies including the following.

In two mid-latitude continental events (in the US) in which CG flash data were available, Pickering et al. (1998) found that it simulated the CG flash rate was in reasonable agreement with the observations.

Using observations from a thunderstorm in southern Germany, Fehr et al. (2004) found that the PR93 parameterisation scaled by a factor of 1.10 worked well.

Allen and Pickering (2002) used it in a global chemical transport model, but their evaluation of the IC/CG ratio was limited to the United States. They found that the PR93 parameterisation was realistic and captured much of the variability in the IC/CG ratio. Grewe et al. (2001) used it in a global chemistry-climate model with a focus on predicting tropospheric $NO_x$ and ozone.

An alternative parameterisation for the IC/CG ratio is that by Bond et al. (2002) which is a linear fit to the data reported by Mackerras et al. (1998, J. Geophys. Res., https://doi.org/10.1029/98JD01461), and, like PR93, is a function of latitude. These data were obtained from 11 ground sites using CGR3 (Cloud-Ground Ratio# 3) instruments covering latitudes 59.9°N to 27.3°S between 1986 and 1991. Bond et al. (2002) applied their parameterisation to the global LIS observations from 1998 to 2000 to obtain estimates of the total number of IC and CG

flashes, and found that on average, the IC/CG ratio was 3.76 over the tropics (35°N–35°S). This value is comparable to 3.14 that we obtain from our study using the PR93 parameterisation (notwithstanding the difference in the year(s) of the two studies). The corresponding values for the ratio CG/total flash are 0.21 and 0.24, which are quite similar.

In our ACCESS-UKCA model, the amount of NO produced per flash is the same for both IC and CG flashes and, therefore, the partitioning of the total flash rate into the CG and IC flash rates only influences the shape of the vertical distribution of $LNO_x$ (which is discussed in Section 3.7.2), with the total $LNO_x$ released remaining independent of the partitioning.

Based on the above, we believe that the PR93 parameterisation works well. Obviously, additional data of CG flashes, particularly those covering the globe, would help further constrain $z_R$.

**Changes in manuscript:** We have included in the last paragraph of Section 2.1 (P7L16–23) the following:

"The PR93 parameterisation has been used frequently, with further validation for case studies reported by Pickering et al. (1998) and Fehr et al. (2004). Allen and Pickering (2002) and Grewe et al. (2001) used it in global atmospheric chemistry models, with the former evaluating it for cases in the US. The averaged values of $z_R$ and the CG to total flash ratio obtained from the PR93 parameterisation in the present study are 3.14 and 0.24, respectively. These values are comparable to $z_R \sim 4$ and the CG to total flash ratio $\sim 0.2$ obtained by Barthe and Barth (2008) using $dH$ calculated directly from modelled cloud temperature and total hydrometeor mixing ratio in the PR93 parameterisation. Using IC/CG measurements, Bond et al. (2002) derived a parameterisation for $z_R$ as a linearly decreasing function of latitude and obtained $z_R = 3.76$ and the CG to total flash ratio = 0.21 over the tropics (35°N–35°S)."

In the last paragraph of Section 2.2 (P9L8–11), we add

"Since the amount of NO produced per flash is taken to be the same for both IC and CG flashes, the partitioning of the total flash rate into the CG and IC flash rates only influences the shape of the vertical distribution, with the total $LNO_x$ released remaining independent of the partitioning."

 8)

Section 2.2

The authors seem to assume that the energy of CG and IC flashes is the same. Is it so?. If yes, please state it clearly and add appropiate citations supporting this assumption (for instance Ridley et al 2005, Ott et al 2010 and / or others). It is also assumed that 330 moles NO / flash is produced independently of whether the flash is CG or IC. Why 330 ?.

In this paper the amount of NO per flash is prescribed to 330 moles NO / flash. What is the underlying reason for choosing 330 moles NO / flash instead of the 310 moles NO / flash concluded by Miyazaki et al 2014?.

**Response:** We assume that the amount of NO production per CG flash and IC flash, which is used directly by the model, is the same, and this is now stated clearly in the 1$^{st}$ paragraph of Section 2.2.

There is a default scaling factor in the model that is set to 2 which is multiplied with a base NO production of $10^{26}$ molecules per flash. This yields 330 moles NO per flash. Thus, the scaling factor corresponding to the Miyazaki et al. (2014) value of 310 moles NO per flash will be 1.87.

The assumption that the amount of NO production per CG flash and IC flash is the same is based on the studies by DeCaria et al. (2005), Ridley et al. (2005), Ott et al. (2007, 2010) and Cummings et al. (2013).

There is a large uncertainty in the average NO production per flash reported in the literature: 33–660 moles (Schumann and Huntrieser, 2007) and more recently ~70–700 moles (Bucsela et al., 2019). Based on the verification studies that we cite in Section 3.7.1, the range is 170–665 moles per flash. Given that, we did not think that there was any advantage in changing the default value of 330 moles NO per flash to 310 moles NO per flash concluded by Miyazaki et al. (2014), which are very similar values anyway and lie near the middle of the uncertainty range. Moreover, we are more interested in differences in the impact on composition caused by the different flash rate parameterisations given the same NO per flash value. However, comparing the modelled tropospheric total column $NO_2$ with observations does suggest that if we were to match the average CAMS $NO_2$ column value in Table 5, the new flash-rate parameterisation with 310 moles NO per flash would probably yield a very slightly better prediction than the 330 moles NO per flash value used.

**Changes in manuscript:** We have revised Section 2.2.

The sentence in the 1st paragraph of Section 2.2 is modified to

"In this study, $P_{NO}$ is set at $S_f \times 10^{26}$ molecules NO per flash where the scaling factor $S_f = 2$ by default irrespective of whether a flash is IC or CG, which is equivalent to 330 moles NO per flash."

Additional discussion given in Section 2.2 (P8L11–30):

"The value $P_{NO} = 330$ moles NO per flash used in our model lies close to the middle of the range of current literature. Recent estimates include: a global average value of 310 moles NO per flash obtained by Miyazaki et al. (2014) using an assimilation of multiple satellite measurements of atmospheric composition and the LIS/OTD lightning flash data into a global CTM; 665 moles NO per flash estimated by Nault et al. (2017) using airborne observations of atmospheric composition, satellite based Ozone Monitoring Instrument (OMI) $NO_2$ columns and the GEOS-Chem model; 280 ± 80 moles NO per flash by Marais et al. (2018) using the OMI $NO_2$ columns and satellite based lightning data together with GEOS-Chem; 180 ± 100 moles NO per flash by Bucsela et al. (2019) for three northern midlatitude regions that were primarily continental; and 170 ± 100 moles NO per flash by Allen et al. (2019) for the tropics. The last two stem from the same OMI $NO_2$ columns and ground-based lightning measurements. Values used in calculating global estimates of $LNO_x$ include: 360 moles NO per flash by Ott et al. (2007), and 500 moles NO per flash for selected extratropical regions and 260 moles NO per flash for the rest of the globe by Murray et al. (2012)."

"We assume that both CG and IC flashes yield the same amount of NO, which follows studies such as DeCaria et al. (2005), Ridley et al. (2005), Ott et al. (2007, 2010) and Cummings et al. (2013). On the other hand, some studies consider or find that the less frequent CG flashes yield a greater amount of NO per flash than IC flashes (Price et al., 1997; Koshak et al., 2014; Luo et al., 2017), A few studies suggest that $P_{NO}$ may not be constant over the globe, with higher production rates in extratropics than tropics (Huntrieser et al., 2008; Murray et al., 2012) and globally variable production rates (Miyazaki et al., 2014). Differences in land and ocean production rates have also been noted. Boersma et al. (2005) found that land flashes were ~1.6 times more productive than those over the ocean, and conversely Allen et al. (2019) estimated marine flashes to be twice as productive than those over land. Clearly, further measurements and process understanding are needed to reconcile differences in $LNO_x$ production."

Added in the last paragraph of Section 3.7.3 (P30L3–6) "Clearly, the comparison also depends on the selected value of NO produced per flash. We have used the model default value of 330 moles NO per flash. However, if we were to match the average CAMS column value in Table 5, the new parameterisation with 310 moles NO per flash, the value suggested by Miyazaki et al. (2014), would probably yield a somewhat better prediction."

**9)**

By using equation (15) in Price et al JGR 1997 the authors could estimate (assuming the energy per flash) the number of NO molecules produced per joule. This an interesting magnitude to show and it is possible since they have computed the amount of global LNOx (Table 4, equation (21)) and are assuming that the energies of CG and IC flashes are the same, and that aproximately 75 % of predicted total flashes per second (Table 1) are CGs while 25 % are ICs.

**Response:** Point taken. This has been calculated.

**Changes in manuscript:** In the 1st paragraph of Section 2.2 (P8L2–5), we add "Assuming a mean energy release of 0.67 GJ per IC flash and 6.7 GJ per CG flash (Price et al., 1997), with 24% of the total modelled flashes being CG, the production of 330 moles NO per flash corresponds to $9.4 \times 10^{16}$ molecules NO $J^{-1}$. If we use a mean energy release of 0.9 GJ per IC flash and 3.0 GJ per CG flash based on Schumann and Huntrieser (2007), then the NO production is calculated to be $14.2 \times 10^{16}$ molecules NO $J^{-1}$."

**10)**

In line 16 of page 7, the authors comment a little bit how the produced amount of LNOx is vertically distributed. It is mentioned that it is distributed evenly vertically from 500 hPa (aprox 6 km) to the cloud top for IC flashes, and from surface to 500 hPa for CG flashes. What is the rationale and / or the physical, chemical and transport reasons (and / or possible observations) for choosing / supporting such vertical distribution ?. This is a bit obscure to me.

**Response:** We agree this was a bit obscure. In our model, the calculated amount of $LNO_x$ at a grid point location at a given time step is distributed evenly in the vertical in log-pressure coordinate from 500 hPa to the cloud top for intra-cloud (IC) flashes, and from 500 hPa to surface for cloud-to-ground (CG) flashes. The method is motivated by the data analysis of Price and Rind (1993). Their observations from 139 thunderstorms cover cold cloud thickness (i.e., the cloud top height minus the freezing level) values ranging between 5.5–15 km and freezing level values between 2.7–5 km. The ratio $z$ = IC/CG increases from 0 to 4.6 with cold cloud thickness from 5.5 to 15 km but remains relatively constant with freezing level. We take the level below which the CG generated $LNO_x$ is distributed as the observed minimum freezing level plus half of the minimum cold cloud thickness, i.e. $(2.7+5.5/2) \approx 5.5$ km. The selected 500 hPa level is closest to this 5.5 km value.

The use of the log-pressure (rather than linear pressure) coordinate yields a vertical distribution of $LNO_x$, with more $LNO_x$ released at higher levels.

The non-uniform shape of the averaged modelled vertical distributions in Figure 7 is largely caused by the averaging of the $LNO_x$ profile from every time step over spatial and temporal variations in the cloud-top height.

Our averaged model profiles of $LNO_x$ in Figure 7 compare better with Ott et al. (2010). But we believe thar further measurements are required to better understand the nature of $LNO_x$ distribution in the vertical.

**Changes in manuscript:** New text based on the above is added – see the last paragraph of Section 2.2 (P9L1–11).

**11)**

In line 15 of page 9, it is said "... by substituting Eq. (8) into Eq(13) ...", wouldn't it be the opposite?.

**Response:** Thanks for pointing that out. It should be "…by equating Eq. (14) and Eq. (13) …"

**Changes in manuscript:** As above.

12)

Section 3.5

According to the authors ACCESS-UKCA was setup as a free running simulation for 2 years (2005 and 2006), and the simulation was started using the model initial conditions taken from a nudged model run (see line 10 in page 13). I think the authors should be a bit more specific on this technical matter. I underdtand that the nudging somehow guarantees / ensures that the basic dynamics in the lower-middle atmosphere is identical in simulations in which other changes (implementation of a lightning scheme for instance) are made. Is the nudging applied to all altitudes (pressure levels of the model)?. Also, it is not clearly indicated whether lightning was included (or not) in the first free running simulation. Was it?.

**Response:** Thanks for raising this point. It has been addressed as below.

**Changes in manuscript:** The last paragraph in Section 3.5 (P15L9–17) has been modified to read "ACCESS-UKCA was setup as a free running simulation for 2 years (2005–2006) for each of the above runs, and the simulation was started using model initial conditions taken from a previously spun-up, nudged model run that used a Newtonian relaxation nudging (Uhe and Thatcher, 2015) within model levels 20–45 (between altitudes ~ 3 km to 14 km) and the default lightning scheme. The variables nudged were the horizontal wind components and potential temperature by using ECMWF's ERA-Interim reanalyses (Dee et al., 2011) on pressure levels. The idea was to start the simulation with meteorological/transport errors minimised in the free troposphere to the extent possible. The first year of the free running simulation was used as a spin-up period and the model output for the year 2006 used for analysis reported below. (We also did Runs 1 and 2 with nudging for the years 2005–2006 with the same initial conditions as for the free running simulations, and the results are summarised in Section 5)."

13)

I consider that to "see" the influence of lightning only in a CTM one should proceed as the following: First run your code in a free-running dynamic mode without considering lightning. Then run a second model simulation also without lightning but now using the nudge, that is, the horizontal wind and temperature fields in the tropo-stratosphere are nudged at each model time step of the first free-running dynamic ACCESS-UKCA run. Then, what I would do, is to run a third nudged ACCESS-UKCA simulation (with the lightning scheme on) that is nudged to the first free-running dynamics ACCESS-UKCA run. Finally, I would repeat the third simulation for each of your lightning schemes (PR92 and your new one) and will always compare their output with the results of the second nudged ACCESS-UKCA runs. In this way you will ensure that you are really carrying out comparisons between simulations of the atmosphere with and without lightning that are not biased by dynamical effects.

It is not completely clear to me if your "nudged model run" is really free of dynamical effects. Please comment on this and try to be more specific.

**Response:** Thanks for the comment. There are essentially two points in the above comment:

First, that there should also be a run with no lightning-generated NO$_x$ to see how atmospheric composition is impacted when LNO$_x$ is not considered. We did do such a run (free running), but at the time we did not think that it was necessary to present results from this case given that our aim is not to demonstrate the importance of LNO$_x$ on atmospheric composition by doing a no LNO$_x$ case, which has already been demonstrated in many studies cited in the Introduction (e.g. Grewe et al., 2007; Dahlmann et al., 2011), but more to assess the impact of flash-rate parameterisations on atmospheric composition. However, given that we already have model results from such a run, we now give some broad results from the zero LNO$_x$ emissions case to put changes in the tropospheric composition arising from changes in the flash-rate parameterisations in perspective.

Second, we have already given some information on nudging. The first year of the two-year free running simulation was used as a spin-up period and the model output for the year 2006 used for analysis. We also conducted simulations for the two-year period with nudging and the results are summarised in the new Section 5.

**Changes in manuscript:** We now present volume-weighted, troposphere averaged NO$_x$, O$_3$, OH and CO values obtained from the no-LNO$_x$ emissions model simulation for comparison purposes (in the respective sections).

We now give a new Section 5 on nudging in the model.

**14)**

Section 3.6

As a remark, by looking at Table 1 I see that the output of RUN 1 (PR92) gives quite low global lightning flash frequency (32.92 flashes / s). Note that the UKCA-UM model was already used by Finney et al 2016 and applied a scaling factor of 1.44 to match PR92 global flash frequency to LIS/OTD observations. In your case the scaling factor would be 1.40.

**Response:** This sounds correct. However, as elaborated in a response below, applying this scaling factor to the PR92 flash rate would yield the correct global flash rate compared to the LIS/OTD observations, but in the process, it would ruin the good agreement in flash rate over land compared to the observation (and this is because the PR92 oceanic parameterisation is deficient).

**Changes in manuscript:** None.

**15)**

Could you please explain a bit the underlying reasons for your model runs (including RUN 2) to underpredict in spring in the SH and NH and overpredict in autumn in the SH (see Figures 3 a/b) ?.

**Response:** We can only speculate as to the reasons for this. The underprediction in spring in the Northern Hemisphere and overprediction in autumn in the Southern Hemisphere could be due to a displacement of lightning activity across the equator. The underprediction in spring in the Southern Hemisphere appears to be due to model deficiency over land (see also the next response below).

**Changes in manuscript:** We have now stated the above in Section 3.6 where the results are compared (P17L13–P18L1).

**16)**

Both PR92 and the new lightning scheme proposed fail in accurately describing the tropical oceanic flash rate (see Fig. 4c). There is a considerable overestimation of RUN 1 (PR92) and RUN 2 (new lightning scheme). What is the reason for this?. This has consequences on the simulation results shown in Fig. 5a (observations) and Fig. 5d (new scheme) where the tropical oceanic overestimated flash rate is apparent. Please comment a bit on this behaviour.

Regarding land, note that North America, the Indian and Australian continents are not very much well described either in RUN 1 and RUN 2 (new scheme). Please give reasons for this.

**Response:** Figure 4c shows that although the new oceanic scheme (Run 2) has improved the prediction of flash rate over the ocean compared to Run 1 (PR92), significant latitudinal/spatial differences remain compared to the observations. The modelled latitudinal distribution is narrower and more peaked than the data over the tropical and beyond, particularly over the ocean. Similarly, as the Referee has pointed out, there are significant differences over North America (particularly the US), India, and Australia.

We add (P20L21–P21L1) "It is remarkable that the simple PR92 scheme based on the convective cloud-top height is able to simulate the broad observed global distribution of flash density over land at low latitudes (except parts of India), but does not properly reproduce the extension of lightning flash density into the temperate latitudes, particularly in the Northern Hemisphere."

It is hard to pinpoint the particular reasons for the model-data differences, but we have mentioned some generic factors that could potentially be responsible (P19L20–P20L1) – "The reason for this may be the inherent limitation of the simple flash parameterisation approach based on convective cloud-top height or uncertainty/biases in the modelled convection (e.g., Allen and Pickering, 2002; Tost et al., 2007). Another potential factor could be greater vertical wind shear outside the tropics which extends the horizontal lightning channel length (Huntrieser et al., 2008), which is not accounted for in the cloud-top height-based approaches."

We also add (P20L1–3) "The LIS/OTD observations have some limitations too, such as a short sampling duration (just minutes) for a particular global location and lightning detection efficiencies not being perfect (Clark et al., 2017)."

**Changes in manuscript:** As above.

17)

In commenting the use of scaling factor for flash frequency (line 1 and 2 of page 20) you should also cite the works by Tost et al 2007, Finney et al 2016 and Clark et al 2017 (among others) that applied such scaling factors in different models.

**Response:** Point taken.

**Changes in manuscript:** The references have been cited.

18)

Regarding scaling for NO produced per flash, authors have prescribed an amount of 330 moles NO/flash which immediately conditions the desired lightning generated NOx (LNOx) as can be clearly seen from equation (21). Any comment on this ?.

**Response:** That is correct, but we have added (P22L1–2) that "If the NO production per flash differs for IC and CG flashes then $P_{NO}$ can be taken as a weighted average over mean IC and CG flash fractions."

**Changes in manuscript:** As above.

The authors are assuming that all lightning flashes produce 330 moles NO / flash (no matter if CG or IC and independently of occurring in land or ocean). However, it is known that CG strokes over water usually carry more charge into them which leads to a higher transported current. This is an indication that, on average, CGs over water are more energetic than CGs over land and, consequently, CGs over ocean would produce a larger LNOx (see the paper by Nag and Cummings in GRL 2017). The latter is an indication of different land / ocean convection regimes. This is not considered by any lightning scheme (quantifiying the occurrence rate, not the energy). Authors should add comments on these deficiencies so that readers can have a fair perspective of the many limitations of lightning schemes (any).

**Response:** We think that currently there is no agreement whether CG and IC lightning flashes produce the same amount of NO or whether CG flashes produce more. As we state in the paper, some studies consider or find that the less frequent CG flashes yield a greater amount of NO per flash than IC flashes (Price et al., 1997; Koshak et al., 2014; Luo et al., 2017), whereas in others both CG and IC flashes yield approximately the same amount of NO on average (DeCaria et al., 2005; Ridley et al., 2005, Ott et al., 2007, 2010; Cummings et al., 2013), as is assumed in the present study. A few studies suggest that the NO production per flash may not be constant over the globe, with higher production rates in extratropics than tropics (Huntrieser et al., 2008; Murray et al., 2012) and globally variable production rates (Miyazaki et al., 2014).

Differences in land and ocean production rates have also been noted. Boersma et al. (2005) found that land flashes were ~1.6 times more productive than those over the ocean, while Allen et al. (2019) estimated marine flashes to be twice as productive than those over land.

The study by Nag and Cummings (2017, Geophys. Res. Lett., https://doi.org/10.1002/2016GL072270) that the Referee mentions suggests higher first stroke peak currents for lightning occurring over ocean than land. This finding is based on an analysis of lightning data from five circular regions, each with 50 km diameter, over land and ocean in Florida. While it is an interesting study, it is not related to NO production and it is not clear how the results are directly applicable, or how they can be extrapolated, to the present global study (although one may infer that higher first stroke peak currents means higher NO production per flash).

In summary, we think that more research is needed to understand the characteristics and variability of CG and IC lightning flashes, particularly from the point of view of NO production, and to incorporate this understanding in global chemistry models.

**Changes in manuscript:** We have added new text in 2nd to 4th paragraphs of Section 2.2 (P8L6–30).

Section 3.7

Subsection 3.7.1

In my view the lack of scaling flash frequencies (and the fact of using a prescribed P_NO = 330 moles NO per flash) artificially magnifies the difference between the PR92 LNOx (4.8 Tg N / yr) and the one resulting from the new lightning scheme (RUN 2) leading to 6.6 Tg N / yr. If authors

would have scaled (to match observations) the flash frequencies of each tested lightning scheme (especially the one of PR92), the resulting LNOx of PR92 and TS1 would have been much closer.

**Response:** It would be incorrect to apply a scaling factor to the flash rate (or frequencies) to get the total LNO$_x$ right unless the global spatial distribution of the flash rate is correct so that it can be scaled at every location by the same factor. For example, if we were to obtain the total LNO$_x$ of 6.61 Tg N from the PR92 scheme, we would need to apply a scaling factor of 6.61/4.84 = 1.37 to the PR92 flash rates. Based on the flash rate values in Table 1, this would give a globally averaged PR92 flash rate of 1.37*32.92 = 45 flashes per second, the same as the new scheme, but partitioning of that over land and ocean would give 1.37*32.56 = 44.6 and 1.37 * 0.36 = 0.49 flashes per second, respectively, compared to the new scheme values of 35.88 and 9.08 flashes per second, respectively. So, although the scaling of PR92 now gives the right total LNO$_x$, it has unrealistically amplified the flash rate over land, which is obviously not supported by the LIS/OTD data in Table 1. Thus, global scaling does not fix the problem with the PR92 or any other scheme and spatial mismatches over land and ocean remain (leading to errors in the predicted LNO$_x$ distribution) as long as the relative flash rates over land and ocean remain incorrectly parameterised. One solution is to develop improved flash rate parameterisations, as has been attempted in the present paper.

**Changes in manuscript:** We have already provided some text on this in Section 3.6. See the paragraph starting with "Modelled flash rates depend critically on modelled…" (P22L6–17).

21)

In connection with this, I miss a deep discussion on the reasons for selecting 330 moles NO / flash. For example, there are recent papers (not cited by the authors) by Bucsela et al JGR-Atm 2019 and Allen et al JGR-Atm 2019 where, based on OMI + WWLLN observations, find that LNOx can be 180 moles NO / flash +- 100 in midlatitudes summertime NH. Complementarily, the paper by Allen et al 2019 finds that LNOx can range between 70 and 270 moles NO / flash in the tropics.

**Response:** As was mentioned above (see Point 8 above), there is a large uncertainty in the average NO production per flash reported in the literature: 33–660 moles (Schumann and Huntrieser, 2007) and similarly 70–700 moles (Bucsela et al., 2019).

More recently, using airborne observations of atmospheric composition, satellite-based OMI NO$_2$ columns and the GEOS-Chem model, Nault et al. (2017) estimated 665 moles NO per flash. Marais et al. (2018) used the OMI NO$_2$ columns and satellite-based lightning data together with GEOS-Chem and derived a global production rate of $280 \pm 80$ moles NO per flash. The estimates of $180 \pm 100$ moles NO per flash by Bucsela et al. (2019) and $170 \pm 100$ moles NO per flash by Allen et al. (2019), which essentially stem from the same OMI NO$_2$ columns and WWLLN ground-based lightning measurements, are on the lower side of the above ranges.

To estimate global LNO$_x$, Ott et al. (2007) used 360 moles NO per flash, whereas Murray et al. (2012) used 500 moles NO per flash for selected extratropical regions and 260 moles NO per flash for the rest of the globe.

Our default value of 330 moles per flash lie close to the middle of the above ranges and is very similar to the global average value 310 moles per flash derived by Miyazaki et al. (2014) using data assimilation.

While we are more interested in investigating the differences in the impact on composition caused by the different flash rate parameterisations given the same NO per flash value, our comparison of the modelled tropospheric total column NO$_2$ with observations suggests that if we were to match the average CAMS NO$_2$ column value in Table 5, the new flash-rate parameterisation with 310

moles NO per flash as used by Miyazaki et al. (2014) would probably yield a somewhat better prediction than the 330 moles NO per flash value used by our model.

One reason for the large uncertainty is the fact that lightning is inherently a complex process and specifying a single NO production rate per flash is probably too simplistic. Clearly, further advances in both measurements and process modelling/parameterisations are needed to improve the representation of lightning in atmospheric chemistry models.

**Changes in manuscript:** Please see the new text in 2[nd] to 4[th] paragraphs of Section 2.2 (P8L6–30), and also the related responses above (Point 8).

The references mentioned by the Referee have now been cited.

22)

I disagree with the sentence in lines 11-12 of page 22 that the new flash-rate parameterization (Fig. 6b) agrees better with annual LNOx distribution obtained by Miyazaki et al 2014 (Fig. 6c). There are large land geographical regions (North America, Australia, India, EuroAsia) where the predicted LNOx by PR92 and the new scheme are pretty similar and very different with respect the LNOx distribution derived by Miyazaki et al 2014 (Fig. 6c). This is mainly due to the very different flash densities of both PR92 and RUN 2 (Fig. 5c and 5d) compared to observations (Fig. 5a). So, the global flash frequency (and LNOx) could be similar to considered observations (LIS / OTD and Miyazaki's 2014) but, to me, a more demanding comparison would require detailed comparison of flash frequencies (and LNOx) per continental region (North America, South America, Africa, EuroAsia, ...). Could the authors provide a Table showing such comparison?.

**Response:** We much appreciate this comment. It prompted us to do a couple of things. First, we asked Dr Kazuyuki Miyazaki (now at Jet Propulsion Laboratory) if he could supply the data used in producing the middle-left plot in Figure 6 in Miyazaki et al. (2014) that we presented as Figure 6c in our paper for comparison purposes. Dr Miyazaki kindly supplied us with the data, but also mentioned that the units were incorrect in their paper – they should be $10^{-13}$ kg N m$^{-2}$ s$^{-1}$ instead of $10^{-12}$ kg N m$^{-2}$ s$^{-1}$. We double-checked this by doing a global sum of the LNO$_x$ data supplied and it comes out to be 6.3 Tg N per year as reported in their paper. We now replace the old plot (Figure 6c) by a new plot based on the data supplied by Dr Miyazaki, with the correct units noted in the figure caption and with the same colour scheme as our model plots.

Second, our model plots (Figure 6a and 6b) were not correct either – there was a multiplication factor missing in the analysis script. We now present the corrected plots and have double checked them by doing a global sum of LNO$_x$.

In view of the Referee's comment, together with the above corrections, we have revised the text as indicated below.

Given that the Miyazaki et al. LNO$_x$ data are only assimilated model fields with associated uncertainties and not direct measurements, it is not useful to do a full quantitative comparison for different parts of the globe. Also, it is clear that the new flash-rate scheme mainly differs from the PR92 over the ocean, so the oceanic component remains the focus of the comparison. Nevertheless, we have now calculated the total LNO$_x$ from the Miyazaki et al. plot for the Northern Hemisphere, Southern Hemisphere, land and ocean, and these are compared with the modelled values reported in Table 4.

**Changes in manuscript:** Figure 6c revised based on the data supplied by K. Miyazaki. The text revised to read (P25L1–16):

"The modelled mean global distributions of LNO$_x$ from the two runs presented in Figure 6a and b are essentially in proportion to the flash density distributions given in Figure 5c and Figure 5d, respectively. The new flash-rate scheme (Run 2) leads to a larger and broader distribution of LNO$_x$ over the ocean compared to the PR92 scheme, while over land they are very similar."

"In the absence of any direct measurements of global spatial distribution of LNO$_x$ for comparison we present in Figure 6c the annual LNO$_x$ distribution obtained by Miyazaki et al. (2014) using an assimilation of satellite measurements of atmospheric composition and the LIS/OTD lightning flash data into a global CTM for the year 2007. This plot is a reproduction of their Figure 6 (middle-left plot) based the data[1] supplied by K. Miyazaki (personal communication, 2020) at a horizontal resolution of 2.8° × 2.8°. Over the ocean, the new flash-rate scheme (Figure 6b) agrees much better with the assimilated field than does the PR92 scheme, but is clear that the oceanic LNO$_x$ distribution in the plot with assimilation is broader, more diluted in the tropics, and even extends to high latitudes which is not seen in Figure 6b nor indicated by the observed flash-rate distributions in Figure 5a and Figure 5b (this could be due to limitations of the data assimilation used). Over land, the LNO$_x$ distributions predicted by both PR92 and the new scheme are similar and broadly agree with Figure 6c at low latitudes (except parts of India), but do properly not describe the extension of LNO$_x$ into the temperate latitudes, particularly in the Northern Hemisphere. Figure 6c yields a total LNO$_x$ of 6.36, 3.67, 2.69, 5.58 and 0.78 Tg N yr$^{-1}$ for the globe, NH, SH, land and ocean, respectively, which except for SH are closer to the Run 2 values than to the Run 1 values in Table 4. Direct and more extensive measurements would be necessary for a better evaluation of the predicted LNO$_x$ distribution."

23)

Subsection 3.7.2

The vertical distribution of LNOx is crucial. The authors compare their chosen vertical distribution with those of Pickering et al 1998 and Ott et al 2010. The paper adopts an alternative vertical distribution closer to Ott's. However I miss a full discussion explaining / supporting the reasons that moved the authors to use the vertical LNOx distribution (blue dots) shown in Fig. 7.

Please comment and justify your election of vertical distribution. Do you have supportive observations?. Why do you use these profiles?.

The relative energy of global ICs with respect to global CGs has consequences and / or conditions the LNOx vertical distributions. For instance, the vertical LNOx introduced by Pickering et al 1998 is consistent with their election of IC flashes being 10 % as energetic as CG flashes. In fact, according to Pickering et al 1998, if global IC flashes contained less than 10 % energy as CG flashes, the upper troposphere (UT) peak (upper part of the "C-shaped" distributions) in the mass profiles might not be as pronounced. Consequently, if ICs are equally energetic as CGs (as authors have assumed) the UT peak would be even more significant and this is not consistent with the vertical distribution used by the authors that rather seems to be a kind of mean between Pickering's and Ott's distribution. But, again, what are your physical / chemical / transport reasons supporting such profiles?.

**Response:** We have already provided additional details on the vertical distribution of LNO$_x$ in our model in one of our responses above (Point 10). As mentioned, unfortunately there are no direct measurements to verify the modelled LNO$_x$ profiles, so they are essentially unconstrained.
* * *
[1] The units in Miyazaki et al.'s (2014) plot are incorrect – they should be 10$^{-13}$ kg N m$^{-2}$ s$^{-1}$ instead of 10$^{-12}$ kg N m$^{-2}$ s$^{-1}$ (K. Miyazaki, personal communication, 2020). The reproduced **Error! Reference source not found.**c has the correct units.

As mentioned previously, we do not even know definitively whether CG and IC lightning flashes produce the same amount of NO or whether CG flashes produce more. Similarly, the vertical distribution of $LNO_x$ is another component that has considerable uncertainty and disagreement between studies. For example, as mentioned in the paper, the profiles of Pickering et al. (1998) show peaks near the surface and in the upper troposphere (the so-called 'C-shaped' profile), whereas those by Ott et al. (2010) show very little $LNO_x$ mass in the boundary layer with the majority of $LNO_x$ remaining in the middle and upper troposphere (the so-called 'backward C-shaped' profile). Due to the lack of direct measurements, we do not think there is an objective way to establish as to which of the profile shapes is correct or if there is a variability in the profile shape. Our aim in this section was mainly to compare our model profiles with what has been reported in the literature.

We think that additional measurements and analysis are necessary to make further progress on the vertical distribution of $LNO_x$.

**Changes in manuscript:** Additional details on the vertical distribution of $LNO_x$ in the model is provided in response to a previous comment on this topic (see Point 10). The section concludes with "There are no direct measurements to verify any of the $LNO_x$ profiles and we believe further work is needed to constrain them."

 24)

Subsection 3.7.3

While I understand the authors' reasoning for tropospheric NO2 verification, I do not fully agree with your conclusions of this section.

As I see it, the conflict in your procedure starts in line 6 of page 25. Here you indicate that since N_v_trop_180 is not available from observations, you take the average of the curves (in Figure 8) showing the predicted N_v_trop_180, that is, the mean tropospheric NO2 column vs latitude resulting from RUN 1 (PR92) and RUN 2 (new scheme) over the reference longitude of 180 degrees in 2006. Doing this somehow "contaminates" the reference, that is, the CAMS data. This "contamination" leads to curves like the ones shown in Fig. 9 where, inevitably, RUN 1 and RUN 2 for global, land and oceanic scenarios are strangely close to the CAMS values (considered as reference).

Do Fig. 9 show total NO2 columns or only the lightning contribution to the zonal annual-mean tropospheric NO2 column ?. If total, please state it clearly.

I miss comparison of your NO2 values (shown in Fig. 9) with NO2 values reported in Bucsela et al 2019 (see Fig. 3(a) there) from OMI + WWLLN observations in northern midlatitude regions.

Please elaborate on this a bit.

**Response:** We agree with the Referee, and a similar point was also raised by Referee #2. We have done the following.

These are total tropospheric NO₂ columns.

With regards to comparing the tropospheric NO2 columns, since we did not have $N_{v,trop,180}$ directly from observations, we used the model generated latitudinal variation of $N_{v,trop,180}$ in the derivation of the 'observed' $N_{v,trop}$. The quantity $N_{v,trop}$ thus obtained was then used to compare with the modelled $N_{v,trop}$. But, as the Referee has rightly pointed out, this approach influences the model-data comparison because the data then partially depend on the model results which in turn biases the comparison in favour of a better model performance.

We have now used a much more justifiable approach whereby we calculate $N_{v,trop,180}$ directly from the Ozone Monitoring Instrument (OMI) satellite data of tropospheric $NO_2$ columns (http://www.temis.nl/airpollution/no2.html) and use this in the CAMS reanalysis data to obtain $N_{v,trop}$. With this, the model performance does not turn out to be as strong as before (as expected), but there is no change in the overall conclusion from the model-data comparison.

Bucsela et al. (2019) who used the Ozone Monitoring Instrument (OMI) $NO_2$ column data in their work focus on $LNO_x$ in three northern midlatitude regions that are primarily continental (i.e. North America, Europe, and East Asia). It is clear from our work that the major differences in the flash-rate parameterisations are over the ocean, whereas over land the PR92 (which has been evaluated in many studies) and new formulae give very similar results. In any case, we have also used the OMI data now in a certain way (i.e., to obtain $N_{v,trop,180}$) and this is described in Section 3.7.3.

**Changes in manuscript:** The quantity $N_{v,trop,180}$ is now calculated using the Ozone Monitoring Instrument (OMI) satellite data of tropospheric $NO_2$ columns (http://www.temis.nl/airpollution/no2.html; Boersma et al., 2017, 2018) and the model-data comparison is revised accordingly.

The pertinent Section 3.7.3 has been fully revised, including revised Figures 8 and 9 and Table 5, and additional references of Boersma et al. (2017, 2018). We have also changed the section heading from "Tropospheric $NO_2$ verification" to "Modelled tropospheric total column $NO_2$ and validation".

 25)

Section 4 / Impact on chemical tropospheric composition

Let me start by indicating that in this section I miss a more detailed discussion on explicit chemical reactions and species in the context of the production / loss of the lightning affected species (NOx, O3, OH and CO) in the different geographical regions. As mentioned in line 32 of section 2, the model includes 306 chemical reactions and 86 species. This chemical set (plus the aerosol chemistry) is quite rich so that key chemical processes could have been pointed out. This is not really done.

Please try to indicate the key processes that, according to the model's reaction set, play the most important role(s) for the formation / loss of each of the species investigated. This is very important and illuminating for the readers.

**Response:** While we appreciate the comment by the Referee, we feel that the emphasis of this section (which is also reflected in the title of the paper) is to present the impact of the new flash-rate parameterisations on tropospheric composition relative to the default PR92 parameterisation and compare the results where appropriate with available observations. The UKCA model's chemistry scheme is already described by Archibald et al. (2020) (and references therein), who also present a comprehensive evaluation of the model for different geographical locations, and at https://www.ukca.ac.uk, and these are already cited in our paper. Additionally, most global atmospheric chemistry models (e.g., those cited in the Introduction) have the same key processes concerning tropospheric chemistry, and these have been reported widely in the scientific literature. Therefore, we think that the details suggested by the Referee would not add something new to what is already available in the scientific literature.

But, as indicated below, where possible we briefly indicate the chemical reaction(s) that are thought to be most relevant.

**Changes in manuscript:** As described below, we briefly mention relevant chemical processes concerning ozone, OH and CO.

Section 4.1 (NOx)

As said above I think that the comparison between modelled tropospheric NO2 columns and observations shown in section 3.7.3 is not completely convincing.

Here you compare the total tropospheric NO2 colums resulting from PR92 and the new lightning scheme and its difference. I think it would have been clearer for readers to show only the corresponding lightning contributions to the tropospheric NO2 column.

**Response:** We agree that the $NO_2$ comparison in Section 3.7.3 was not completely convincing. As described in our response to Subsection 3.7.3 above (Point 24), we have revised the comparison by using the OMI data and making the CAMS reanalysis data independent of the model.

We think that showing the difference in total $NO_x$ between the two runs is more appropriate because this is the eventual impact on the tropospheric $NO_x$. Because the two runs only differ in their treatment of lightning, the total $NO_x$ difference (in Figure c) should be very close to the lightning only $NO_x$ difference anyway.

**Changes in manuscript:** See the changes made in Section 3.7.3 above (see Point 24).

Section 4.2 (O3)

Could you please indicate the explicit chemical mechanisms that (according to the adopted chemical set) are controling the balance of O3 at 20 m and at 6400 m due to lightning activity ?. What are the key chemical processes controlling ozone population at the two considered reference altitudes?. Are they the same or different?. This is an interesting information not commented in the paper.

Why, according to the authors, the new lightning scheme is not really able to account for the O3 observations in Fig. 12(c) and Fig. 12 (e)?.

**Response:** Thanks for the question about the chemical mechanisms/processes, but we believe that a full answer is very complex and outside the scope of the present "lightning flash-LNOx" paper. Our position is that the changes in atmospheric composition due the differences in the lightning flash-rate parameterisations are noted in the paper but investigating the detailed chemistry causing them would require a separate study.

There are several references that give a comprehensive coverage of tropospheric ozone, with a couple of more recent ones being Monks et al. (2015, Atmos Chem Phys, https://doi.org/10.5194/acp-15-8889-2015) and Archibald et al. (2020, Elem Sci Anth, https://doi.org/10.1525/elementa.2020.034).

We can point to some broad mechanisms that could potentially explain the differences between $O_3$ from Run 1 (PR92) and Run 2 (TS1).

At 20 m (Fig. 11a), because $LNO_x$ is increased in Run 2 by using the new flash-rate parameterisation, mostly through the oceanic formula, $O_3$ increases virtually everywhere in the Southern Hemisphere, particularly over the tropical Pacific and Indian Oceans. This behaviour is influenced by low ambient $NO_x$ concentrations where the $O_3$ production increases with NO

concentration. $O_3$ is produced through photodissociation of $NO_2$ which is produced through oxidation of NO by $HO_2$ and $RO_2$ radicals (e.g., $NO + HO_2 \rightarrow NO_2 + OH$). In the Northern Hemisphere, the increase in $O_3$ is less beyond the tropic, partly because the smaller oceanic area results in a smaller increase in $LNO_x$ by the new oceanic flash-rate. Carpenter et al. (1997, J Geophys Res, https://doi.org/10.1029/97JD02242) suggest that the tropospheric production potential of the Southern Hemisphere is more responsive to the availability of NO than that of the (more polluted) Northern Hemisphere.

At the 6400-m altitude (Fig. 11b), there is a greater increase in $O_3$ compared to that near the surface, because most $LNO_x$ emissions occur in the middle to upper tropical troposphere where the photochemical production of ozone is most efficient.

Transport and deposition processes would also influence the $O_3$ concentration distribution.

With regards to Fig. 12c and Fig. 12e, there could be other reasons, such as the model not getting the transport correctly, particularly from polluted sources (e.g., North America), differences in emissions and distributions, and lightning emissions may not be at the right locations. Additionally, with Fig. 12c for Mauna Loa, the relatively large disagreement is likely due to the model resolution issues. Mauna Loa is located at an elevation of 3397 m on an island which is smaller in size than the grid resolution of the model and therefore it is difficult to correspond the sampling height to a particular vertical model level. We used the modelled concentrations from the bottom model level for all sites.

**Changes in manuscript:** In this section, we include additional text, and modify existing material as follows (P32L15–21):

"Tropospheric ozone chemistry is complex, but broadly speaking the $O_3$ increases in the Southern Hemisphere are influenced by low ambient $NO_x$ concentrations where the $O_3$ production increases with NO concentration. $O_3$ is produced through photodissociation of $NO_2$ which is produced through oxidation of NO by $HO_2$ and $RO_2$ radicals (e.g., $NO + HO_2 \rightarrow NO_2 + OH$). In the Northern Hemisphere, the increase in $O_3$ is less beyond the tropic, partly because the smaller oceanic area results in a smaller increase in $LNO_x$ through the use of the new oceanic flash-rate parameterisation. Carpenter et al. (1997) suggest that the tropospheric production potential of the Southern Hemisphere is more responsive to the availability of NO than that of the (more polluted) Northern Hemisphere."

On P32L23–25,

"This is because most $LNO_x$ emissions occur in the middle to upper tropical troposphere where the photochemical production of ozone is most efficient."

The model resolution difficulties in simulating Mauna Loa are highlighted on P33L13–15.

28)

Section 4.3 (OH)

Could you please show only the lightning contribution to the total OH tropospheric column ?. It is also important to show readers what are the crucial chemical reactions due to the increase of OH at 20 m and 6400 m.

The authors openly admit that the UKCA StratTop configuration produces an overestimation of OH. It would be interesting for readers if the authors could dig into their chemical scheme and indicate what chemical processes could be playing a role (or could somehow explain) the modelled overestimation.

Please comment.

**Response:** We have focused on showing the difference in tropospheric composition as a result of the use of the new lightning flash-rate parametrisation over the default PR92 parameterisation. Consequently, we have been selective in what can be usefully presented, particularly additional figures. With regards to the comment on showing only the lightning contribution to the total OH tropospheric column, we think it would suffice to give the total tropospheric OH burden without any lightning, i.e. $7.6 \times 10^5$ molecules cm$^{-3}$, and that way the differences between this and the Run 1 and Run 2 values $10.6 \times 10^5$ and $12.0 \times 10^5$ molecules cm$^{-3}$, respectively, obtained with lightning can be compared and used to determine the lightning only contribution to the total OH burden.

Again, like ozone, it is difficult to elaborate on all the complex chemical mechanisms/cycles that are relevant for OH and that may explain the differences between OH from Run 1 and Run 2 at the two altitudes. But it is clear that the broad hemispheric differences in OH are qualitatively similar to those for $O_3$. With an increase in NO due to the new flash-rate parameterisation, OH increases (e.g., via the recycling of $HO_2$ by reaction with NO, $NO + HO_2 \rightarrow NO_2 + OH$). There is some decrease in OH, particularly in parts of the Northern Hemisphere at 20 m (Figure 17a). In highly polluted air, $NO_2$ can be an OH sink (Lelieveld et al., 2016). Of course, transport would also influence these patterns, both horizontally and vertically

The observation that the UKCA StratTrop configuration yields substantially larger OH in the Northern Tropics at low altitudes compared to observations and to the ACCMIP multi-model estimates is due to Archibald et al. (2020), which is a general tendency of the model and not specifically attributed to the change in the LNO$_x$ in the present paper. It is difficult to be definitive as to what chemical processes could be playing a role in in this overestimation, but broadly speaking this may be at least partly due to possible differences in precursor emissions compared to reality and the ability of the photolysis scheme used in the model. The reasons for composition biases in the model are continually being investigated, and improvements to aspects of UKCA is an ongoing process involving several research groups. We anticipate further research reporting on these model aspects in the future.

**Changes in manuscript:** In this section, we include additional text and modify existing material as follows (P38L16–19):

"The broad hemispheric differences in OH are qualitatively similar to those for $O_3$. With an increase in NO due to the new flash-rate parameterisation, OH increases (e.g., via the recycling of $HO_2$ by reaction with NO). In highly polluted air, $NO_2$ can be an OH sink (Lelieveld et al., 2016). Of course, transport would also influence these patterns, both horizontally and vertically."

On P39L6–9:

"Overall, we find that, with the new flash-rate parameterisations, there is a 13% increase in the annual-average volume-weighted global tropospheric OH, from $10.6 \times 10^5$ to $12.0 \times 10^5$ molecules cm$^{-3}$. The increase over the ocean is by $1.6 \times 10^5$ (16.3%) and that over land by $0.9 \times 10^5$ molecules cm$^{-3}$ (7.6%). For comparison, the respective values obtained from the model simulation with zero LNO$_x$ emissions are $7.6 \times$, $7.3 \times$ and $8.1 \times 10^5$ molecules cm$^{-3}$."

29)

Section 4.4 (CO)

Are the authors showing in Fig. 17 the total annual-mean tropospheric CO or only the one due to lightning ?.

**Response:** This is total annual-mean tropospheric CO, and this has been made clear.

30)

Recommendation:

This paper reports on a improved CTH-based lightning scheme with the maritime lightning flash frequency being more realistic that the one of the PR92 lightning parameterization. The paper could be published in ACP but only after the authors have appropriately answered the questions and comments that I have addressed. There a number of points that need clarification and improvement before this manuscript can be accepted.

**Response:** We really appreciate your helpful comments.

---

## Author Comment (AC5) · 13 Mar 2021

**Reply by the authors to Dr Declan Finney's comment on**
**"Assessing and improving cloud-height based parameterisations of global lightning flash rate, and their impact on lightning-produced NO$_x$ and tropospheric composition" (#acp-2020-885)**

Comment:

Thank you to the authors for their work. I appreciate their bringing together these details regarding the cloud-top height scheme, and its importance in atmospheric chemistry models.

I have a comment relating to p3(L26-29) and p25(L29-31).

From the evaluation of Clark et al. (2017), I would say that there is not much between the PR92 scheme and the Yoshida et al. (2009) scheme based on cold-cloud depth (CCD). The CCD scheme does, however, show a much smaller increase in lightning activity in the climate change projections.

The CCD scheme incorporates the freezing level, and indirectly relates to the climate change effects on cloud structure and cloud ice. Therefore, I see it as an important lightning scheme that includes the popular cloud-top height variable but also doesn't ignore potential changes in cloud structure under climate change.

Have the authors considered the scheme? And I would like to suggest that this alternative approach to modifying the cloud-top height-based lightning scheme is at least presented and discussed in their paper.

Clark et al. (2017) GRL. https://doi.org/10.1002/2017GL073017

Yoshida et al. (2009) JGR-Atmos. https://doi.org/10.1029/2008JD010370

**Response and changes in the manuscript:** We thank Dr Declan Finney for his comment on our work and for raising the point about the cold cloud depth (CCD) approach of calculating lightning flash rate.

The quantity CCD is defined as the convective cloud-top height minus the freezing (i.e., 0°C) level. Based on satellite observations, Yoshida et al. (2009) derived an empirical relationship in which lightning flash rate is proportional to the fifth power of CCD.

Clark et al. (2017) show that the PR92 flash-rate parameterisation, which is solely based on cloud-top height, gives the best spatial correlation with satellite data ($r = 0.83$), followed by the CCD based parameterisation of Yoshida et al. (2009) ($r = 0.80$). So, yes, there is not much difference between the two schemes. In a way, that is understandable. The thunderstorm data analysis presented by Price and Rind (1993, Geophys. Res. Lett., https://doi.org/10.1029/93GL00226) suggests that freezing levels remains relatively constant compared to CCD values, meaning that it is largely the cloud-top height that provides the variation in the lightning flash rate in the CCD based approach. If that is generally true, then both the cloud-top height and the CCD based schemes would perform very similarly, which is what the comparison analysis of Clark et al. (2017) shows.

The study by Clark et al. (2017) also shows, as pointed out by Dr. Finney, that compared to the cloud-top height based schemes considered, the CCD-based scheme yields a much smaller increase in lightning flash density under future projected warming (with the RCP8.5 scenario). They attribute this behaviour to the fact that with warming global temperatures there is an increase in the freezing level in the deep tropics, so the relative increase in CCD is smaller than that in cloud-top height, and hence an increase in flash density projected by a cloud-top height based scheme under a warming climate would generally be greater than that from a CCD based scheme.

For our present study, we did not consider a CCD based scheme; all schemes are based on cloud-top height. Essentially, the CCD approach heuristically adds an additional parameter (i.e. the freezing level) to the cloud-top approach. It is based on the reasoning/assumption that the vertical ice charged region of a convective cloud is a better parameter for representing lightning than cloud-top height alone. It is perhaps the simplest extension of the cloud-top height approach and could potentially quantify lightning flash rate better. However, based on the 'present-day' observations and evaluation given by Price and Rind (1993) and Clark et al. (2017), the two schemes are very similar. Thus, the studies done so far do not definitively tell as to what extent the incorporation of the freezing level represents the influence of cloud structure and cloud ice on lightning flash rate. Clearly, more observations and comprehensive evaluation are necessary to test these schemes further, and to test any potential advantages of the CCD approach.

A full discussion in the paper on the CCD approach versus the PR92 cloud-top height approach would be unbalanced without also discussing other approaches mentioned in the paper, such as those based on convective precipitation and upward mass flux; convective available potential energy (CAPE); maximum vertical velocity and updraft volume; upward cloud ice flux; and combinations of these. Moreover, such a discussion, we think, is outside the scope of our paper which focuses on the cloud-top height based approach (as implied by the title of the paper). However, we have extended the relevant text in the 2$^{nd}$ last paragraph of Introduction (P3L28–P4L7 in the non-tracked version of the revised paper), as follows, to expand on the CCD approach a little:

"For example, Clark et al. (2017) tested flash-rate parameterisations based on cloud-top height, cold cloud depth (CCD), mass flux, convective precipitation rate, and cloud-top height with column-integrated cloud droplet number concentration, in a global model, and found that the PR92 parameterisations had the best correlation with the observations, closely followed by the CCD based parameterisation of Yoshida et al. (2009). The PR92 scheme had a higher value of the spatial standard deviation compared to observations due to a large land-ocean contrast in this parameterisation. The quantity CCD is defined as the convective cloud-top height minus the freezing level. The thunderstorm data analysis presented by Price and Rind (1993) indicates that freezing levels remains relatively constant compared to CCD values, meaning that it is largely the cloud-top height that provides the variation in the lightning flash rate in the CCD based scheme, which suggests that the cloud-top height and the CCD based schemes would perform very similarly."